# H₂O₂ repurposes plant O₂ sensing to regulate post-hypoxia responses

**Salma Akter**[1,2,6], **Monica Perri**[2,6], **Mikel Lavilla-Puerta**[2], **Sophie Lichtenauer**[3], **Yuming He**[2], **Vinay Shukla**[2], **Laura Dalle Carbonare**[2], **Yuri Telara**[2], **Daai Zhang**[2], **Beatrice Ferretti**[2,4], **Dona M. Gunawardana**[1,2], **William K. Myers**[1], **Pedro Barreto**[3], **Beatrice Giuntoli**[5], **Markus Schwarzländer**[3], **Emily Flashman**[2✉] & **Francesco Licausi**[2✉]

Understanding plant molecular responses to flooding is crucial for strategies to increase resilience. Plants respond to submergence-induced low oxygen (hypoxia) through decreased plant cysteine oxidase (PCO) activity, which stabilizes group VII ethylene response factors (ERFVIIs), master regulators of metabolic and anatomic acclimation responses[1–4]. Rapid reoxygenation on desubmergence induces a burst of reactive oxygen species (ROS) generation and metabolic reconfiguration[5,6]; however, how plants mitigate this post-hypoxic stress to facilitate submergence recovery has remained unknown. Here we report that ERFVIIs are also important in post-submergence recovery, remaining stable upon reoxygenation through ROS-mediated PCO inhibition. Stabilized ERFVIIs are retained at hypoxia-responsive promoters, becoming repressors of typical hypoxia marker genes but upregulators of genes involved in ROS homeostasis and oxidative stress protection. Our findings suggest that PCOs and ERFVIIs integrate signals from both oxygen and ROS to coordinate ERFVII stability through submergence-induced hypoxia and desubmergence stress to promote plant survival and recovery.

When plants undergo flood-induced submergence, a reduction in oxygen (O₂) availability (hypoxia) affects their ability to generate ATP through oxidative phosphorylation. In response, plants can rapidly acclimate to hypoxia by switching to anaerobic metabolism to maintain basal ATP production for limited periods of time. This switch is triggered through the activity of ERFVIIs, transcription factors that bind to hypoxia-responsive promoter elements (HRPEs) to increase expression of various genes, including core hypoxia response genes (HRGs) for fermentative respiration[7–10]. Under normoxic conditions, ERFVIIs are degraded by the catalytic activity of O₂-sensing PCO enzymes and the Cys/Arg N-degron pathway; however, ERFVIIs are stabilized in hypoxia owing to reduced PCO activity[1–4].

Although the molecular response to hypoxia has been well characterized, how plants tolerate stress associated with desubmergence is less clear. During hypoxia, ROS production—in particular, superoxide ($O_2^{\bullet-}$) and hydrogen peroxide ($H_2O_2$)—starts to increase owing to incomplete O₂ reduction at electron transport chains, as well as the activity of NADPH oxidases such as RBOHD[11,12]. Upon reoxygenation, reactivation of mitochondrial and photosynthetic activities involving proteins that may have been damaged during hypoxia causes electron leakage in the electron transport chains and membrane-associated processes[5], which further increases ROS formation, culminating in a ROS burst[6]. Given that hypoxia stress entails ROS production at its onset and after reoxygenation, there is likely to be cross-talk between cellular responses to both signals. However, whether there is a direct interaction between ROS and the plant oxygen-sensing machinery has remained unknown.

## ERFVIIs are required for post-hypoxia recovery

ERFVIIs have been demonstrated to have crucial roles in modulating response to various stresses[13–15]. We therefore considered that ERFVIIs might also contribute to tolerance of the reoxygenation-associated ROS burst and the probable resulting oxidative stress. We compared recovery from hypoxia and reoxygenation in *Arabidopsis* wild-type and *erfVII* mutant plants by subjecting 7-day-old seedlings to severe hypoxia (0.1% O₂) or normoxia (21% O₂) for 24 h (Fig. 1a). Although we did not observe differences between wild-type and mutant plants at the end of the hypoxic treatment, *erfVII* seedlings demonstrated strongly decreased survival after 4 days of reoxygenation in comparison to the wild type (Fig. 1b,c). Root growth was impaired after reoxygenation in the *erfVII* seedlings (Fig. 1d), resulting in lower seedling biomass accumulation compared with the wild type (Fig. 1e). Repetition of this experiment showed similar differences between *erfVII* and Columbia-0 (Col-0), despite a more severe effect of the hypoxic treatment (Extended Data Fig. 1a–e). When repeating the experiment, we also assessed root viability at the end of the hypoxic treatment using Evans blue staining. This showed increased root tip death during reoxygenation compared with hypoxia alone (Extended Data Fig. 1f,g). Together, these results indicate a potential role for ERFVIIs in coping with reoxygenation stress.

Given previous reports that stabilized *Arabidopsis* ERFVIIs Related to Apetala (RAP)2.2, RAP2.3 and RAP2.12 caused increased levels of ROS-related genes in *Arabidopsis* seedlings[8,13], we speculated that

[1]Department of Chemistry, University of Oxford, Oxford, UK. [2]Department of Biology, University of Oxford, Oxford, UK. [3]Institute of Plant Biology and Biotechnology, University of Münster, Münster, Germany. [4]Department of Pharmacy and Biotechnology, University of Bologna, Bologna, Italy. [5]Department of Biology, University of Pisa, Pisa, Italy. [6]These authors contributed equally: Salma Akter, Monica Perri. ✉e-mail: emily.flashman@biology.ox.ac.uk; francesco.licausi@biology.ox.ac.uk

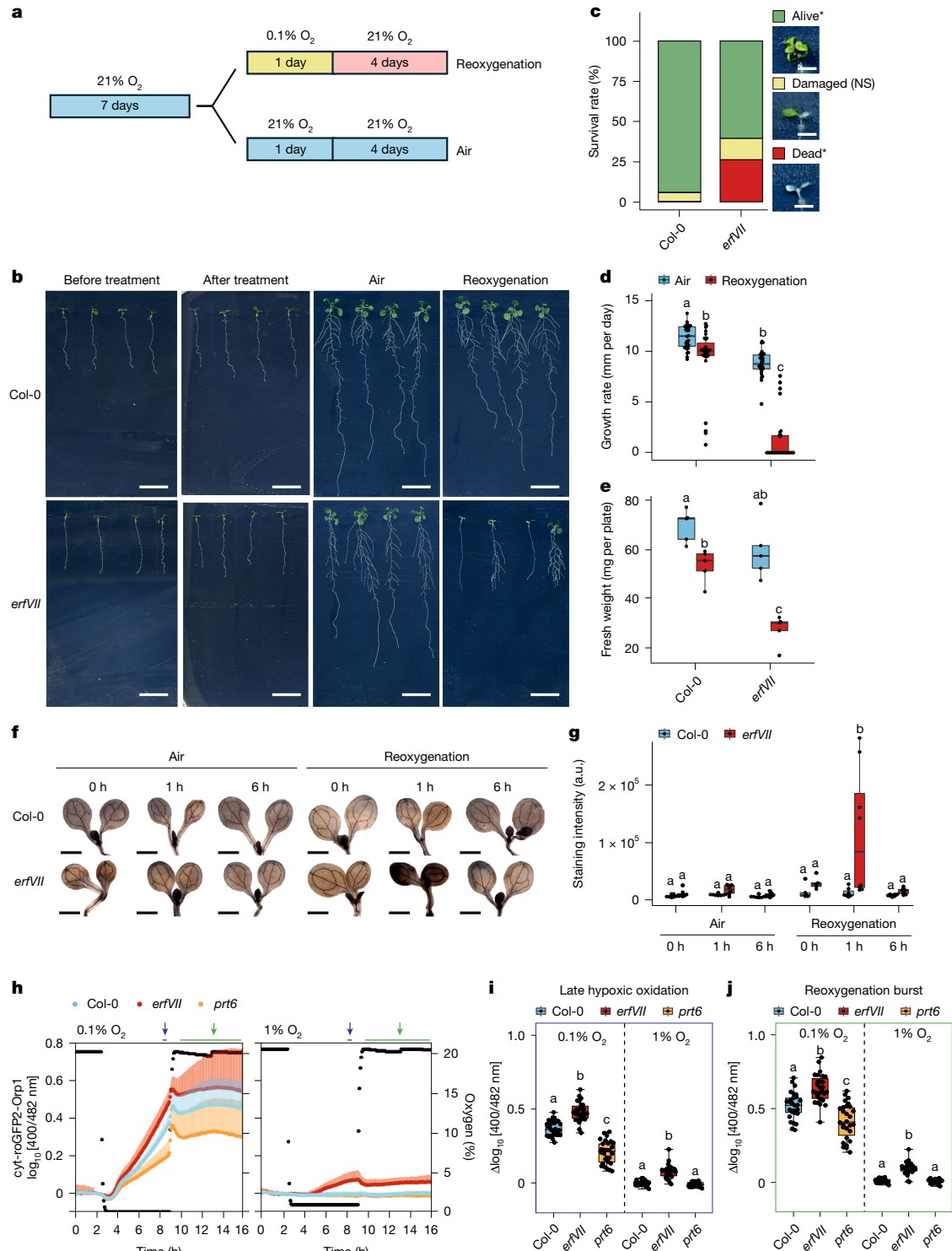

**Fig. 1 | ERFVIIs mediate plant tolerance upon reoxygenation. a**, Schematic of experimental design; 7-day-old *Arabidopsis* seedlings were exposed to severe hypoxia (0.1% $O_2$) or air (21% $O_2$) in darkness for 24 h and subsequently returned to aerobic conditions for 4 days. **b**, Phenotype of Col-0 and *erfVII* seedlings before and after hypoxia treatment or air control and after 4 days of reoxygenation. Scale bar, 1 cm. **c**, Percentage of alive, damaged or dead seedlings after 4 days of post-hypoxia reoxygenation or air control. Two-sided $\chi^2$ test followed by post hoc test with Bonferroni correction was used to analyse this dataset ($P < 0.05$). Asterisks indicate statistical differences between Col-0 and *erfVII*. NS, not significant. **d**, Growth rate of primary roots after 4 days of reoxygenation or air control. **e**, Fresh weight per plate after 4 days of reoxygenation or air control. **f**, DAB staining of Col-0 and *erfVII* seedlings that had been exposed to 0.1% or 21% $O_2$ for 24 h in darkness and subsequently returned to aerobic conditions for 0 h,

1 h or 6 h. Scale bar, 0.5 cm. **g**, Quantification of DAB staining intensity represented in arbitrary units (a.u.). Two-way analysis of variance (ANOVA) followed by Tukey's HSD test ($P < 0.05$) was applied to analyse the datasets in **d**–**g**; different letters indicate statistically distinct groups ($P < 0.05$). **h**, Multiwell fluorimetry of cytosolic oxidative stress using 7-day-old *Arabidopsis* seedlings expressing biosensor roGFP2-Orp1 in Col-0, *erfVII* and *prt6* background over time, each normalized to the baseline oxidative state before the start of hypoxic treatment. **i,j**, Amplitudes of late hypoxic roGFP2-Orp1 oxidation before reoxygenation (**i**, purple arrow in **h**) and maximum oxidative burst during reoxygenation (**j**, green arrow in **h**), each normalized to the baseline oxidative state before the start of hypoxic treatment. Replicate numbers and box plot descriptors for **d**, **e** and **g** are provided in Supplementary Data 1.

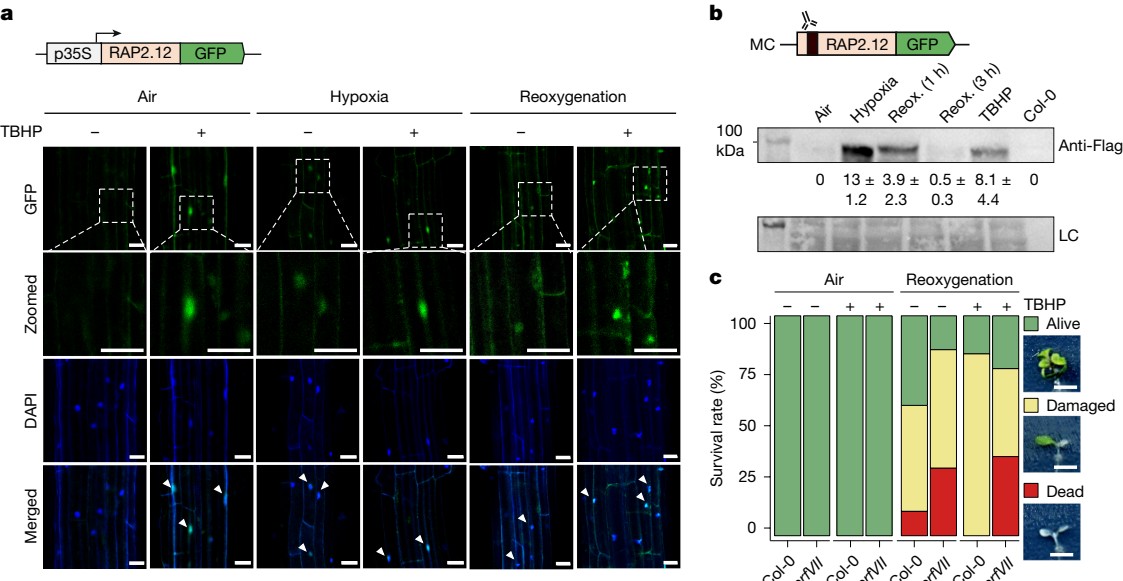

**Fig. 2 | RAP2.12 stabilizes in the nucleus upon oxidative stress as well as during reoxygenation. a**, Localization of RAP2.12–GFP (green) in 7-day-old *Arabidopsis* seedlings upon treatment with 1 mM TBHP under normoxia (21% $O_2$) or hypoxia (1% $O_2$) and after 3 h of reoxygenation. Nuclear localization was confirmed using DAPI staining (blue). White arrowheads indicate nuclear colocalization. Scale bar, 50 μm. This experiment was repeated once. **b**, Western blot analysis of Flag-tagged RAP2.12 abundance in air or under hypoxic conditions (6 h, 1% $O_2$), followed by 1 h or 3 h of reoxygenation (reox.), or 1 mM TBHP treatment. The loading control (LC) corresponds to a compacted image of the membrane after Ponceau staining. The relative protein levels calculated from two independent experiments and mean ± s.d. values are shown below. Unedited gel images are shown in Supplementary Fig. 1. **c**, Percentages of alive, damaged or dead seedlings after 4 days of post-hypoxia reoxygenation or air control, with or without 1 mM TBHP pretreatment for 2 h. Col-0 in air, $n = 22$; Col-0 in air + TBHP and hypoxia + TBHP, $n = 28$; Col-0 in hypoxia, $n = 27$; *erfVII* in air, $n = 21$; *erfVII* in air + TBHP, $n = 23$; *erfVII* in hypoxia, $n = 26$; *erfVII* in hypoxia + TBHP, $n = 25$.

ERFVIIs might limit ROS flux following reoxygenation. Staining with $H_2O_2$ indicator 3′3′-diaminobenzidine (DAB) indicated higher production of $H_2O_2$ in *erfVII* compared with in wild-type seedlings after 1 h of reoxygenation (Fig. 1f,g). To investigate the live dynamics of $H_2O_2$, we used the roGFP2-Orp1 fluorescent biosensor[16] in wild-type, *erfVII* and *prt6* mutant plants during 6 h of severe (0.1% $O_2$) and mild (1% $O_2$) hypoxia followed by reoxygenation (21% $O_2$) (Fig. 1h and Extended Data Fig. 1h–m). We observed a sustained increase in sensor oxidation upon reoxygenation, which was significantly higher in *erfVII* mutant than in Col-0 plants (Fig. 1h,j and Extended Data Fig. 1h,j,k,m). In *prt6* plants, in which ERFVIIs are constitutively stabilized[1,2], sensor oxidation throughout reoxygenation was similar to that observed for Col-0 following mild hypoxia but reduced compared with Col-0 following severe hypoxia (Fig. 1j). These data strongly suggest that ERFVIIs contribute to post-submergence recovery by reducing ROS load upon reoxygenation.

## ERFVIIs are stabilized under oxidative stress

We next investigated the abundance and localization of RAP2.12 in *Arabidopsis* cells after reoxygenation. Seven-day-old transgenic *Arabidopsis* seedlings constitutively expressing RAP2.12 fused with green fluorescent protein (GFP) were exposed to 21% or 1% $O_2$ for 6 h, followed by 3 h of reoxygenation. Confocal imaging revealed elevated RAP2.12–GFP localization in the nuclei of hypoxic plants compared with normoxic plants (Fig. 2a), as expected (Extended Data Fig. 2a); however, the GFP signal also persisted in the nucleus after reoxygenation. To determine whether this persistence was related to intracellular ROS elevation, we applied 1 mM *tert*-butyl hydroperoxide (TBHP), an exogenous ROS donor[17,18]. This resulted in an increased RAP2.12–GFP signal in the nuclei of plants exposed to both air and hypoxia (Fig. 2a). We confirmed these observations using a transgenic line expressing RAP2.3–GFP (Extended Data Fig. 2b,c) and further validated protein stability using Flag-tagged RAP2.12 (Fig. 2b and Extended Data Fig. 2d,e). The stability of *Arabidopsis* ERFVIIs in the nucleus for up to 3 h following post-hypoxia reoxygenation was consistent with exposure to increased ROS production through prolonged hypoxia. Although previous reports have indicated rapid ERFVII decay upon reoxygenation[2,15], this followed short periods of hypoxia (60–90 min); when the duration of hypoxia is ≥3 h (refs. 19,20), ERFVII stability seems to be more prolonged.

We speculated that oxidative-stress-induced ERFVII stabilization could affect seedling survival in reoxygenation tolerance assays. Indeed, pretreatment of seedlings with 1 mM TBHP for 2 h before hypoxia (24 h, 0.1% $O_2$) increased post-reoxygenation survival in Col-0 seedlings but not in *erfVII* mutants (Fig. 2c and Extended Data Fig. 2f,g). This suggests a role for ROS in priming oxidative stress tolerance during reoxygenation through ERFVIIs.

To obtain a quantitative measure of ERFVII stability, we next generated an *Arabidopsis* line constitutively expressing the full RAP2.3 coding sequence (CDS) fused in-frame with the nanoluciferase enzyme (RAP2.3–nLuc). Luciferase activity was measured in 7-day-old RAP2.3–nLuc seedlings treated with or without TBHP (1 mM) in normoxia, 6 h hypoxia (1% $O_2$) and 3 h reoxygenation conditions (Fig. 3a). TBHP-treated seedlings in normoxia exhibited significantly higher luciferase activity compared with controls in a dose- and time-dependent manner (Fig. 3b,c), consistent with ERFVII stabilization under aerobic conditions when cells were exposed to $H_2O_2$ (Fig. 2a,b). During reoxygenation, and in the absence of exogenous TBHP, luciferase activity did not decrease significantly from hypoxic levels, remaining higher than that in normoxic seedlings (Fig. 3a). These data confirm stabilization of ERFVIIs upon both oxidative stress and during reoxygenation, with the latter probably due to increased ROS production during extended hypoxia. Pretreatment with ROS scavenger ascorbate (10 mM) partially suppressed TBHP-induced RAP2.3 accumulation (Extended Data Fig. 3a) and delayed RAP2.3 accumulation on post-hypoxia reoxygenation (Extended Data Fig. 3b). RAP2.3–nLuc activity was also increased by antimycin A and diuron (Extended Data Fig. 3c,d) but not by other

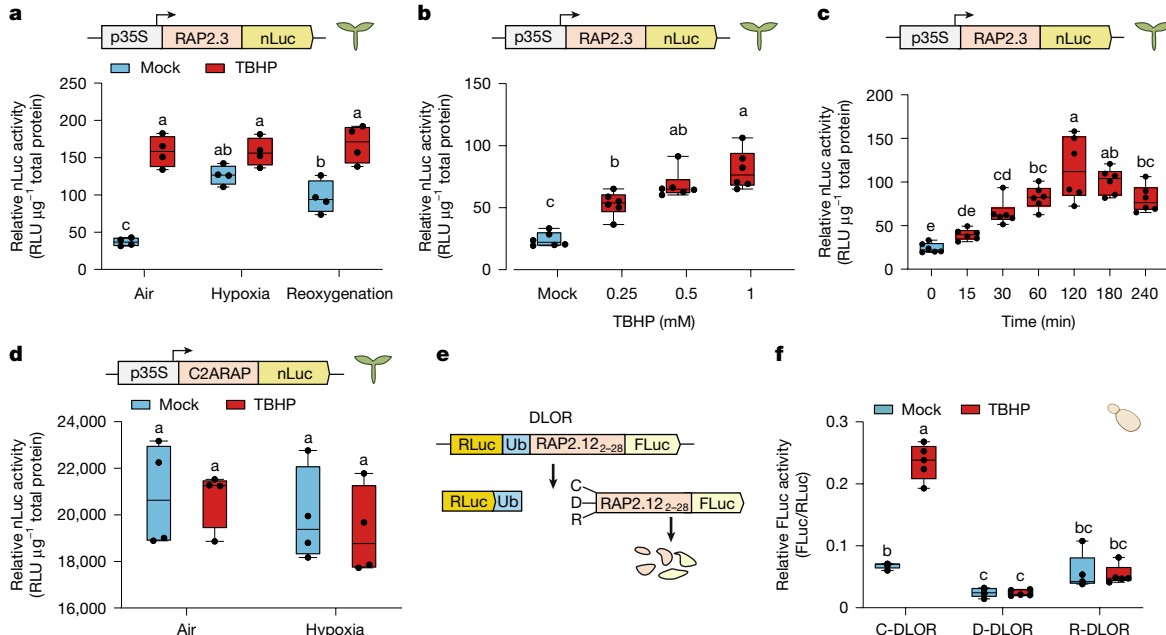

**Fig. 3 | Oxidative stress increases ERFVII stability in a N-degron pathway-dependent manner. a**, Relative nLuc activity of *35S:RAP2.3–nLuc* seedlings treated under hypoxic (1% $O_2$) or aerobic (21% $O_2$) conditions for 6 h, followed by 3 h of reoxygenation, 1 mM TBHP or mock treatment ($n = 4$). **b**, Relative nLuc activity of *35S:RAP2.3–nLuc* seedlings treated with different TBHP concentrations under aerobic conditions over 4 h ($n = 6$). **c**, Relative nLuc activity of *35S:RAP2.3–nLuc* seedlings treated with 1 mM TBHP under aerobic conditions measured over 4 h ($n = 6$). **d**, Relative nLuc activity of the *35S:(C2A)RAP2.3–nLuc* variant following 1 mM TBHP or mock treatment, under hypoxic (1% $O_2$) or aerobic (21% $O_2$) conditions ($n = 4$). **e**, Schematic representation of DLOR. **f**, Relative FLuc activity in *S. cerevisiae* cultures expressing *At*PCO4 together with either C-DLOR, R-DLOR or D-DLOR, exposed to 0.5 mM TBHP or mock treatment for 30 min ($n = 5$). Statistical differences were evaluated using one-way (**b** and **c**) or two-way (**a**, **d** and **f**) ANOVA ($P < 0.05$) followed by Tukey's HSD test ($P < 0.05$). In **a–d** and **f**, box plots indicate the median (middle line) and 25th and 75th percentiles (box limits); whiskers denote 1.5× the interquartile range, and outliers are shown as individual points.

ROS inducers including arsenite, cadmium, high light or methyl viologen[21–24] (Extended Data Fig. 3e–g), suggesting that ERFVII stabilization is induced selectively by specific ROS signals.

## ROS obstruct the Cys/Arg N-degron pathway

We next tested whether ROS signals interfered with the Cys-branch of the N-degron pathway that controls ERFVII stability. A reporter line expressing a fragment (amino acids 1–28) of RAP2.12 fused with the firefly luciferase enzyme (RAP2.12$_{1–28}$–FLuc)[3] exhibited behaviour similar to that of the RAP2.3–nLuc reporter (Extended Data Fig. 3h), with elevated FLuc activity in TBHP-treated aerobic and reoxygenated seedlings. This indicated that ROS-mediated ERFVII stabilization exclusively required the amino-terminal region of ERFVIIs, known to contain the conserved N-degron MCGGAII[2]. Similarly, replacing the N-terminal cysteine (Nt-Cys) of RAP2.3–nLuc with alanine (C2ARAP2.3–nLuc) abolished the increase in luciferase signal caused by TBHP that had been observed in Cys2-RAP2.3–nLuc (Fig. 3a,d). Similar results were obtained when we analysed Cys2-RAP2.3–nLuc in *ate1/2* mutant and *prt6* mutant backgrounds (Extended Data Fig. 3i). This indicated that ROS-induced prevention of ERFVII degradation was specifically due to the inability of the N-degron pathway to process protein N termini.

We next used a heterologous double luciferase oxygen reporter (DLOR) assay developed in budding yeast *Saccharomyces cerevisiae*[25] (Fig. 3e), in which the Nt-Cys-RAP2.12$_{2–28}$-coupled reporter output (FLuc/RLuc) is dependent on transgenic PCO enzymes (in this case, *At*PCO4). TBHP treatment of yeast caused a fast and dose-dependent increase in DLOR output, demonstrating specific TBHP-induced ERFVII stabilization, similar to our observations in plants (Fig. 3f and Extended Data Fig. 3j–l). The Arg/N-degron pathway in yeast is mediated by ATE1-dependent arginylation of Nt-Glu-, Nt-Asp- or Nt-Cys-SOO(O)

H-initiating proteins and UBR1-catalysed ubiquitinylation of Nt-Arg- proteins[26,27]. We therefore also tested two alternative versions of DLOR, in which the Nt-Cys (C) of the RAP2.12$_{2–28}$ domain was mutated to Asp (D) or Arg (R), respectively[28]. The luciferase signal was increased by TBHP treatment only in C-DLOR yeast; it did not change significantly in the D-DLOR or R-DLOR strains (Fig. 3f). These results indicate that TBHP does not interfere with ATE1 and UBR1 function; thus, they suggest that ROS-induced ERFVII stabilization, at least in yeast, occurs through inhibition of the PCO-catalysed step.

## $H_2O_2$ inhibits recombinant PCO enzymes

We next examined whether TBHP or $H_2O_2$ treatment could hinder the activity of PCO enzymes. As direct quantification of PCO catalytic activity in planta remains technically challenging, we treated 10 µM recombinant *At*PCO4 with 50 µM TBHP or $H_2O_2$ before removing the TBHP or $H_2O_2$ and testing PCO activity towards a peptidic ERFVII substrate (RAP2$_{2–15}$) using previously described assays[28]. Both treatments led to a significant reduction in *At*PCO4-catalysed RAP2$_{2–15}$ oxidation compared with untreated enzyme (Fig. 4a). Although some Nt-Cys oxidation was detected following direct treatment of RAP2$_{2–15}$ only with 1 mM $H_2O_2$ for 1 h (Extended Data Fig. 4a–d), all five *At*PCOs were sensitive to $H_2O_2$ (Fig. 4b), and, in a dose-dependence assay, $H_2O_2$ demonstrated a half-maximal inhibitory concentration of 8.36 (±1.09) µM towards 2 µM *At*PCO4 (Fig. 4c). Hypoxia-inducible[3] *At*PCO1 and *At*PCO2 were similarly sensitive to $H_2O_2$ (12.51 ± 1.39 µM and 4.18 ± 1.38 µM, respectively) (Extended Data Fig. 4e,f). These results support the primary effects of $H_2O_2$ and TBHP on N-degron-pathway-mediated ERFVII stability being through reduced PCO activity.

To explore the mechanism of direct PCO inhibition, we first tested whether $H_2O_2$ caused oxidative modification of the *At*PCO4 enzyme.

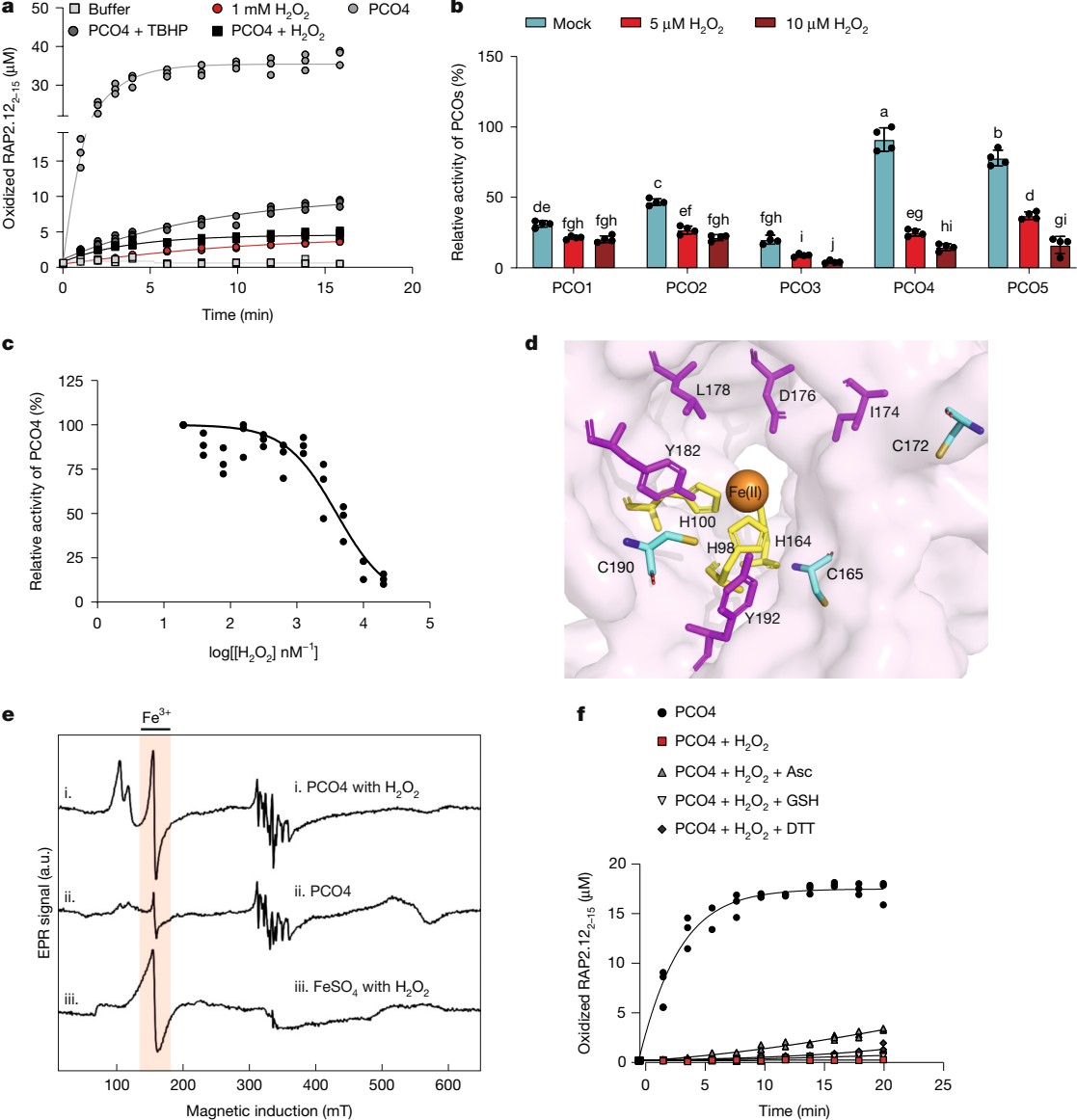

**Fig. 4 | $H_2O_2$ inhibits recombinant *At*PCO enzymes. a**, Oxidation of RAP2$_{2-15}$ by 50 μM $H_2O_2$- and 50 μM TBHP-treated or non-treated recombinant *At*PCO4 enzyme. Non-enzymatic RAP2$_{2-15}$ oxidation by 1 mM $H_2O_2$ was included for comparison with oxidation of RAP2$_{2-15}$ by *At*PCO4 (*n* = 3). **b**, $H_2O_2$-mediated inhibition of (10 μM) *At*PCOs 1–5; statistical differences were evaluated using two-way ANOVA followed by Tukey's HSD test (*P* < 0.05). Lines show the mean, with error bars representing the s.d. (*n* = 4). **c**, $H_2O_2$ dose-dependent effect on (2 μM) PCO4 enzyme activity (*n* = 3). **d**, Active site view of crystal structure of *At*PCO4 (PDB 6S7E); the Fe cofactor (orange sphere) is bound by a triad of His residues (His98, His100 and His164, yellow sticks), and Cys residues found to be oxidized by $H_2O_2$ are shown in cyan. **e**, X-band CW-EPR spectra of (i) *At*PCO4 with $H_2O_2$, (ii) *At*PCO4 only and (iii) FeSO$_{4(aqueous)}$ with $H_2O_2$. The spectrum in (i) shows Fe(III) signal intensity at 150 mT, similar to that seen in (iii) but with signal splitting at $g_{eff}$ = 4.29, $g'_{eff}$ = 6.4 and $g''_{eff}$ = 5.7, first as typical rhombic coordination of Fe(III) with two axially symmetric species. **f**, Activity restoration test of 10 μM *At*PCO4 enzyme treated with 100 μM $H_2O_2$, using known cellular reductants glutathione (GSH, 1 mM) and ascorbate (Asc, 1 mM), as well as DTT (1 mM) (*n* = 3).

Whole-protein mass spectrometry of $H_2O_2$-treated enzyme revealed a +209-Da mass increase (Extended Data Fig. 5a), which we considered to be likely to be due to non-specific oxidation of susceptible residues. Use of a sulfinic-acid-specific DiaAlk probe[29] confirmed that Cys residues of *At*PCO4 underwent this modification upon $H_2O_2$ treatment (Extended Data Fig. 5b). Further interrogation using proteomic analysis based on liquid chromatography coupled with tandem mass spectrometry (LC–MS/MS) of $H_2O_2$-treated and non-treated *At*PCO4 showed that Cys190, Cys172 and Cys165 of *At*PCO4 formed sulfinic (+31.98 Da) and sulfonic (+47.97) acids; all of these residues were close to the *At*PCO4 active site (Fig. 4d and Extended Data Fig. 5c–f). Oxidation of any of these Cys residues could therefore potentially hinder the catalytic ability of *At*PCO4. The PCOs are Fe(II)-dependent thiol dioxygenases[4]

(Fig. 4d); therefore, we also examined whether ROS-mediated oxidation of PCO active site Fe(II) occurred. Continuous-wave electron paramagnetic resonance (CW-EPR) spectroscopy revealed that $H_2O_2$ treatment resulted in an increase in rhombic Fe(III) signal at the *At*PCO4 active site, similar to that seen upon treatment of isolated Fe(II)$_{aqueous}$ with $H_2O_2$ (Fig. 4e). This indicated that Fe(II) oxidation could also contribute to PCO inhibition. Finally, we tested whether $H_2O_2$-mediated inhibition of *At*PCO4 could be restored using known cellular reductants glutathione and ascorbate[30,31], as well as a synthetic thiol reductant 1,4-dithiothreitol (DTT)[32]. None of the reductants was able to recover the activity of $H_2O_2$-treated enzyme (Fig. 4f), consistent with the duration of ERFVII stabilization seen in vivo. Collectively, these results indicate non-specific impacts of $H_2O_2$ treatment on PCO function, probably

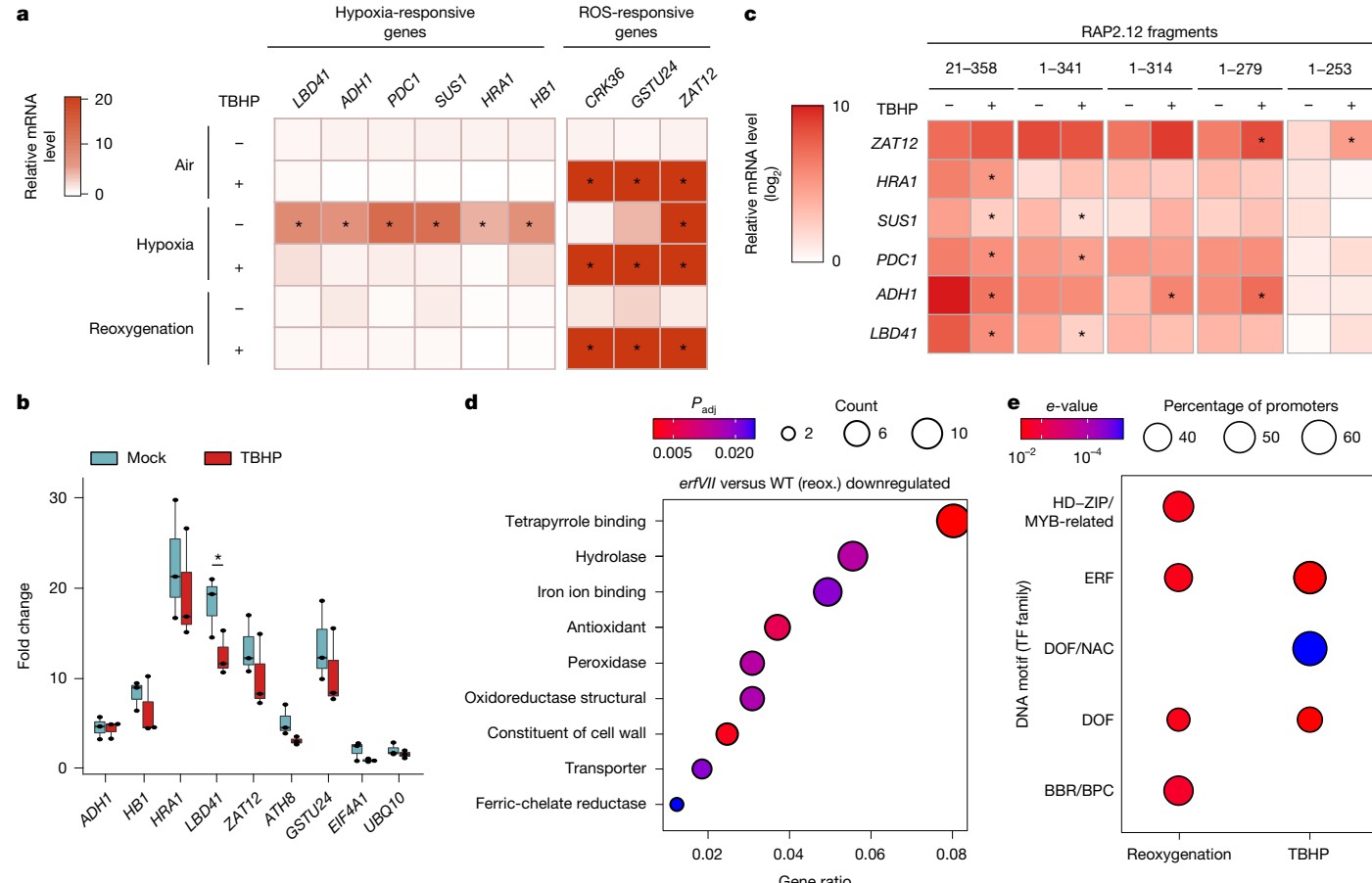

**Fig. 5 | Oxidative stress reverses ERFVII-dependent regulation of HRGs, and the PCO–ERFVII module regulates responses to hypoxia and oxidative stress. a**, Relative expression of hypoxia-responsive and ROS-responsive genes in air (21% O₂), hypoxia (1% O₂, 6 h) or 3 h of reoxygenation (21% O₂), in the presence or absence of 1 mM TBHP, in Col-0 seedlings (*n* = 4). Statistical analyses were conducted using two-way ANOVA followed by Tukey's HSD test (*P* < 0.05). **b**, ChIP–qPCR analysis of Δ13RAP2.12–GFP for hypoxia-responsive and oxidative stress gene promoters, compared with negative controls (*EIF4A1, UBQ1O*). Asterisks indicate statistically significant differences between the two mean values for each gene (two-sided Student's *t*-test, *P* < 0.05, *n* = 3). Box plots indicate the median (middle line) and 25th and 75th percentiles (box limits); whiskers denote 1.5× the interquartile range, and outliers are shown as individual points. Statistical differences among genes are reported in Supplementary Data 2. **c**, Relative expression of five hypoxia-responsive genes and *ZAT12* in truncated

versions of RAP2.12–nLuc seedlings treated with 1 mM TBHP under hypoxic conditions for 6 h (*n* = 4). Statistical analyses were conducted using two-sided Student's *t*-test (*P* < 0.05). **d**, Gene ontology enrichment results for molecular functions of downregulated genes in *erfVII* seedlings compared with wild type, upon reoxygenation (reox.). Circle size indicates the gene count per gene ontology term, with colour maps indicating the false discovery rate value (adjusted *P* value; *P*ₐdⱼ). Statistical significance was determined using a one-sided hypergeometric test; *P* values were adjusted for multiple comparisons using the Benjamini–Hochberg method. **e**, Comparison of motif enrichment of promoters of downregulated genes in *erfVII* seedlings versus wild type under TBHP treatment and reoxygenation conditions. Motif similarity was assessed using Pearson correlation coefficients, with *e*-values representing the alignment *P* value multiplied by the number of motifs in the target database. TF, transcription factor.

combining both catalytic inactivation and effects on protein structure, which together contribute to the ERFVII stabilization observed in vivo.

## ROS disrupt the hypoxic gene response

The H₂O₂-mediated stabilization of ERFVIIs observed here conflicted with previous reports of rapid repression of hypoxia-inducible genes upon reoxygenation[19,33], as well as lack of activation of the same set of genes under oxidative stress conditions[34]. We therefore investigated whether ROS signalling affected the hypoxia response in *Arabidopsis*. We exposed 7-day-old seedlings to 1 mM TBHP or mock control under normoxia (21% O₂), hypoxia (1% O₂) and reoxygenation (21% O₂) and monitored the expression of ERFVII-regulated genes known to respond to either oxidative stress or hypoxia. Real-time quantitative PCR (qPCR) analysis confirmed upregulation of all six HRGs under low oxygen conditions and downregulation upon reoxygenation (Fig. 5a). TBHP fully counteracted the induction of these genes in hypoxia (Fig. 5a). These results indicate that ROS could interfere with the hypoxia responses

of the plant. Consistent with these observations, expression of the core HRGs in the *RAP2.3–nLUC* transgenic *Arabidopsis* lines was also elevated in hypoxia, but significantly decreased upon TBHP treatment under hypoxia and/or under reoxygenation (Extended Data Fig. 6a). Induction of oxidative-stress-responsive genes was upregulated upon TBHP treatment (Fig. 5a and Extended Data Fig. 6a), confirming that transcriptional machinery was not affected and that the plants did experience oxidative stress as a result of TBHP treatment. Instead, TBHP seemed to selectively prevent HRG expression. Supplementation with 10 mM ascorbate significantly reduced HRG inhibition by TBHP in three of the six genes considered (*ADH1, HB1* and *PDC1*; Extended Data Fig. 6b). We therefore concluded that the cellular redox state influences the intensity of the molecular response mounted under hypoxia.

HRGs are controlled by ERFVIIs through their interaction with HRPEs[7] (Extended Data Fig. 2a). To understand how TBHP treatment decreases *HRG* expression, we investigated whether oxidative stress decreased the binding of ERFVIIs to HRPEs. We used a transgenic *Arabidopsis* line with a synthetic promoter harbouring a five-time repeat of HRPE driving

the expression of a *nLuc* gene and measured the expression levels of *nLuc* and the three core hypoxic genes in air, hypoxia and reoxygenation with or without TBHP-mediated oxidative stress (Extended Data Fig. 6c). Expression of *nLuc* was upregulated under hypoxic conditions but downregulated upon oxidative stress, similar to the expression of HRGs (Extended Data Fig. 6c). These results indicate that TBHP treatment interferes with transactivation of genes whose promoters include at least one HRPE, and not through a different DNA motif.

We next investigated whether oxidative stress hindered the ability of ERFVIIs to physically interact with the HRPE. We performed a chromatin immunoprecipitation (ChIP) experiment using a transgenic line expressing a GFP-tagged truncated version of RAP2.12 lacking 13 amino acid residues at the N terminus (Δ13RAP2.12–GFP[2,8]) and measured enrichment in the promoter regions of four hypoxia-responsive genes (*ADH1*, *HB1*, *HRA1* and *LBD41*) and three ERFVII-dependent oxidative-stress-responsive genes (*ZAT12*, *ATH8* and *GSTU24*) (Fig. 5b). The *UBQ10* and *EI4A1* promoters, which are not controlled by ERFVIIs, were used as negative controls. ChIP–qPCR revealed enrichment for all genes under normoxic conditions compared with the negative controls, confirming that Δ13RAP2.12–GFP binds to the promoter regions of these genes. A significant reduction in genomic enrichment was observed only for the *LBD41* promoter upon TBHP treatment; there was no statistically significant difference for any of the other promoters. These results indicate that oxidative stress may only mildly affect binding of RAP2.12 to selected genomic regions.

We speculated instead that the transactivation capacity of ERFVIIs might be altered under oxidative stress to suppress expression of HRGs. We excluded involvement of HRA1, a RAP2.12 interactor that has previously been shown to attenuate the hypoxic response in developing leaves[35], on the basis of real-time qPCR analysis that did not show any differences in HRG expression between the wild type and a *hra1-1* mutant (Extended Data Fig. 7a). We therefore decided to identify the region of the RAP2.12 protein required for HRG repression under oxidative stress. To this end, we generated DNA constructs to express five different truncated versions of the RAP2.12–nLuc fusion that lacked conserved ERFVII motifs (CMVIIs)[36] (Extended Data Fig. 7b) and used these to stably transform an *erfVII* knockout mutant. All these RAP2.12 versions were expected to avoid N-degron pathway degradation owing to removal of the entire N-Cys degron (RAP2.12$_{21–358}$) or substitution of the Cys2 with Ala (Extended Data Fig. 7b). Indeed, an nLuc activity assay showed that 1 mM TBHP treatment before hypoxia (1% O$_2$, 6 h) did not enhance the stability of the chimeric RAP2.12 protein (Extended Data Fig. 7c). In the same samples, we confirmed TBHP-induced HRG repression in plants expressing RAP2.12$_{21–358}$–nLuc and observed variable behaviour across the other four truncation variants, depending on the HRG considered (Fig. 5c). Removal of the last 18 amino acids, corresponding to a well-characterized transcription activation domain (CMVII-5)[37], in the MA-RAP2.12$_{1–341}$ version abolished repression of *HRA1* and *ADH1* (Fig. 5c). The MA-RAP2.12$_{1–253}$ variant, lacking CMVII-4, CMVII-5, CMVII-7 and CMVII-8, showed the lowest HRG expression, with no further repression by TBHP (Fig. 5c). Two variants (MA-RAP2.12$_{1–279}$ and MA-RAP2.12$_{1–314}$) consistently showed no significant HRG reduction between TBHP and mock treatments (Fig. 5c and Extended Data Fig. 7d). The same transgenic lines also did not show repression of HRGs following reoxygenation (Extended Data Fig. 7e). On the basis of the CMVIIs removed in these truncated RAP2.12 variants, we concluded that CMVII-8 is required for ROS-dependent repression of HRGs under conditions that entail elevated ROS levels.

## ERFVIIs redirect post-hypoxia responses

Having shown that ERFVII stability following reoxygenation represses the canonical hypoxia response, we investigated more broadly the contribution of ERFVII transcription factors to transcriptome reconfiguration under this condition. To this end, we sequenced the polyA-enriched total mRNAs of 7-day-old wild-type and *erfVII* seedlings treated with strict hypoxia (0.1% O$_2$) for 24 h, including an aerobic (21% O$_2$) control treatment, followed by reoxygenation for 3 h. Multidimensional scaling analysis showed a moderate effect of ERFVII inactivation under control conditions and a substantial difference between *erfVII* and the wild type under hypoxia, which was reduced by reoxygenation (Extended Data Fig. 8a). Approximately 25% of hypoxia-induced genes were repressed in *erfVII* compared with the wild type[38,39] (Supplementary Table 1). Upon reoxygenation, many genes in *erfVII* did not show the upregulation or downregulation observed in the wild type, revealing a substantial contribution of ERFVIIs to rearrangement of the transcriptome when seedlings were returned to normoxia following hypoxia (Extended Data Fig. 8b). As the ERFVIIs have been characterized by several independent studies as positive regulators of gene expression[37,38,40], we focused on the genes that showed significantly lower expression in the *erfVII* mutant compared with wild type following reoxygenation. Transcripts associated with antioxidant activity, cell wall remodelling and transport of molecules across membranes were enriched in this subset (Fig. 5d). We also carried out a separate RNA sequencing experiment to compare the *erfVII* and wild-type genotypes under TBHP-induced oxidative stress (1 mM, 6 h). TBHP elicited ERFVII-dependent transcriptional responses that were distinct from those induced by reoxygenation (Extended Data Fig. 8c–f). ERFVIIs were required for TBHP-mediated repression of genes involved in salicylate signalling, senescence, hypoxia and oxidative stress (Extended Data Fig. 8e). Conversely, absence of ERFVIIs during oxidative stress prevented expression of genes associated with plastids, chlorophyll biosynthesis and photosynthesis (Extended Data Fig. 8f).

We next compared the set of genes significantly less expressed in *erfVII* compared with the wild type under reoxygenation and oxidative stress conditions. The two gene lists did not largely overlap, probably owing to the substantial transcriptome rearrangement caused by hypoxia (Supplementary Tables 1 and 2). Nevertheless, inspection of the promoters of these genes revealed significant enrichment of DNA motifs recognized by ERFs and DNA binding with one finger transcription factors (Fig. 5e and Supplementary Tables 3–6), leading us to speculate that ERFVII may form partnerships with transcription factors from these protein families to mediate induction of these genes.

Overall, these results expand our understanding of the molecular roles of ERFVIIs as both activators and repressors of transcription (in a condition-dependent and gene-selective manner) in response to hypoxia, reoxygenation and oxidative stress.

## Discussion

Plant flood survival requires tolerance to both submergence-induced hypoxia and the reoxygenation stress associated with desubmergence. The molecular response to submergence-induced hypoxia has been well characterized and comprises PCO and O$_2$-dependent regulation of ERFVII stability and consequent upregulation of genes enabling adaptive metabolic reconfiguration[1–3,7]. However, the impact on this oxygen-sensing machinery of the ROS burst known to be associated with reoxygenation has been less well characterized[41–47]. We therefore investigated whether there was any cross-talk between reoxygenation, ROS and PCO–ERFVII function. We first confirmed that ERFVIIs have a role in facilitating both tolerance to and recovery from hypoxia–reoxygenation stress (Fig. 1). Using a combination of in vitro biochemical assays and in vivo reporter assays, we then confirmed that ERFVIIs persist in the nucleus after reoxygenation (Figs. 2 and 3) and collected evidence that this is caused by H$_2$O$_2$-mediated inactivation of the plant Nt-Cys-degron pathway at the PCO level, leading to ERFVII stabilization (Fig. 4). Although we could not rule out a contribution of ERFVII dimerization through Nt-Cys disulfide bond formation on the basis of our analysis of peptides, our biophysical analysis of oxidized AtPCO4 revealed non-specific mechanisms of direct inhibition. In human

cells during prolonged hypoxia, Cys-initiating N-degron substrates regulated by PCO homologue 2-aminethanethiol dioxygenase can be oxidized by ROS at the Nt-Cys to form Nt-Cys-sulfonic acid, redirecting them to degradation through lysosomal autophagy[48]. By contrast, our mass spectrometry analysis did not show substantial sulfonic acid formation at the Nt-Cys of RAP2.12 following TBHP treatment (Extended Data Fig. 4a–d), suggesting that the primary mechanism by which ROS affect ERFVII stability is PCO inhibition. Future work in *Arabidopsis* is expected to shed light on the potential contribution of this further proteolytic pathway to degradation of ERFVIIs after 3 h of reoxygenation or under chronic oxidative stress.

Notably, we found that ROS-stabilized ERFVIIs did not promote but rather repressed the expression of HRGs (Fig. 5a) and that ERFVIIs modulated transcription of a different set of genes in response to ROS than they did in response to hypoxia (Fig. 5a and Extended Data Fig. 8a–d), suggesting that the PCO–ERFVII signalling pathway can be redirected in a stress-specific manner. The ROS-dependence of these effects was supported with antioxidant treatments, and although these can have pleiotropic effects, the data consistently support a role for ERFVIIs in post-hypoxic stress (Extended Data Fig. 6b). ERFVII transcription factors have previously been proposed to mediate responses to various stresses, including high salinity, heat and temperature[13,14,49]. These stresses have been proposed to affect the degradation of ERFVIIs through the N-degron pathway; for example, salt stress has been reported to decrease NO production[14,50], and it has been speculated that it may impair ATE or PRT6 function[25]. Our data, showing that exogenous supply or endogenous production of $H_2O_2$ results in PCO inhibition and ERFVII stabilization, suggest that other conditions that involve enhanced $H_2O_2$ accumulation, such as cold stress[51] or pathogen attack[52,53], may also invoke ERFVII stabilization by means of this method.

PCOs can respond sensitively to $O_2$ availability through a kinetic effect on ERFVII oxidation. However, the response of PCOs to the presence of $H_2O_2$ is less finely tuned; inhibition of PCO function by $H_2O_2$ seemed to occur in a less specific manner, probably through a combination of both oxidation of the active site Fe(II) to Fe(III), as detected by EPR, and oxidation of Cys residues in or near the active site, as detected by probes detecting Cys-sulfinic acids and tryptic digest mass spectrometry. It is also possible that other oxidative modifications took place that we were unable to detect. These modifications are all capable of affecting substrate binding and catalysis to decrease or ablate the activity of the enzyme. There is a precedent for ROS-mediated inhibition of oxygen-sensing enzymes in mammalian systems, in which $H_2O_2$-mediated inhibition of factor inhibiting HIF, an Fe(II) and 2-oxoglutarate dependent oxygenase, has been reported[54]. We could not observe repair of $H_2O_2$-mediated PCO inhibition with cellular antioxidants in vitro; however, it would be of interest to examine whether modulation of redox status in cells could affect ERFVII stabilization, or indeed whether enzymes that can reverse Cys oxidation may have a role in protection of PCOs from ROS-mediated damage.

Our findings that despite ERFVII stabilization and retention at HRPEs, HRGs are 'turned off' upon ROS treatment, whereas responses to oxidative stress are 'turned on', provide an explanation for the role of ERFVIIs as an important hub for control of adaptation to different adverse environmental conditions[14] (Extended Data Fig. 8g). The versatility of ERFVIIs in producing stress-tailored responses can be explained by their interactions with distinct proteins and DNA motifs. Our results indicate that oxidative stress causes only mild dislocations of ERFVIIs from hypoxia-responsive genes and is more likely to turn ERFVIIs from positive to negative regulators of gene transcription. This could explain the limited overlap of the transcriptional changes observed between hypoxia and other stresses that involve ERFVII stabilization, despite the crucial role of these transcription factors in enabling tolerance. A systematic survey of the transcriptional activity of RAP2.12 truncations showed that CMVII-8, an intrinsically disordered region[55], was required to reverse regulation of HRGs under oxidative stress (Fig. 5c

and Extended Data Fig. 7c–e). Notably, this motif showed strong transactivation capacity in yeast monohybrid assays and was categorized as a subtype 4 activation domain, characterized by negatively charged residues and aromatic amino acids. Although the exact mechanism remains to be defined, it is possible that CMVII-8 contacts the mediator complex following the acidic exposure model. The neighbouring CMVII-4 and CMVII-7 have been shown to support the AP2 domain in binding the MED25 subunit[40,56] The nuclear redox milieu could determine the composition of the mediator complex that interacts with ERFVII, as shown in animal and plant cells[57,58], ultimately imparting activation or repressive capacity. Future studies could shed light on this aspect.

The ability of ERFVII to regulate different subsets of genes in response to different stimuli raises questions about the evolution of these transcription factors. Was the last common ancestor of the ERFVIIs, which probably arose at the origin of land plants[38], a generic regulator of stress tolerance required to cope with an environment of increasing ROS? Or, rather, did the ERFVIIs evolve first as signal transducers for hypoxia, with their role only later expanding to accommodate the response to other stresses? In today's environment, however, it seems that the PCO–ERFVII sensing and signalling nexus is more nuanced than a simple response to fluctuations in $O_2$ availability. Its sensitivity to oxidative stress, including that which occurs upon reoxygenation, allows dynamic control in response to a range of cues. The overall impact of this ability to switch between subsets of genes explains the key roles of ERFVIIs in plant survival of both submergence and desubmergence.

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

# Methods

## Experimental model and subjects

**Plant materials and growth conditions.** *Arabidopsis thaliana* Col-0 was used as the wild-type ecotype. The genotypes RAP2.12[1–28]–FLuc, RAP2.12–GFP and Δ13RAP2.12–GFP were as previously described[2,3]. Seeds were sown in a 3:1 soil/vermiculite mixture, stratified at 4 °C in the dark for 3 days, then germinated at 22 °C/20 °C with a 16 h:8 h light/dark photoperiod and 100 μmol photons $m^{-2}$ $s^{-1}$ intensity. For in vitro propagation, seeds were sterilized using 70% ethanol for 1 min and incubated in 10% sodium hypochlorite (NaClO) for 10 min, followed by 6 washes in 1 ml sterile distilled water. For growth in liquid medium, 100 μl of seed suspension, corresponding to 20–40 seeds, was inoculated in 1 ml of sterile half-strength MS medium (basal salt mixture 2.15 g $l^{-1}$, pH 5.7) supplemented with 1% sucrose in each well of 6-well plates. For growth in solid media, seeds were incubated in the dark at 4 °C for 2 days and subsequently on half-strength MS medium[59], supplemented with 1% (w/v) sucrose and 0.8% (w/v) agar, and grown at 22 °C with a 16:8 day/night photoperiod at 100 μmol photons $m^{-2}$ $s^{-1}$ intensity.

**Yeast strains and culture.** A haploid parental strain BY4742 (Matα; His3-Δ1; Leu2-Δ0; Lys2-Δ0; Ura3-Δ0; Euroscarf #Y10000) was cotransformed following the LiAc/SS carrier DNA/PEG method[60] with PCO4–pAG415GPD and different versions of the DLOR–pAG413GPD (C-, D- or R-DLOR). All plasmids used for yeast expression were produced in previous work[61]. Before transformation, cells were grown at 30 °C on YPDA (20 g $l^{-1}$ peptone, 10 g $l^{-1}$ yeast extract, 20 g $l^{-1}$ glucose (Duchefa) and 20 mg $l^{1}$ adenine hemisulfate (Sigma-Aldrich), supplied with 20 g $l^{-1}$ agar (Duchefa) when necessary). Transformants were selected on SD medium containing 6.7 g $l^{-1}$ yeast nitrogen base (DIFCO), 1.37 g $l^{-1}$ yeast dropout medium (Sigma-Aldrich) and 20 g $l^{-1}$ glucose, plus supplements (0.16 M uracil, 0.8 M histidine–HCl, 0.8 M leucine and 0.32 M tryptophan (Sigma-Aldrich) when complete), with 20 g $l^{-1}$ agar when solid.

**Bacterial strains.** Bacterial strain *Escherichia coli* BL21 (DE3) was used for expression of recombinant PCOs. Bacteria were cultured in 2YT medium at 37 °C until the optical density at 600 nm ($OD_{600}$) reached 0.6. Protein expression was induced by addition of 0.8 mM isopropyl-β-D-thiogalactoside (Sigma-Aldrich) at 18 °C for 16 h with shaking at 170 rpm in an incubator (Eppendorf).

## Method details

**DNA construct generation.** For generation of the 35S:RAP2.3–nLuc construct, an 806-nucleotide synthetic string containing an *Arabidopsis* codon-optimized nLuc sequence including the RAP2.2 intron (Supplementary Table 7) was synthesized in the pMK-RQ backbone by GeneArt (Thermo Fisher Scientific). The destination vector pK7GWnL2 was generated through ligation between pK7GW2 (ref. 62) and GWnLuc-intron after restriction using XbaI and MluI (Thermo Fisher Scientific). The *Arabidopsis* RAP2.3 CDS was amplified from Col-0 complementary DNA without stop codon (RAP2.3Δstop), with overlapping *AttB* sites introduced by PCR using GoTaq DNA polymerase (Promega). Entry clone vector was then generated as a BP reaction between the RAP2.3 CDS PCR product and pENTR/D-TOPO (Life Technologies). The resulting entry vector was recombined into the generated pK7GWnL2 destination vector using LR clonase mix II (Thermo Fisher Scientific). Primers used for RAP2.3Δstop cloning and screening are listed in Supplementary Table 8. For generation of the *35S:RAP2.3–GFP* construct, the entry vector containing RAP2.3 CDS was recombined with a pK7GW2F[62] destination vector using LR clonase mix II (Thermo Fisher Scientific).

For HRPE–nLuc construct design, DNA containing the SacI-attR1-ccdB-attR2-NanoLuc-HindIII sequence (Supplementary Table 7) was de novo synthesized by the GeneArt service (Thermo Fisher Scientific) and cloned into the pBGWL7 Gateway destination vector[63] The entry

vector containing the HRPE:5′-UTR 35S sequence[64] was recombined into the destination vector by Gateway cloning.

For Flag-tagged RAP2.12, the Golden Braid cloning system[65] was used. The RAP2.12 CDS was resynthesized to substitute the 129–195 nucleotide (non-conserved) region with a plant-optimized sequence coding for a 3×Flag tag, and a 3×HA tag was added to the end of the CDS before the stop codon (Supplementary Table 7). This DNA string was cloned in the pUPD2 plasmid (elements B3–B4) using BsmBI and T4 ligase. The promoter of RAP2.12 (2,196 nucleotides upstream of the first ATG codon) was synthesized with the BsaI site (483 nucleotides) removed and cloned into pUPD2 to generate an A1–B2 element level 0 plasmid. An Alpha2-level vector was generated by assembling the RAP2.12 promoter (A1–B2), the double-tagged (Flag/HA) RAP2.12 CDS (B3–4), an extra eGFP tag (B5) and the NOS terminator (B6–C1), using BsaI and T4 DNA ligase. The resulting expression cassette was assembled with the *promOLE1:gOLE1–TagRFP:Tnos* cassette (Alpha1) into a binary Omega1 plasmid using BsmBI and T4 DNA ligase.

The DNA constructs to overexpress fragments of RAP2.12 were also generated the Golden Braid cloning system. Level 0 was created using BsmBI and T4 DNA ligase with PCR products corresponding to CDS portions coding for the desired RAP2.12 fragments. Alpha2 plasmids were generated to include the expression cassette *prom2x35S:RAP2.12 fragment–nLuc:Tnos* using BsaI and T4 DNA ligase. The resulting expression cassette was assembled with the *promOLE1:gOLE1–TagRFP:Tnos* cassette (Alpha1) into a binary Omega1 plasmid using BsmBI and T4 DNA ligase.

For recombinant protein production, the five *pco* genes from *Arabidopsis* had previously been cloned into the NdeI and XhoI sites of pET28a (Novagen) and transformed into *E. coli* NEB5α competent cells (New England Biolabs), and the sequences were validated by Sanger sequencing (Source Biosciences), as previously described[66].

**Arabidopsis transformation.** Agrobacterium-mediated transformation was performed to obtain RAP2.3–nLUC, RAP2.3–GFP and HRPE:nLuc stable transgenic lines using floral dip medium as previously described[67]. $T_0$ seeds were selected for resistance on agarized half-strength MS medium supplemented with the corresponding antibiotic and subsequently transferred in soil. The presence of the transgene was detected by PCR using GoTaq DNA polymerase (Promega). $T_3$ generation plants were used for the experiments.

**Low oxygen and reoxygenation treatments.** For hypoxia treatments, seedlings were grown in six-well plates in liquid media and subjected to anaerobic conditions inside Hypoxic Workstations (Whitley) continuously flushed with an artificial humidified atmosphere containing a mixture of oxygen (1%) and nitrogen gases (99%) at 22 °C for 6 h. For severe hypoxia treatment, seedlings were grown vertically in square plates and treated with 0.1% $O_2$ v/v $O_2/N_2$ for 24 h. During the hypoxic treatments, the seedlings were maintained in the dark to avoid oxygen release by photosynthesis. Seedlings used for control samples were maintained under aerobic conditions (21% $O_2$ v/v $O_2/N_2$) in the dark for equal times. After low oxygen treatment, plants were transferred to aerobic growth conditions for reoxygenation treatment.

**Root length and survival rate measurements.** To assess reoxygenation tolerance, 7-day-old seedlings were treated as previously described. Four or five plates, each containing five to seven seedlings, were used to test for each condition. Primary root length was measured both before and after 4 days of recovery, and fresh weight and survival rate were assessed following 4 days of recovery. Transparent squared plates containing *Arabidopsis* seedlings were scanned using an EPSON Perfection V750 PRO scanner with a resolution of 720 dots per inch. Growth rate was measured as increase in length of the primary root divided by days of recovery. Primary root length and lateral root density were assessed using ImageJ[68] (v.1.54j).

**TBHP treatment.** Oxidative stress in plants was induced by treatment with 1 mM TBHP diluted in Milli-Q water for 6 h in normoxia and 6 h in hypoxia. After 3 h, the medium was replenished with TBHP.

**Histochemical H$_2$O$_2$ staining and quantification.** ROS were visualized with DAB (Fluorochem) staining to detect H$_2$O$_2$ using methods described previously[69] with minor modifications. Seedlings were incubated with 1 mg ml$^{-1}$ DAB, vacuum infiltrated for 5 min and incubated for 4–5 h in the dark with shaking. After staining, seedlings were washed with distilled water and bleached in several washes of 70% ethanol. Five to eight seedlings were analysed per condition using a Leica M165C stereo microscope with ×2.5 magnification, followed by quantification of pixel intensity using ImageJ[68] (v.1.54j).

**Evans blue staining for cell viability.** Approximately 25–30 *Arabidopsis* seedlings per treatment were collected at designated time points during hypoxia and subsequent reoxygenation for both Col-0 and *erfVII* genotypes. Seedlings were incubated in 0.25% (w/v) aqueous Evans blue solution prepared in 0.1 mM CaCl$_2$ (pH 5.6) for 15 min in the dark at room temperature. Following staining, seedlings were washed three times with Milli-Q water to remove excess dye. Root tissues were then visualized using a Leica M165C stereo microscope.

**Fluorescent biosensing of oxidative stress in *Arabidopsis*.** *Arabidopsis* wild-type plant lines stably expressing protein sensor roGFP2-Orp1 with cytosolic and nuclear localization were as described previously[16]. Sensor lines in *erfVII* and *prt6* background[70] were generated by Agrobacterium-mediated transformation using floral dip with a pH2GW7:cyt-roGFP2-Orp1 expression construct. Positive transformants were selected on selection medium with resistance marker hygromycin B on the basis of fluorescence. Measurements were performed using two independent sensor lines to control for any potential effects of sensor insertion loci. Replicates from the two lines were then combined for each genotype. Leaf discs of 5-week-old plants and 7-day-old seedlings were submerged in wells of a 96-well plate filled with 200 µl standard assay medium (10 mM MES (pH 5.8 with KOH), 10 mM MgCl$_2$, 10 mM CaCl$_2$, 5 mM KCl). A single leaf disc was placed in each well with the abaxial side up for the leaf disc experiment, whereas five or six seedlings were used per well in seedling experiments. Ratiometric readout of the biosensor was performed using a multiwell fluorimeter (ClarioStar Plus, BMG Labtech) in top optics mode, using 30 excitation flashes distributed in a 3-mm orbital average diameter (leaf discs) and a 3-mm spiral average diameter (seedlings). Samples of both genotypes were measured side by side using the same gain for the fluorophore channels (Ex1: F:400-10, Ex2: F:482-16; dichroic mirror F:LP504; Em: F:520-10) for all samples for maximum comparability. Wild-type plants without sensor expression were included for background correction. Emissions of each sample were collected every 189 s (leaf discs) or 243 s (seedlings).

Oxygen gradients were performed by targeted influx of N$_2$ into the reader system using an atmospheric control unit. For leaf disc experiments, different O$_2$ concentrations were tested on consecutive days using material from the same plants, with consistent measurement parameters. For seedling experiments, different batches of plants of the same age were used for each O$_2$ concentration. In vivo responsiveness of the roGFP2-Orp1 protein sensor was routinely validated after experiments using the same tissue by subsequent treatment with 20 mM DTT to drive the sensor to a fully reduced state, followed by two washes with standard assay medium before addition of 20 mM 2,2′-dithiodipyridine to oxidize the sensor. For data analysis, wild-type autofluorescence was subtracted from biosensor intensities before calculation of 400 nm/482 nm ratios for each time point. Ratio data were log$_{10}$-transformed to increase symmetry.

**Confocal imaging.** Seven-day-old seedlings were used for GFP detection after treatment. For nuclear localization, seedlings were stained in phosphate-buffered saline (PBS) containing 1 µg ml$^{-1}$ 4′,6-diamidino-2-phenylindole (DAPI, Thermo Fisher Scientific) and washed three times in PBS. Imaging was performed using a ZEISS LSM 880 Airyscan microscope (Department of Biology, University of Oxford), equipped with a ×25 objective lens, upon laser excitation at 405 nm and collection at 410–495 nm for DAPI imaging, and excitation at 488 nm and collection at 498–560 nm for GFP imaging. Confocal images were analysed using ZEISS ZEN Lite software (v.3.11).

**Western blot.** Equal amounts of total protein (100 µg) were resolved by 10% SDS–PAGE and transferred to a polyvinylidene difluoride membrane (Power Blotter Pre-cut Membranes) using Power Blotter 1-Step Transfer Buffer (Invitrogen). Membranes were probed with an anti-Flag M2-Peroxidase (HRP) antibody (Sigma-Aldrich, catalogue. no. A8592) at 1:5,000 dilution overnight at 4 °C. Following incubation, membranes were washed three times with PBST (1× PBS containing 0.1% Tween-20) for 5 min each at room temperature. Immunoblots were developed on film using SuperSignal West Atto Ultimate Sensitivity Substrate (Thermo Fisher Scientific) and imaged on the iBright CL1500 Imaging System (Thermo Fisher Scientific). To verify equal protein loading, the membranes were subsequently rinsed with distilled water and stained with 0.1% (w/v) Ponceau S solution in 5% acetic acid for 5 minutes at room temperature to visualize total protein bands. Excess stain was removed by washing the membrane with distilled water until clear background was obtained. The stained membrane was then imaged for documentation.

**Luciferase assay.** Total proteins were extracted in passive lysis buffer (Promega). Firefly Luciferase activities were measured using a ONE-Glo Luciferase Assay kit (Promega), and the Nano-Glo Luciferase Assay System (Promega) was used to measure the activity of the nLuc enzyme. The luciferase signal was normalized on the basis of the total protein concentration using the Bradford assay[71].

**ROS pretreatment.** For ROS pretreatment, 7-day-old seedlings grown vertically were incubated with 1 mM TBHP for 2 h in the dark. Following pretreatment, seedlings were used for tolerance assays as described above.

**Yeast treatments.** For TBHP treatments, colonies were inoculated in 5 ml of -His -Leu SD medium, grown overnight, diluted to half in fresh medium and further diluted to OD$_{600}$ = 0.1. Cultures were grown for 5 h before the treatment. TBHP was then supplied for up to 30 min at different concentrations (0, 0.25, 0.5 and 0.75 mM). Samples of culture (50 µl) were collected for luciferase assays, centrifuged at 15,000 rpm for 5 min, frozen and extracted in 50 ml of PLB. Luciferase was measured using the Dual-Luciferase Reporter Assay System (Promega) as described previously[25]. One millilitre of culture was used for OD$_{600}$ spectrometric measurements.

**ROS scavenger and inducer treatments.** Seven-day-old *Arabidopsis* seedlings were exposed to high light (1,600–1,800 µmol m$^{-2}$ s$^{-1}$) for 15 min to 1 h, or treated with ascorbate (10 mM), cadmium (10 mM), arsenic (10 mM), diuron (1 mM), antimycin A (200 µM) or methyl viologen (1 mM) for 4 h in the dark under air conditions.

**Recombinant protein production.** Expression and His6-tag affinity purification of *At*PCOs were performed as previously described[66]. Following affinity purification, the His6-tag was cleaved using TEV protease, and the cleaved tag was removed using a HisTrap HP column (GE Healthcare). Proteins were then purified with a HiLoad 26/600 Superdex 75 prep-grade size-exclusion column (GE Healthcare) equilibrated with 50 mM Tris (pH 7.5) and 0.4 M NaCl. Protein purity was assessed with SDS–PAGE.

**In vitro H$_2$O$_2$ oxidation assay of PCOs.** Recombinant PCO enzymes (*At*PCO1 to *At*PCO5; 10 µM) were incubated with H$_2$O$_2$ or an equal volume of H$_2$O in 50 mM HEPES buffer at pH 7.4 (herein termed reaction buffer) at 4 °C for 30 min. Excess H$_2$O$_2$ was removed using a Micro Bio-Spin P-6 chromatography column (Bio-Rad) equilibrated with reaction buffer. Then, 200 µM RAP2$_{2-15}$ peptide (CGGAIISDFIPPPR, purchased from GL Biochem, China) was reacted with 1 µM PCO (H$_2$O$_2$ treated or non-treated) at 25 °C for the required time. For determination of half-maximal inhibitory concentration, 2 µM *At*PCO1, *At*PCO2 or *At*PCO4 was incubated with a series of H$_2$O$_2$ concentrations (0–2 mM) at 4 °C for 30 min. Subsequently, 100 µM RAP2$_{2-15}$ peptide was incubated with 1 µM PCO or with buffer alone (H$_2$O$_2$-treated or untreated) as a control at 25 °C for the specified reaction time. After each reaction, 5-µl samples were quenched in 45 µl of 5% formic acid to stop the enzymatic reaction. Peptide masses were subsequently analysed using an Agilent RapidFire RF360 sampling robot connected to an Agilent 6530 Accurate-Mass Q-ToF mass spectrometer operated in positive electrospray mode. Product distributions were assigned on the basis of the relative integrated areas of peaks corresponding to products of interest. Spectra were visualized using Qualitative Analysis (v.B.07.00), and Agilent RapidFire Integrator (v.4.3.0.17235) was used to calculate integrated peak areas.

**H$_2$O$_2$ oxidation assay of RAP2$_{2-15}$ peptide.** Stock solutions of RAP2$_{2-15}$ (200 µM) were prepared in reaction buffer and treated with 400 µM freshly prepared DTT for 45 min at room temperature (25 °C) to ensure peptides were monomeric (in all experiments unless otherwise specified). Peptides were then treated with H$_2$O$_2$ under the conditions required by the experiment. Time-course experiments were conducted in 2-ml deep-welled plates and analysed in real time using RapidFire mass spectrometry as described above.

**Peptide fragmentation by LC–MS/MS.** DTT-reduced RAP2$_{2-15}$ (20 µM) was treated with 1 mM H$_2$O$_2$ at room temperature for 1 h. A 5-µl sample was diluted with 45 µl 5% formic acid for measurement. LC–MS/MS was carried out using an Acquity UPLC system coupled to a Xevo G2-XS Q-ToF mass spectrometer on a Chromolith Performance RP-18e 100-2 mm HPLC column (Merck) at 40 °C as above. Ions with an *m/z* ratio of 1,474.7 (+32 Da of AtRAP2$_{2-15}$) were selected for sequential fragmentation under a collision energy of 80 V. Spectra were visualized using Qualitative Analysis (v.B.07.00). Fragments were assigned by comparison of the obtained spectrum with computationally predicted fragment patterns, calculated using the web tool Protein Prospector (v.6.3.1; University of California, San Francisco).

**Protein analysis.** For protein mass measurement, 100 µM of recombinant *At*PCO4 enzyme was incubated with 1 mM H$_2$O$_2$ or an equal volume of H$_2$O in reaction buffer for 1 h at 25 °C. Excess H$_2$O$_2$ was removed using a Micro Bio-Spin P-6 chromatography column (Bio-Rad) equilibrated with reaction buffer. The total mass of the H$_2$O$_2$-treated or non-treated *At*PCO4 was measured using RapidFire mass spectrometry as described above.

**Cysteine oxidation detection.** H$_2$O$_2$-treated or non-treated *At*PCO4 was incubated with 1 mM BioDiaAlk in the dark for 1 h at 25 °C, followed by reduction with 10 mM DTT for 1 h at 25 °C. Equal amounts of each protein were separated by SDS–PAGE and transferred on to polyvinylidene fluoride membranes, followed by streptavidin–HRP blotting at 1:1,000 dilution or anti-his-HRP blotting at 1:10,000 dilution. The protein signal was visualized by chemiluminescence (ECL Plus, Pierce).

**LC–MS/MS data acquisition.** Recombinant *At*PCO4 enzyme (15 µg) was incubated with H$_2$O$_2$ or an equal volume of H$_2$O in reaction buffer for 1 h at 25 °C. After removal of excess H$_2$O$_2$, the enzyme was reduced with 85 mM DTT in 50 mM ammonium bicarbonate (Ambic) for 40 min

at 56 °C, followed by incubation with 55 mM iodoacetamide in 50 mM Ambic for 30 min in the dark at room temperature. For elimination of excess iodoacetamide, samples were reduced again with 85 mM DTT in 50 mM Ambic for 10 min in the dark at room temperature. In-solution trypsin digestion was performed by addition of trypsin in a 1:50 (w/w) ratio overnight at 37 °C, followed by desalting using C18 ZipTip. The resulting tryptic peptides were resuspended in 40 µl of Milli-Q water with 2% acetonitrile and 0.1% formic acid, and 2 µl was analysed on a nanoAcquity UPLC system (Waters) connected to an Orbitrap Elite mass spectrometer (Thermo Fischer Scientific) possessing an EASY-Spray nano-electrospray ion source (Thermo Fischer Scientific). The peptides were trapped on an in-house packed guard column (75 µm internal diameter × 20 mm, Acclaim PepMap C18, 3 µm, 100 Å) using solvent A (0.1% formic acid in water) at a pressure of 140 bar. The peptides were separated on an EASY-spray Acclaim PepMap analytical column (75 µm internal diameter × 50 mm, RSLC C18, 3 µm, 100 Å) using a linear gradient (length: 100 min, 3% to 60% solvent B (0.1% formic acid in acetonitrile), flow rate: 300 nl min$^{-1}$). The separated peptides were electrosprayed directly into the mass spectrometer, which was operated in data-dependent mode using a collision-induced dissociation (CID)-based method that performed beam-type CID fragmentation of the peptides. The instrument was controlled using Orbitrap Eclipse Tune 3.5/3.1 and Xcalibur 4.5/4.4. Full scan mass spectra (scan range: 350–1,500 *m/z;* resolution: 120,000; AGC target: 1e6; maximum injection time: 250 ms) and subsequent CID MS/MS spectra (AGC target: 5e4; maximum injection time: 100 ms) of the 10 most intense peaks were acquired in the ion trap. CID fragmentation was performed at 35% of normalized collision energy, and the signal intensity threshold was kept at 500 counts.

**Processing data.** Data were analysed with Peaks v.8.5. The raw MS file was searched against the TAIR database. Trypsin with a maximum of three missed cleavages and one unspecific end was selected as the protease. Carbamidomethylation (cysteine) was set as a fixed modification, and oxidation (methionine) and deamination (asparagine, glutamine) were set as variable modifications. The precursor mass tolerance was set to 15 ppm. Fragment mass tolerances for CID were set to 0.8 Da. All peptides present at −log$_{10}$[*P*] > 20 and spectra were manually checked and validated or disqualified.

**EPR of *At*PCO4.** EPR spectroscopy was performed on a Bruker EMXmicro spectrometer with a Premium bridge connected to an ER-4122SHQE-W1 cavity fitted to an Oxford Instruments ESR900 cryostat. The microwave source was operated at 9.3891(17) GHz, and spectra were recorded at 10 K with liquid helium cryogen. Protein (200 µM) and control solutions were frozen in liquid N$_2$. Spectra were obtained as two 5-min scans from 10 mT to 650 mT using a time constant of 20.48 ms, a microwave power of 200 µW, modulation amplitude of 1 mT and modulation frequency of 100 kHz.

**RNA extraction and real-time qPCR analyses.** Total RNA was extracted from 60–80 mg of plant material using the phenol–chloroform extraction method as described previously[3]. RNA concentration was quantified using a NanoDrop ND-1000 (Thermo Scientific), and RNA integrity was tested on a 1% agarose gel. One microgram of total RNA was subjected to DNase Treatment (Thermo Scientific) and retrotranscribed using a qPCRBIO cDNA Synthesis Kit (PCR Biosystems). Real-time qPCR was performed with a QuantStudio 5 Real-Time PCR System (Applied Biosystems) using Power SYBR Master Mix (Thermo Fisher Scientific). Ubiquitin-10 (*AT4G05320*) was used as a housekeeping gene for *Arabidopsis* analysis. Four biological replicates were extracted for each condition, each represented by two technical replicates, and the average expression was calculated. The primer pairs used for real-time RT–qPCR are listed in Supplementary Table 9. The relative expression of each individual gene was calculated using the 2$^{-C_t}$ method[72].

**ChIP assay.** ChIP was performed using a modified version of the protocol described in ref. 73. Chromatin was extracted from 2 g of 7-day-old Δ13RAP2.12–GFP seedlings grown in sterile liquid half-strength MS medium, supplemented with 1% w/v sucrose, under controlled conditions (16 h:8 h light/dark photoperiod, at 22 °C). Seedlings were treated with 1 mM TBHP, or dimethyl sulfoxide (DMSO) as a control, in 1 ml of fresh liquid 1% w/v sucrose half-strength MS medium for 6 h in the dark. Seedlings were cross-linked by dipping in 1% formaldehyde for 10 min and quenched with 0.125 M glycine under vacuum infiltration for 5 min. Seedlings were blotted on paper tissue to dry them and immediately frozen in liquid nitrogen. Each sample was ground to powder and resuspended in 2.5 ml nuclei extraction buffer (100 mM MOPS pH 7.6, 10 mM MgCl$_2$, 0.25 M sucrose, 5% dextran T-40, 2.5% Ficoll 400, 40 mM β-mercaptoethanol, 1× protease inhibitor cocktail (P8340); Sigma-Aldrich). The resulting suspensions were filtered twice through Miracloth (Millipore, 25 μm pore size), and the flowthrough was spun (10,000$g$, 5 min, 4 °C) for collection of the nuclei at the bottom of the tube. The supernatant was removed, and the pellet was resuspended in 75 μl nuclei lysis buffer (50 mM Tris-HCl pH 8.0, 10 mM EDTA pH 8.0, 1% SDS) and then incubated on ice for 30 min. Samples were diluted by addition of 625 μl ChIP dilution buffer (16.7 mM Tris-HCl pH 8.0, 167 mM NaCl, 1.2 mM EDTA, 0.01% SDS) and sonicated four times with 95% sonication amplitude (SONICS Vibracell VCX130 sonicator) for 30 s. The volume was adjusted to 900 μl with ChIP dilution buffer containing 1.1% Triton X-100, and the samples were centrifuged (10,000$g$, 5 min, 4 °C). Clean supernatants were transferred to fresh tubes for the subsequent immunopurification steps, and 18 μl (2%) of each sample was put aside to be used as an 'input' control. Then, 5 μg of GFP antibody (Roche, catalogue no. 11814460001) was added to the supernatant with a final concentration of 5.5 ng μl$^{-1}$, and the antibody was pulled down from the nuclear lysate after sonication using Dynabeads Protein G magnetic beads (Thermo Scientific). At the end of the reverse cross-link step, DNA was purified using a QIAquick PCR Purification Kit (Qiagen) and eluted in a final volume of 30 μl. Enrichment of genes in the chromatin immunoprecipitate was detected through real-time qPCR using a CFX384 Touch Real-Time PCR Detection System (Bio-Rad), with a triple technical replicate for each of the four biological replicates, applying the percent input method. To calculate the ratio between immunoprecipitated DNA and input DNA, $\log_{250}$ was subtracted from the raw $C_t$ values of the input, before the $C_t$ immunoprecipitated value was obtained. The final enrichment was calculated as $2^{-ddC_t}$. The primer sequences used are listed in Supplementary Table 10.

**RNA sequencing.** For reoxygenation treatment, Col-0 and *erfVII* seedlings were grown for 7 days in a 6-well plate in vertical media and treated with severe hypoxia (0.1% O$_2$) or air (21% O$_2$) for 24 h in the dark, followed by 3 h or reoxygenation aerobic conditions, in the dark, for 3 h. For oxidative stress treatment, Col-0 and *erfVII* seedlings were grown for 7 days in a 6-well plate in liquid media, followed by 6 h treatment with 1 mM TBHP, in the dark, or mock treatment. At the end of the treatment, samples were collected and frozen in liquid nitrogen. RNA was isolated using a GeneJET RNA Purification Kit (Thermo Scientific) per the manufacturer's instructions. RNA sequencing was performed in paired-end mode using Illumina Sequencing PE150 on the NovaSeq 6000 platform (Novogene). Transcriptomic analyses were conducted in R (v.4.3.1). After a quality check using FastQC, we aligned the reads on the *A. thaliana* full genome (TAIR 10) using Rsubread[74] (v.2.16.1) and counted them using featureCounts[75] (in the Rsubread package). A multidimensional scaling plot was used to assess similarities and differences between samples on the basis of their gene expression profiles. Differentially expressed genes were identified using edgeR[76] (v.3.42.4). Differentially expressed genes with expression fold change of at least |1.5| and false discovery rate less than 0.05 (Supplementary Tables 1 and 2) were selected for subsequent analysis. Gene ontology enrichment analysis of the differentially expressed genes was conducted using clusterProfiler[77] (v.4.10.1).

**Motif discovery and enrichment.** Overlapping downregulated genes in *erfVII* seedlings compared with the wild type under TBHP and reoxygenation treatments were used for DNA motif discovery with STREME (Sensitive, Thorough, Rapid, Enriched Motif Elicitation) in the MEME Suite (v.5.5.9)[78]. For each gene, a 2.5-kb genomic region upstream of the start codon was extracted from the *A. thaliana* TAIR10 reference genome and used as the input sequence set. A shuffled version of the input sequences served as the background control. Identified motifs were subsequently compared with known motif databases using TomTom (MEME Suite v.5.5.9)[79]. Full results are reported in Supplementary Tables 3–6.

## Statistical analyses

Statistical analyses were performed and graphs were made and annotated using GraphPad Prism 10.2.3(403) and R Statistical Software (v.4.3.1). Normal distribution and homogeneity of variance of data were evaluated using by Shapiro–Wilk test and Levene's test, respectively. Student's *t*-test, Mann–Whitney test, analysis of variance or Kruskal–Wallis test followed by Tukey's HSD post hoc test ($P < 0.05$) was performed to establish the statistical significance of differences. Additional information is provided in figure legends. Sample sizes were not statistically pre-determined. All statistical analyses are provided in Supplementary Table 11.

## Materials availability

All unique and/or stable reagents generated in this study are available from the lead contacts without restriction.

## Reporting summary

Further information on research design is available in the Nature Portfolio Reporting Summary linked to this article.

## Data availability

RNA sequencing raw data generated for this study have been deposited in the Sequence Read Archive at the National Centre for Biotechnology Information under BioProject IDs PRJNA1380489 (for the experiments that compared Col-0 and *erfVII* transcriptomes under normoxia, hypoxia and reoxygenation conditions) and PRJNA1171625 (for the experiments comparing the Col-0 and *erfVII* transcriptomes in the control and TBHP treatment groups). Full versions of all images are available at Zenodo (https://doi.org/10.5281/zenodo.18723507)[80]. Source data are provided with this paper.

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

**Acknowledgements** We thank E. Pires of the Department of Chemistry, University of Oxford for assistance with LC–MS/MS experiments, and K. Carroll of the Department of Chemistry and Biochemistry, Florida Atlantic University, for providing the DiaAlk probe. This work was supported by the European Research Council (ERC) under the European Union Horizon 2020 Research and Innovation Program (grant 864888 supporting S.A., D.M.G. and E.F.; and grant 101001320 supporting F.L., L.D.C. and D.Z.). The Biotechnology and Biological Sciences Research Council (UKRI-BBSRC) supported M.P., Y.H. (grant number BB/T008784/1), F.L. and V.S. (grant number BB/Z516946/1). B.F. and Y.T. were financially supported by the Erasmus+ programme. S.A. was financially supported by a Universität Münster Fellowship from the University of Münster Internationalization Fund.

**Author contributions** S.A., M.P., B.G., M.S., E.F. and F.L. conceived and designed the study. S.A., M.P., M.L.-P., S.L., Y.H., V.S., L.D.C., Y.T., D.Z., B.F., D.M.G., W.K.M., P.B., E.F. and F.L. conducted experiments. S.A. performed the experiments and analysed the data shown in Figs. 3b,c and 4a–f and Extended Data Figs. 3a–g, 4a–f, 5a–f and 6b; M.P. performed the experiments and analysed the data shown in Figs. 1a–g, 2a,c, 3a,d and 5a,c–e and Extended Data Figs. 1a–g, 2a,b,f,g, 3h–i, 6a,c, 7a,c,d and 8a–d; M.L.-P. performed the experiments and analysed the data shown in Fig. 3f and Extended Data Fig. 3j–l with contributions from B.G.; S.L. and S.A. performed the experiments and analysed the data shown in Fig. 1h–j and Extended Data Fig. 1h–m, with contributions from P.B. and M.S.; Y.H. performed the experiments and analysed the data shown in Fig. 2b and Extended Data Fig. 2e; V.S. performed the experiments and analysed the data shown in Extended Data Fig. 2d and, together with L.D.C. Fig. 5b; B.F. contributed to Extended Data Fig. 8c–f; Y.T. produced Extended Data Fig. 7e; D.M.G. contributed to Fig. 4a–f and Extended Data Figs. 4e,f and 5a,b; E.F. contributed to Fig. 4e; and W.K.M. contributed to and analysed data shown in Fig. 4e. F.L. designed and assembled the DNA constructs and generated the transgenic plants with help from D.Z. and Y.T. M.P. and S.A. prepared figures. S.A., M.P., E.F. and F.L. wrote the manuscript with input from all authors.

**Competing interests** The authors declare no competing interests.

**Additional information**
**Correspondence and requests for materials** should be addressed to Emily Flashman or Francesco Licausi.

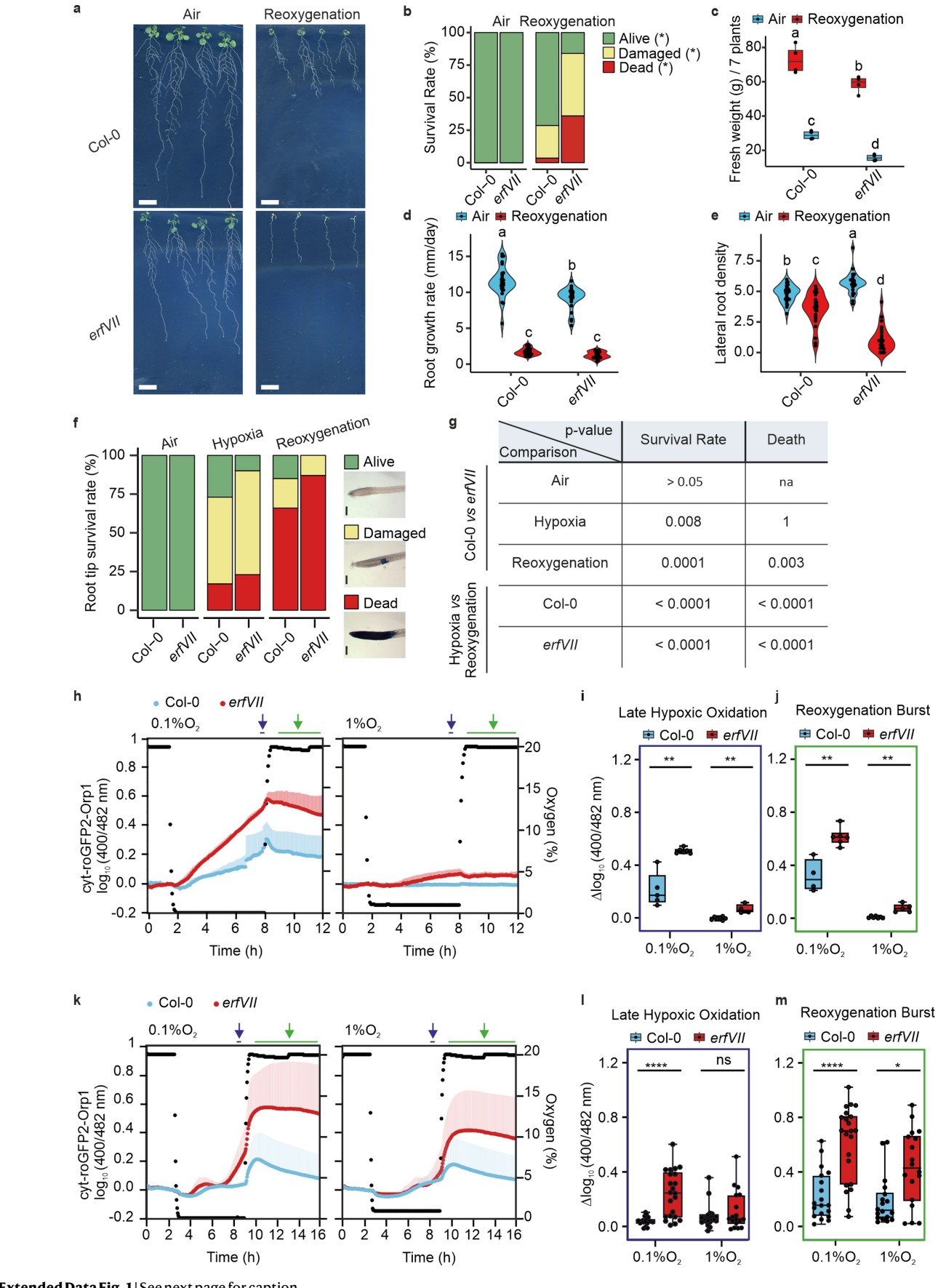

**Extended Data Fig. 1** | See next page for caption.

**Extended Data Fig. 1 | ERFVIIs mediate plant tolerance upon reoxygenation.**
(a) Phenotype of Col-0 and *erfVII* seedlings after hypoxia treatment followed by
4 days reoxygenation, or air control (scale bar, 1 cm). (b) Percentage of alive,
damaged or dead seedlings after 4 days of post-hypoxia reoxygenation or air
control (Col-0 in air, n = 27; Col-0 in hypoxia, n = 28; *erfVII* in air and in hypoxia,
n = 25). (c) Fresh weight after 4 days of reoxygenation or air control (n = 4, each
replicate representing ~7 seedlings). (d) Growth rate of Col-0 and *erfVII* primary
roots after 4 days of reoxygenation or air control (Col-0 in air, n = 27; Col-0 in
hypoxia, n = 28; *erfVII* in air and in hypoxia, n = 25). (e) Lateral root density of
Col-0 and *erfVII* seedlings after 4 days of post-hypoxia reoxygenation or air
control (Col-0 in air, n = 27; Col-0 in hypoxia, n = 28; *erfVII* in air and in hypoxia,
n = 25). (f) Percentage of alive, damaged or dead root tips after 24 h of air, hypoxia
(0.1% $O_2$) or 6 h of reoxygenation (Col-0 in air and *erfVII* in hypoxia, n = 30;
Col-0 in hypoxia, n = 31; *erfVII* in air and in reoxygenation, n = 23; Col-0 in
reoxygenation, n = 26; scale bar, 20 μm). (g) Statistical analyses of root tip
survival after 24 h of air or, hypoxia (0.1% $O_2$) 6 h of reoxygenation (Col-0
in air and *erfVII* in hypoxia, n = 30; Col-0 in hypoxia, n = 31; *erfVII* in air and in
reoxygenation, n = 23; Col-0 in reoxygenation, n = 26; scale bar, 20 μm). Multi-well
fluorimetry of cytosolic oxidative stress using seven-days old Arabidopsis
seedlings (h) or submerged leaf discs of five-week old (k) Arabidopsis plants stably
expressing the biosensor roGFP2-Orp1 in Col-0 wildtype and *erfVII* background,
each normalized to the baseline oxidative state before the start of the hypoxic
treatment. Amplitudes of late hypoxic roGFP2-Orp1 oxidation in seedlings
(i, j) or leaf discs (l, m) before induction of reoxygenation (purple arrow) and
maximum oxidative burst during reoxygenation (green arrow). At 0.1% $O_2$,
sample sizes were: seedlings Col-0 (n = 4), *erfVII* (n = 6); leaf discs Col-0 (n = 19),
*erfVII* (n = 22). At 1% $O_2$, sample sizes were: seedlings Col-0 (n = 7), *erfVII* (n = 4);
leaf discs Col-0 (n = 18), *erfVII* (n = 18). Statistical analyses were conducted using:
(b, f, g): two-sided χ2 test followed by a post-hoc test with a Bonferroni correction
was used to analyse this dataset (p < 0.05); (c, d, e) two-way ANOVA followed by
Tukey's HSD test, where different letters indicate statistically different groups
(p < 0.05); (i-j, l-m) two-sided Mann-Whitney test (p < 0.05). In c, i-j, l-m, boxplots
indicate median (middle line), 25th and 75th percentiles (box limits), whiskers
denote the 1.5x interquartile range; outliers are shown as individual points.

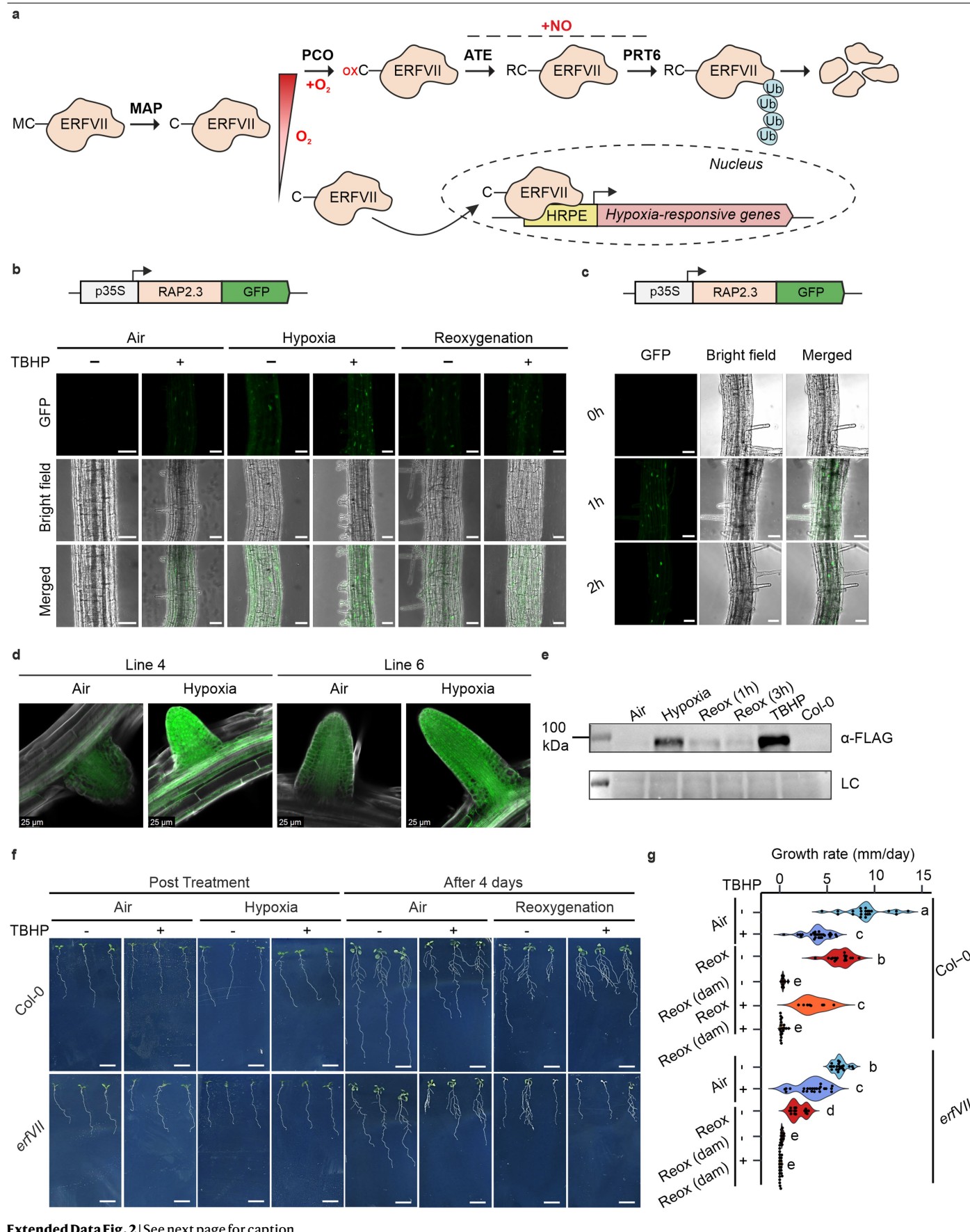

**Extended Data Fig. 2** | See next page for caption.

**Extended Data Fig. 2 | Oxidative stress and reoxygenation stabilize ERFVII and contribute to survival in wild type.** (a) Schematic representation of N-degron pathway-controlled destabilisation of ERFVII. Methionine (M) in the N-terminal position is target of the methionine aminopeptidase enzyme (MAP), leaving an exposed cysteine which, in presence of oxygen ($O_2$) is targeted for oxidation by plant cysteine oxidases (PCO). The resulting Cys-sulfinic acid is sequentially modified by arginyl aminotransferases 1/2 (ATE) triggering proteasomal degradation after ubiquitination by proteolysis E3 ligase (PRT6), in presence of nitric oxide (NO). Upon hypoxic conditions, ERFVII are stabilized and relocalise to the nucleus where they bind the hypoxia responsive promoter element (HRPE) triggering the hypoxic response. Additional abbreviation: M, methionine; C, cysteine; oxC, oxidized cysteine; R, arginine; Ub, ubiquitin. (b) Stabilization of 35S:RAP2.3-GFP (green) in seven-day old Arabidopsis seedlings upon 1 mM TBHP or mock treatment in normoxia (21% $O_2$), hypoxia (1% $O_2$) or after 3 h of reoxygenation, in the dark (scale bar, 50 μm). (c) Time-dependent stabilization of 35S:RAP2.3-GFP (green) in seven-day-old seedlings over 2 h of 1 mM TBHP treatment in air (scale bar, 50 μm), n = 1. (d) Stabilization of two independent lines of RAP2.12-FLAG (green) upon 6 h of air (21% $O_2$) or hypoxia (1% $O_2$), in the dark. (e) Western blot analysis of a second independent line of RAP2.12-FLAG in air, hypoxia (6 h, 1% $O_2$), followed by 1 h or 3 h reoxygenation, or 1 mM TBHP treatment, in the dark. Loading control (LC) corresponds to a compacted image of the membrane after Ponceau staining. Unedited gel images are shown in Supplementary Fig. 1. (f) Phenotype of Col-0 and *erfVII* seedlings after hypoxia treatment (or air control), with or without 2 h 1 mM TBHP pre-treatment, and after 4 days reoxygenation (scale bar, 1 cm). (g) Growth rate of Col-0 and *erfVII* primary roots after 4 days of reoxygenation or air control, with or without 2 h 1 mM TBHP pre-treatment (Col-0 in air, n = 22; Col-0 in air + TBHP and hypoxia + TBHP, n = 28; Col-0 in hypoxia, n = 27; *erfVII* in air, n = 21; *erfVII* in air + TBHP, n = 23; *erfVII* in hypoxia, n = 26; *erfVII* in hypoxia + TBHP. n = 25). Seedlings per category are divided into damaged (dam) and not damaged depending on main root growth after reoxygenation. Statistical analyses were conducted using one-way ANOVA followed by Tukey's HSD test (p < 0.05), where different letters indicate statistically distinct groups. Exact p-values provided in Supplementary Table 11. Experiments producing Fig. 2b-d were repeated once, the western blot shown in 2e was repeated twice.

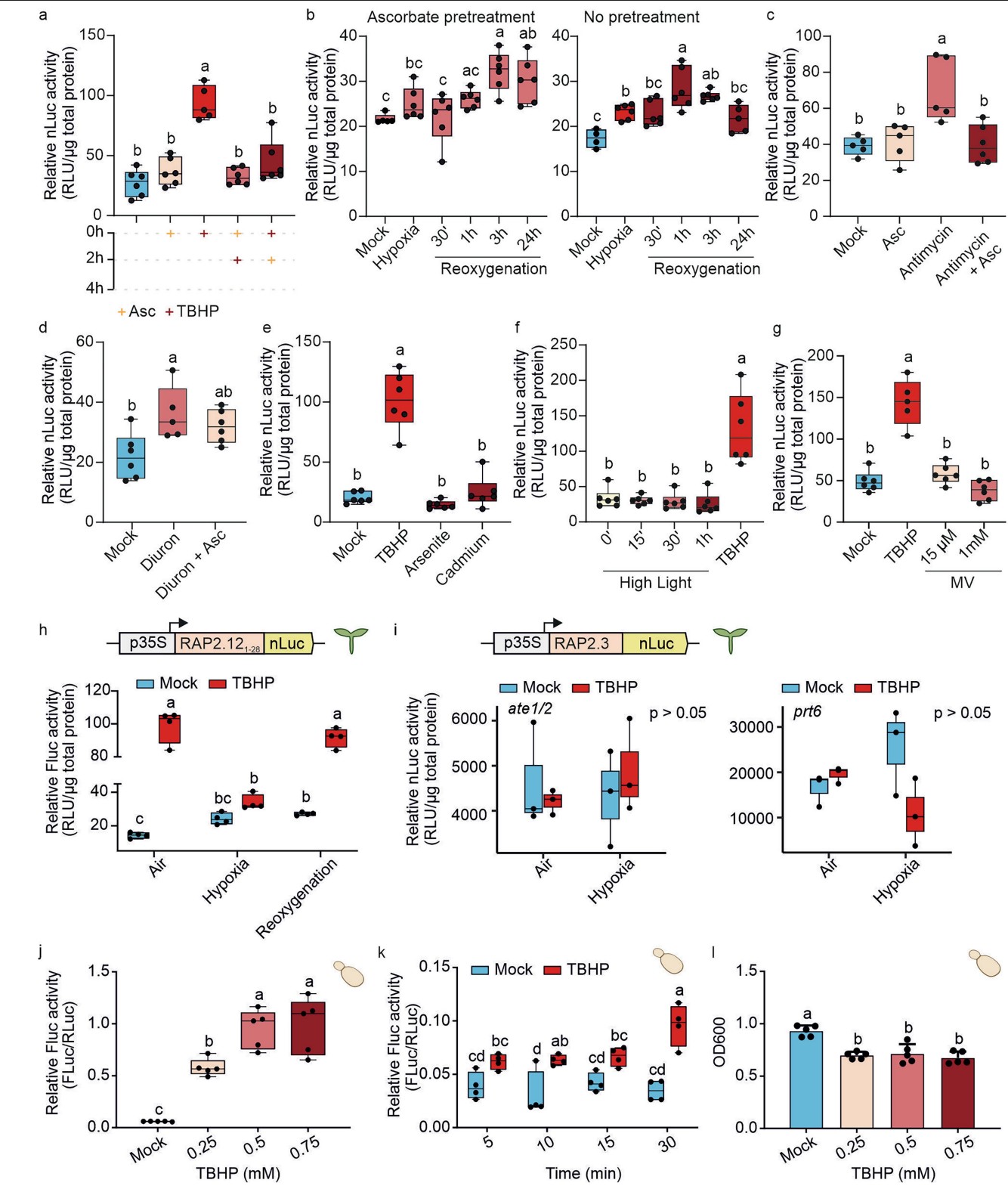

**Extended Data Fig. 3** | See next page for caption.

**Extended Data Fig. 3 | Measurement of ERFVII stability upon oxidative stress in plant and yeast-based reporter systems.** (a) Relative nLuc activity of seedlings treated for 4 h with 10 mM ascorbate (Asc, ROS scavenger; n = 6), *tert*-butyl hydroperoxide (TBHP, ROS inducer; n = 5), or sequential treatments in which the second compound was added at the 2 h time point: Asc followed by TBHP, or TBHP followed by Asc (n = 6). (b) Left: relative nLuc activity of seedlings exposed to air (mock; n = 5) or hypoxia (6 h, 0.1 % $O_2$; n = 6), followed by reoxygenation (30 min to 24 h) with ascorbate pre-treatment (n = 6). Right: relative nLuc activity of seedlings exposed to air (mock; n = 4) or hypoxia (6 h, 0.1 % $O_2$; n = 6), followed by reoxygenation (30 min to 3 h, n = 6; 24 h, n = 5) without ascorbate pre-treatment. (c) Relative nLuc activity of seedlings treated for 4 h with mock (n = 6), ascorbate (Asc, n = 6), antimycin A (mitochondrial ROS inducer) with (n = 5) or without ascorbate treatment (n = 6). (d) Relative nLuc activity of seedlings treated for 4 h with 1 mM diuron (photosystem II inhibitor), with (n = 5) or without ascorbate treatment (n = 6). (e) Relative nLuc activity of seedlings treated for 4 h with mock, 10 mM arsenite or cadmium (ROS inducers), with TBHP as a positive control (n = 6). (f) Relative nLuc activity of seedlings exposed to high light (1600 to 1800 μmoles m$^{-2}$ s$^{-1}$) for 15 minutes to 1 h, with TBHP as a positive control (n = 6). (g) Relative nLuc activity of seedlings treated for 4 h with methyl viologen (MV) at indicated doses (n = 6), with TBHP as positive control (n = 5). All experiments (a-g) were performed using seven-day-old 35S:RAP2.3-nLuc seedlings. (h) Relative FLuc activity of 35S:RAP2.12$_{2-28}$-FLuc seedlings in air, hypoxia (1 % $O_2$), for 6 h, followed by 3 h reoxygenation upon 1 mM TBHP or mock treatment (n = 4). (i) Relative FLuc activity of 35S:RAP2.3-nLuc in *ate1/2* and *prt6* background seedlings in air, hypoxia (1 % $O_2$), for 6 h, followed by 3 h reoxygenation upon 1 mM TBHP or mock treatment (n = 3). (j) Effect of TBHP doses on C-DLOR activity (n = 5). (k) Relative FLuc activity of yeast expressing C-DLOR and PCO4 over time (n = 4). (l) Measurement of optical density at 600 nm ($OD_{600}$) of yeast culture subjected to different doses of TBHP. Lines show the mean, with error bars representing the standard deviation (n = 5). Statistical analyses were conducted using: one-way (a-g, j, l) or two-way (h, i, k) ANOVA followed by Tukey's HSD test (p < 0.05), where different letters indicate statistically different groups. In a-k, boxplots indicate median (middle line), 25th and 75th percentiles (box limits), whiskers denote the 1.5x interquartile range; outliers are shown as individual points.

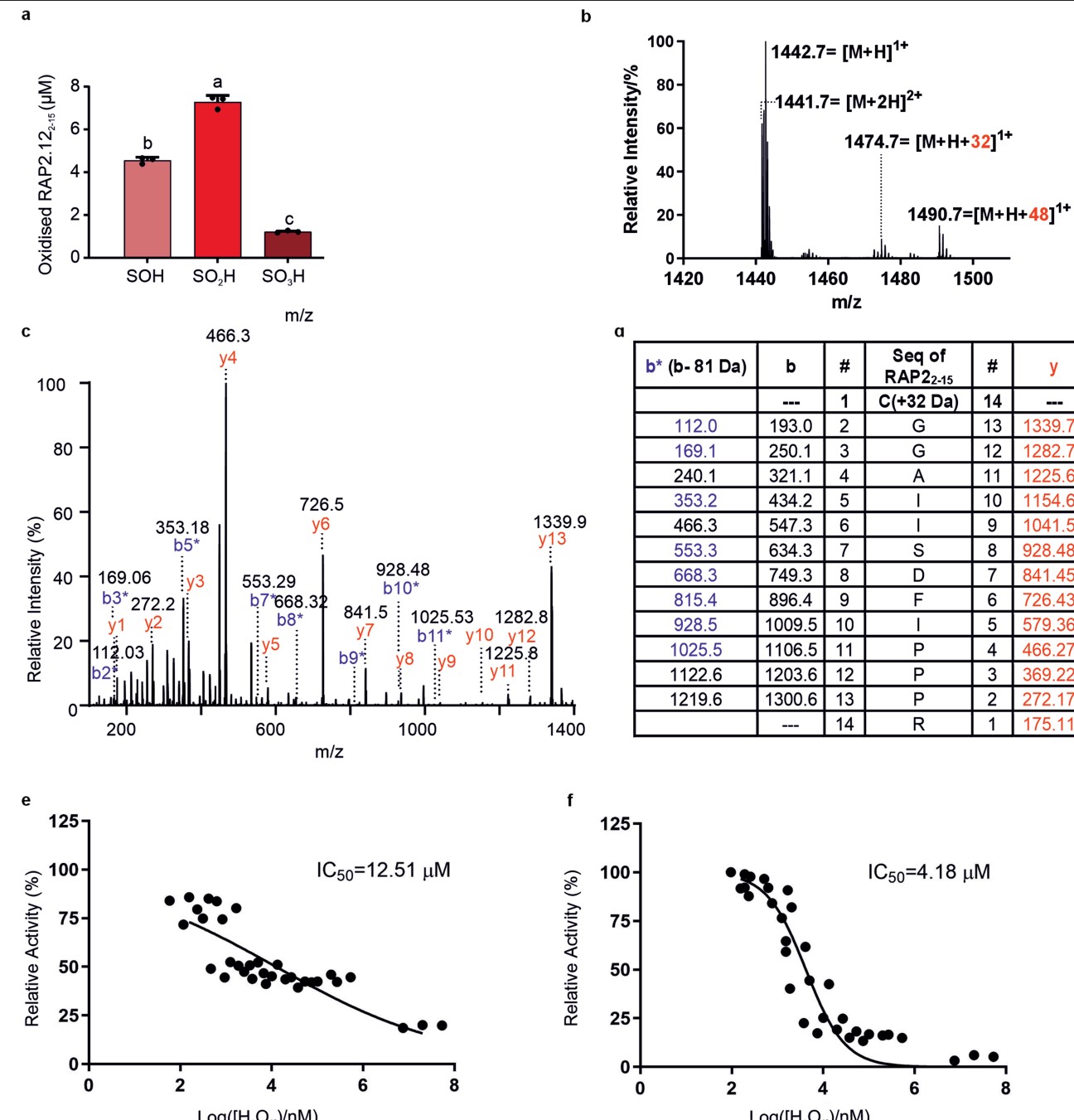

**a**

**b**

**c**

**d**

| b* (b- 81 Da) | b | # | Seq of RAP2$_{2-15}$ | # | y |
|---|---|---|---|---|---|
| | --- | 1 | C(+32 Da) | 14 | --- |
| 112.0 | 193.0 | 2 | G | 13 | 1339.74 |
| 169.1 | 250.1 | 3 | G | 12 | 1282.72 |
| 240.1 | 321.1 | 4 | A | 11 | 1225.69 |
| 353.2 | 434.2 | 5 | I | 10 | 1154.66 |
| 466.3 | 547.3 | 6 | I | 9 | 1041.57 |
| 553.3 | 634.3 | 7 | S | 8 | 928.489 |
| 668.3 | 749.3 | 8 | D | 7 | 841.457 |
| 815.4 | 896.4 | 9 | F | 6 | 726.430 |
| 928.5 | 1009.5 | 10 | I | 5 | 579.361 |
| 1025.5 | 1106.5 | 11 | P | 4 | 466.277 |
| 1122.6 | 1203.6 | 12 | P | 3 | 369.225 |
| 1219.6 | 1300.6 | 13 | P | 2 | 272.172 |
| | --- | 14 | R | 1 | 175.119 |

**e**

IC$_{50}$=12.51 μM

**f**

IC$_{50}$=4.18 μM

**Extended Data Fig. 4 | H$_2$O$_2$ impact on RAP2$_{2-15}$ peptide oxidation via non-enzymatic N-terminal cysteine modification and *At*PCO1/2 inhibition.**
(a) Quantification of oxidised RAP2$_{2-15}$ species after direct H$_2$O$_2$ treatment (1 mM) of 200 μM peptide in real time by RapidFire mass spectrometry reveals ~4.5 μM sulfenic (SOH), ~7 μM sulfinic (SO2H) and ~1 μM sulfonic acid (SO3H) on the N-terminal cysteine residue after 1 h compared to PCO-catalysed oxidation of RAP2$_{2-15}$ (>30 μM after 10 minutes). Lines show the mean, with error bars representing the standard deviation (n = 3). Statistically significant differences were determined by one-way ANOVA followed by Tukey's HSD test (p < 0.05). Different letters indicate statistically different groups. (b) Mass spectrum showing mass changes of 200 μM RAP2$_{2-15}$ after 1 mM H$_2$O$_2$ treatment for 1 h then analysed by liquid chromatography mass spectrometry. The predominant peak ([M + H]$^+$ = 1442.7 Da) represents unmodified RAP2$_{2-15}$, however ions likely representing RAP2$_{2-15}$ dimer ([M + 2H]$^{2+}$ = 1441.7 Da) are also present,

potentially arising from Nt-Cys-SOH and subsequent disulfide bond formation. Peaks at 1474.7 Da and 1490.7 Da represent Nt-Cys-SO2H and Nt-Cys-SO3H, respectively. (c) Tandem mass spectrometry confirms H$_2$O$_2$-mediated oxidative modification on Nt-Cys of RAP2$_{2-15}$; Spectrum shows b ions and y ions of fragmented 1474.7 Da peptide (Nt-Cys-SO2H) following H$_2$O$_2$ treatment RAP2$_{2-15}$ (collision energy 80 V). Expected y ions are observed, however b ions were predominantly observed with a consistent mass loss of 81 Da, termed b* ions. These ions likely correspond to loss of SO$_2$ and NH$_3$ upon fragmentation, as has been observed previously for an oxidative modification on N-terminal cysteine[81]. (d) Table showing the matched mass of the predicted fragments (b, b* and y) with the observed mass of RAP2$_{2-15}$ (1474.7 Da) under H$_2$O$_2$ treatment. (e) H$_2$O$_2$ dose-dependent effect on *At*PCO1 enzyme activity (n = 3). (f) H$_2$O$_2$ dose-dependent effect on *At*PCO2 enzyme activity (n = 3). For (e) and (f), 2 μM enzyme was treated with a series of H$_2$O$_2$ concentrations (0–2 mM) at 4 °C for 30 minutes.

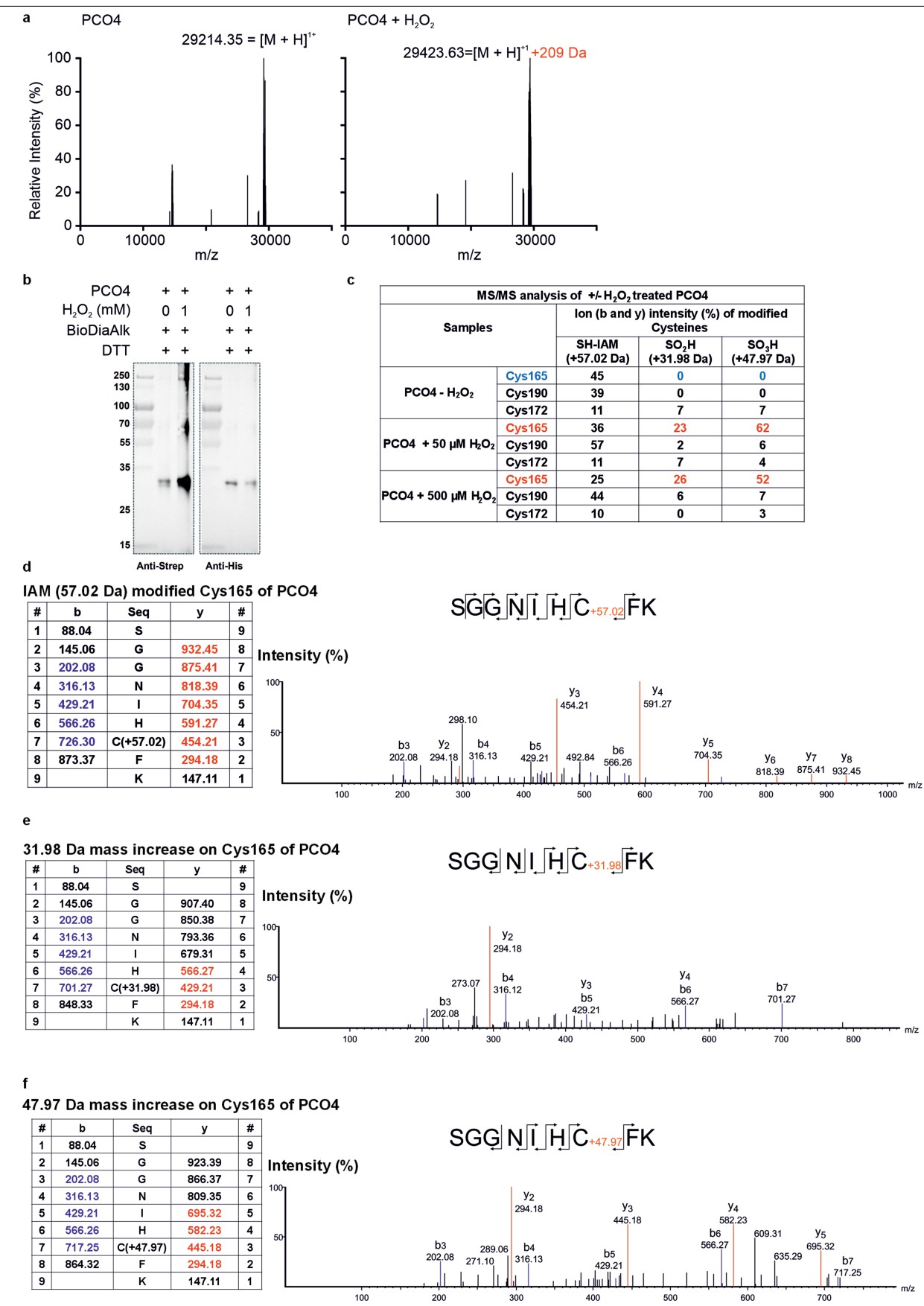

**Extended Data Fig. 5** | See next page for caption.

**Extended Data Fig. 5 | H2O2 inhibits recombinant PCO enzymes.** (a) Intact protein mass spectra of 100 μM $H_2O_2$ treated and non-treated 10 μM $At$PCO4 measured by RapidFire mass spectrometry (representative spectrum of 3 experiments). (b) Streptavidin blot of BioDiaAlk labeling of sulfinic acids in $At$PCO4. This experiment was repeated once. (c) Table showing percentage of ion (b and y) intensity of $H_2O_2$-treated and non-treated $At$PCO4, used to select the Cys165-containing peptide for further interrogation (d) IAM (+57.02) modifications on b ion and y ion fragments of the peptide containing Cys165 of $At$PCO4. (e) Sulfinic acid (+31.98) modifications on b ion and y ion fragments of the peptide containing Cys165 of $At$PCO4. (f) Sulfonic acid (+47.97) modifications on b ion and y ion fragments of the peptide containing Cys165 of $At$PCO4.

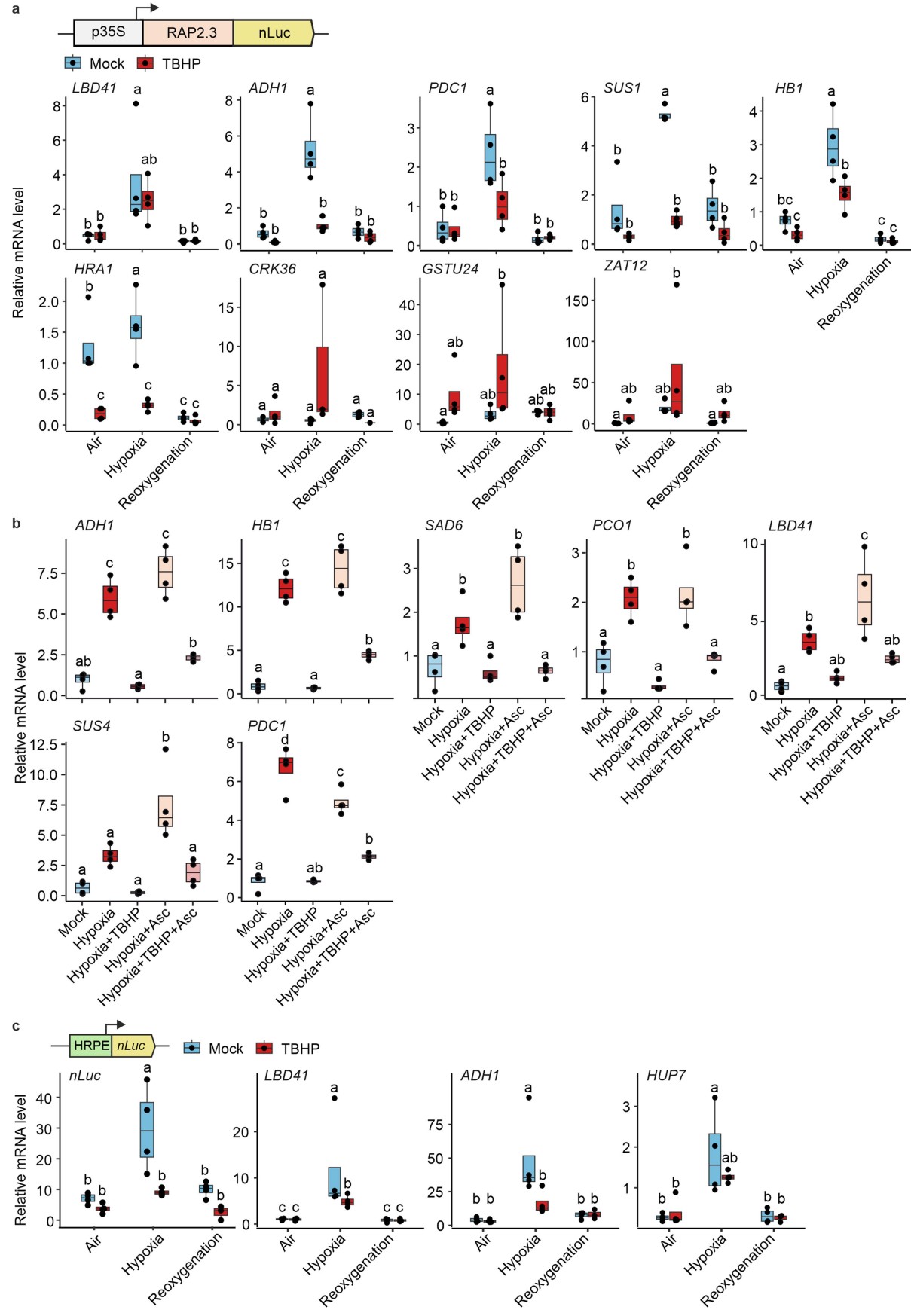

**Extended Data Fig. 6** | See next page for caption.

**Extended Data Fig. 6 | ROS-triggered oxidative stress disrupts hypoxia-responsive gene expression, without damaging the transcriptional machinery.** (a) Relative expression of hypoxia-responsive and ROS-responsive genes in 35S:RAP2.3-nLuc seven-day-old seedlings in air or hypoxia (1% $O_2$), for 6 h, followed by 3 h reoxygenation, upon 1 mM TBHP or mock treatment (n = 4). (b) Relative expression of hypoxia-responsive genes in seven-day-old Col-0 seedlings (n = 4) exposed to 6 h of air (mock), hypoxia (1%) alone, hypoxia combined with 1 mM TBHP, 10 mM ascorbate (Asc), or TBHP and Asc together (n = 4). (c) Relative expression of nLuc and hypoxia-responsive genes in seven-day-old HRPE:nLuc (35S-5'UTR) seedlings subjected to 6 h of air or hypoxia (1 % $O_2$), followed by 3 h of reoxygenation upon TBHP or mock treatment (mock in air, hypoxia, reoxygenation and TBHP in air and hypoxia, n = 4; TBHP in reoxygenation, n = 3). Statistical analyses were conducted using one-way (b) or two-way (a, c) ANOVA followed by Tukey's HSD test (p < 0.05). Different letters indicate statistically different groups. In a-c, boxplots indicate median (middle line), 25th and 75th percentiles (box limits), whiskers denote the 1.5x interquartile range; outliers are shown as individual points.

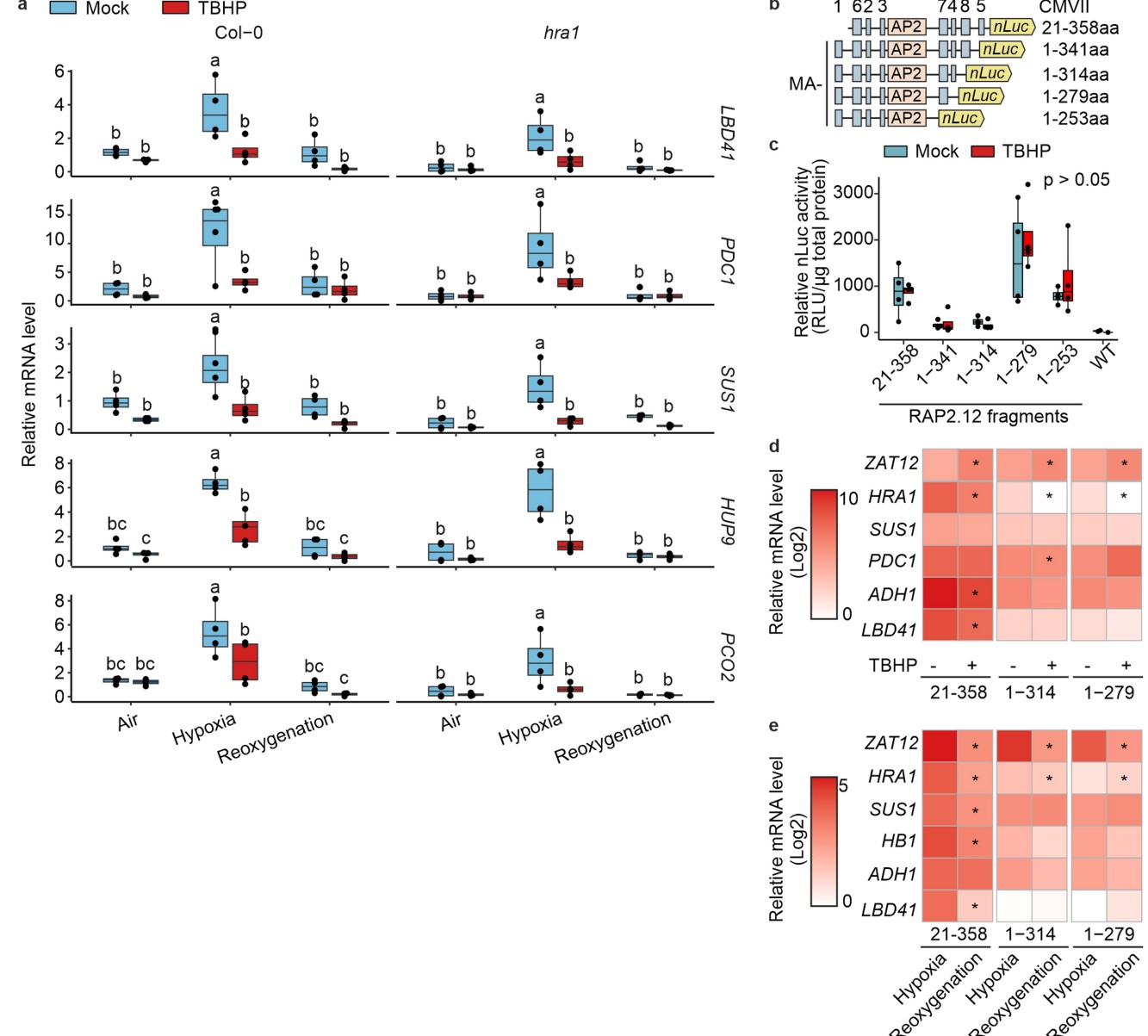

**Extended Data Fig. 7 | Regulation of hypoxia response upon oxidative stress.** (a) Relative mRNA level of five hypoxia-responsive genes (*LBD41, PDC1, SUS1, HUP9, PCO2*) in Col-0 and *hra1-1* seven-day old seedlings subjected to air, hypoxia (1% O$_2$, 6 h) or 3 h reoxygenation with either 1 mM TBHP or mock treatment (n = 4, except for *hra1-1* in reoxygenation for *HUP9, PCO2, SUS1*, where n = 3). Statistical analyses were conducted using two-way ANOVA followed by Tukey's HSD test (p < 0.05). Different letters indicate statistically different groups. Exact p-values provided in Supplementary Data 2. (b) Schematic representation of five RAP2.12 truncated versions fused to nLuc gene generated in this study. (c) Relative nLuc activity of different truncated versions of RAP2.12-nLuc seedlings treated with 1 mM TBHP upon hypoxic conditions for 6 h (n = 4, except Col-0 n = 2). Statistical analyses were conducted using two-sided Student's t test (p < 0.05). (d) Relative mRNA level (Log2) of five hypoxic responsive genes (*HRA1, SUS1, PDC1, ADH1, LBD41*) and a ROS responsive gene (*ZAT12*) in seven-day-old seedlings of three truncated versions of RAP2.12 fused to a nLuc gene subjected to 6 h of hypoxia (1% O$_2$) upon 1 mM TBHP or mock treatment (n = 4). Statistical analyses were conducted using two-sided Student's t test, where asterisks indicate statistical differences between mock and TBHP treatment (p < 0.05). (e) Relative mRNA level (Log2) of five hypoxic responsive genes (*HRA1, SUS1, HB1, ADH1, LBD41*) and a ROS responsive gene (*ZAT12*) in seven-day-old seedlings of three truncated versions of RAP2.12 fused to a nLuc gene subjected to 6 h of hypoxia (1% O$_2$) or hypoxia followed by 3 h of reoxygenation (n = 4, except for 1-279 in hypoxia for *HRA1* and *SUS1*, 1-279 in reoxygenation for *SUS1* and 1-314 in hypoxia for *HRA1*, where n = 3). Statistical analyses were conducted using two-sided Student's t test, where asterisks indicate statistical differences between mock and TBHP treatment (p < 0.05). In a and c, boxplots indicate median (middle line), 25th and 75th percentiles (box limits), whiskers denote the 1.5x interquartile range; outliers are shown as individual points.

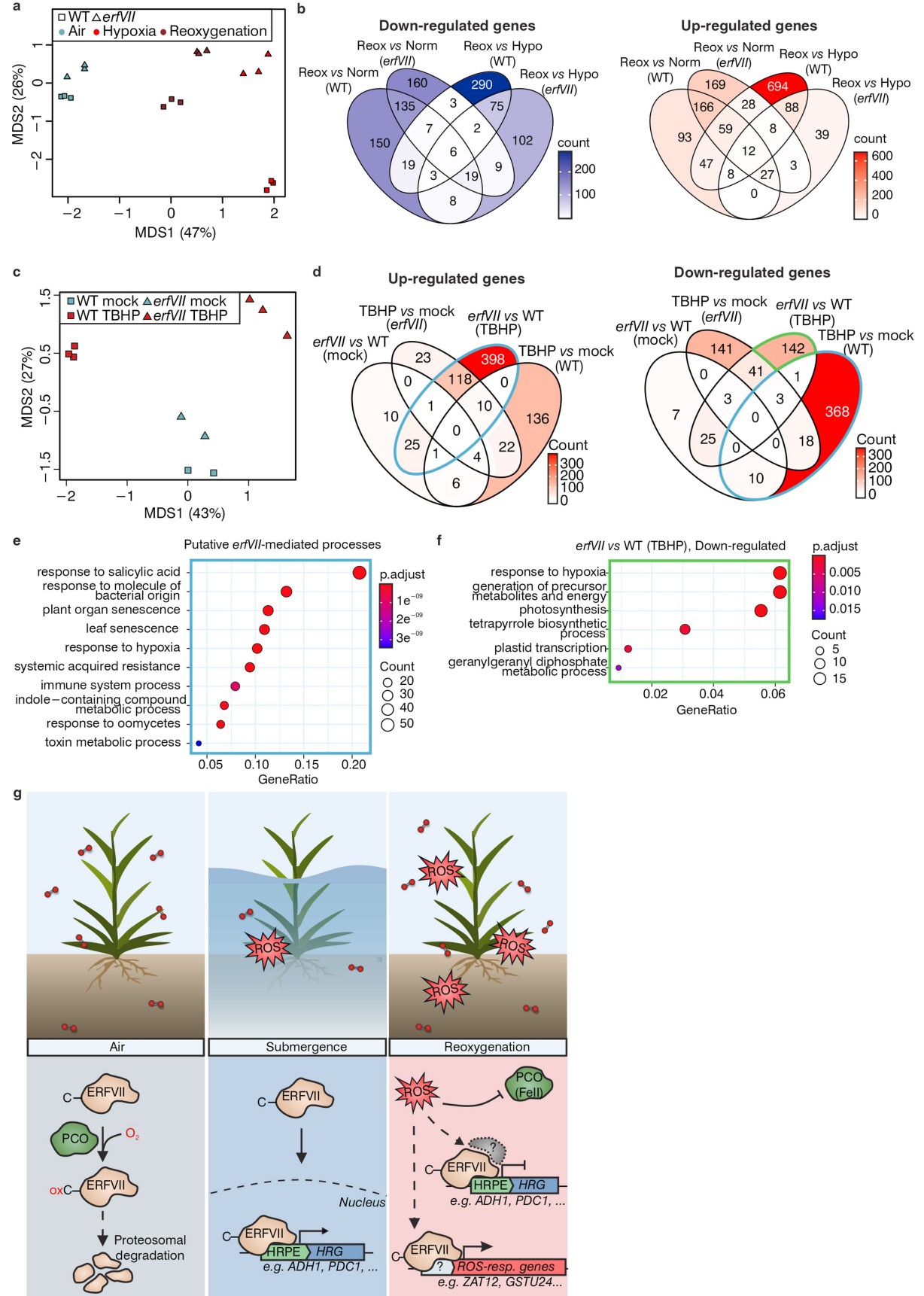

**Extended Data Fig. 8 |** See next page for caption.

**Extended Data Fig. 8 | Oxidative stress induces an ERFVII-dependent transcriptional change, with shared traits between TBHP treatment and reoxygenation.** (a) Multidimensional scaling plot (MDS) conducted on the normalized gene expression values of *erfVII* and Col-0 seedlings treated upon air, strict hypoxia (0.1% $O_2$) for 24 h or reoxygenation for 3 h (n = 3). Horizontal and vertical coordinates show MDS1 and MDS2, respectively, with the amount of variance contained in each component. Each point in the plot represents a biological replicate. Differences in symbols and colour indicate difference in genotype and treatment, respectively. (b) Venn Diagram indicating the overlap of down-regulated and up-regulated genes for *erfVII* or Col-0 treated upon air, hypoxia or reoxygenation. (c) The mock-treated *erfVII* mutant modulates gene expression as if experiencing mild oxidative stress under control conditions, as demonstrated by the separation of samples in the dimension primarily affected by TBHP. Multidimensional scaling plot (MDS) conducted on the normalized gene expression values of *erfVII* and Col-0 seedlings treated with TBHP or mock. Horizontal and vertical coordinates show MDS1 and MDS2, respectively, with the amount of variance contained in each component (43% and 27%). (d) Venn Diagram indicating the overlap of up-regulated and down-regulated genes for *erfVII* or Col-0 treated upon TBHP or mock conditions. (e) GO enrichment results for biological process of genes that are up-regulated genes in *erfVII* seedlings compared to wild type, upon TBHP treatment, and down-regulated in the wild type upon TBHP treatment compared to mock. (f) GO enrichment results for biological process of the down-regulated genes in TBHP-treated *erfVII* seedling compared to wild type. (e-f) Circle size indicates the gene count per GO term, with color maps indicating the False Discovery Rate (FDR) value (p.adjust). Statistical significance was determined using a one-sided hypergeometric test; p-values were adjusted for multiple comparisons using the Benjamini-Hochberg method. (g) ERFVIIs transcription factors are constitutively expressed under aerobic conditions and continuously degraded via the N-degron pathway. Upon low oxygen conditions, such as submergence, lack of oxygen reduces PCO activity, leading to ERFVIIs stabilization and relocalisation into the nucleus, where they activate anaerobic-gene expression. Upon reoxygenation, a ROS burst occurs which inhibits PCO activity, therefore stabilizing ERFVIIs. Additionally, ROS redirects the transcriptional machinery inducing the expression of oxidative stress genes and repressing the canonical hypoxic response.

Francesco Licausi

# Reporting Summary

## Statistics

For all statistical analyses, confirm that the following items are present in the figure legend, table legend, main text, or Methods section.

| n/a | Confirmed | |
|---|---|---|
| ☐ | ☒ | The exact sample size (*n*) for each experimental group/condition, given as a discrete number and unit of measurement |
| ☐ | ☒ | A statement on whether measurements were taken from distinct samples or whether the same sample was measured repeatedly |
| ☐ | ☒ | The statistical test(s) used AND whether they are one- or two-sided<br>*Only common tests should be described solely by name; describe more complex techniques in the Methods section.* |
| ☒ | ☐ | A description of all covariates tested |
| ☒ | ☐ | A description of any assumptions or corrections, such as tests of normality and adjustment for multiple comparisons |
| ☐ | ☒ | A full description of the statistical parameters including central tendency (e.g. means) or other basic estimates (e.g. regression coefficient) AND variation (e.g. standard deviation) or associated estimates of uncertainty (e.g. confidence intervals) |
| ☐ | ☒ | For null hypothesis testing, the test statistic (e.g. *F*, *t*, *r*) with confidence intervals, effect sizes, degrees of freedom and *P* value noted<br>*Give P values as exact values whenever suitable.* |
| ☒ | ☐ | For Bayesian analysis, information on the choice of priors and Markov chain Monte Carlo settings |
| ☒ | ☐ | For hierarchical and complex designs, identification of the appropriate level for tests and full reporting of outcomes |
| ☐ | ☒ | Estimates of effect sizes (e.g. Cohen's *d*, Pearson's *r*), indicating how they were calculated |

*Our web collection on statistics for biologists contains articles on many of the points above.*

## Software and code

Policy information about availability of computer code

| | |
|---|---|
| Data collection | LC-MS/MS analysis of H2O2-treated AtPCO4: mass spectrometry data were acquired using the Orbitrap Eclipse mass spectrometer. Instrument control was through Orbitrap Eclipse Tune 3.5/3.1 and Xcalibur 4.5/4.4. RapidFire MS analysis of RAP22-15 oxidation: mass spectrometry data were acquired using an Agilent RapidFire RF360 sampling robot connected to an Agilent 6530 Accurate-Mass Q-ToF mass spectrometer. Spectra were visualised on Qualitative Analysis (version B.07.00). Colorimetric intensities were collected with a EPSON Perfection V750 PRO scanner. Histochemical staining was conducted with a Leica M165C stereo microscope Confocal imaging was conducted with ZEISS LSM 880 Airyscan microscope (ZEN Lite software (version 3.11)). Ratiometric readout of H2O2 biosensor was performed using a multiwell fluorimeter ClarioStar Plus (BMG Labtech). RNA sequencing: Illumina Sequencing PE150 program on the NovaSeq 6000 platform (Novagene). |
| Data analysis | Data were analysed using GraphPad Prism 10.2.3(403) and R Statistical Software (version 4.3.1, Foundation for Statistical Computing, Vienna, Austria). Image analysis with ImageJ (version 1.54j). Confocal images were analysed using ZEISS ZEN Lite software (version 3.11) RapidFire MS data were analysed using Agilent RapidFire Integrator (version 4.3.0.17235) to calculate integrated peak areas. Peptide fragmentation by LC-MS/MS: peptide analysis was conducted using Peaks v. 8.5 and peptides were compared to predicted fragment patterns calculated using the University of California, San Francisco webpage tool Protein Prospector version 6.3.1. Transcriptomic analyses (conducted in R software version 4.3.1) were aligned on the Arabidopsis thaliana full genome using Rsubread (version |

2.16.1) and counted using featureCounts software (within the Rsubread package). Differentially expressed genes were identified using edgeR version 3.42.4. GO term enrichment analysis was conducted using clusterProfiler version 4.10.1.
DNA motif discovery was conducted on STREME (Sensitive, Thorough, Rapid, Enriched Motif Elicitation) and compared to known motif databases using Tomtom within the MEME Suite (version 5.5.9).

For manuscripts utilizing custom algorithms or software that are central to the research but not yet described in published literature, software must be made available to editors and reviewers. We strongly encourage code deposition in a community repository (e.g. GitHub). See the Nature Portfolio guidelines for submitting code & software for further information.

## Data

Policy information about availability of data

All manuscripts must include a data availability statement. This statement should provide the following information, where applicable:
- Accession codes, unique identifiers, or web links for publicly available datasets
- A description of any restrictions on data availability
- For clinical datasets or third party data, please ensure that the statement adheres to our policy

RNA sequencing raw data generated for this study has been deposited in the Sequence Read Archive (SRA) at the National Centre for Biotechnology Information under BioProject ID PRJNA1380489 and PRJNA1171625 for RNA-sequencing of reoxygenation and oxidative stress, respectively. Numerical data used to generate the graphs displayed in Figures and Extended Data Figures are provided as Source Data supplementary files. Full version of all images are available at https://doi.org/10.5281/zenodo.18723507.

## Research involving human participants, their data, or biological material

Policy information about studies with human participants or human data. See also policy information about sex, gender (identity/presentation), and sexual orientation and race, ethnicity and racism.

| Reporting on sex and gender | N/A |
|---|---|
| Reporting on race, ethnicity, or other socially relevant groupings | N/A |
| Population characteristics | *Describe the covariate-relevant population characteristics of the human research participants (e.g. age, genotypic information, past and current diagnosis and treatment categories). If you filled out the behavioural & social sciences study design questions and have nothing to add here, write "See above."* |
| Recruitment | N/A |
| Ethics oversight | N/A |

Note that full information on the approval of the study protocol must also be provided in the manuscript.

# Field-specific reporting

Please select the one below that is the best fit for your research. If you are not sure, read the appropriate sections before making your selection.

☒ Life sciences          ☐ Behavioural & social sciences          ☐ Ecological, evolutionary & environmental sciences

For a reference copy of the document with all sections, see nature.com/documents/nr-reporting-summary-flat.pdf

# Life sciences study design

All studies must disclose on these points even when the disclosure is negative.

| Sample size | Described in each figure legend. Sample size was defined based on the power of statistical analysis to be used afterwards and on the limitations imposed by sample handling. |
|---|---|
| Data exclusions | No data were excluded |
| Replication | RNA experiments and BioDiaAlk probes were performed once. All other experiments were repeated twice, with similar results confirming replicability |
| Randomization | Plants were randomly assorted at growth facilities and their position randomly permutated to account for covariates (light and temperature gradients). (Bio)chemical and molecular analyses did not require randomisation to account for covariates due to the homogenenous conditions on lab benches. |
| Blinding | Investigators were not blinded but this should have no or minimal effect to the outcome of the experiments described in this manuscript. |

# Reporting for specific materials, systems and methods

We require information from authors about some types of materials, experimental systems and methods used in many studies. Here, indicate whether each material, system or method listed is relevant to your study. If you are not sure if a list item applies to your research, read the appropriate section before selecting a response.

## Materials & experimental systems

| n/a | Involved in the study |
|---|---|
| ☐ | ☒ Antibodies |
| ☐ | ☒ Eukaryotic cell lines |
| ☒ | ☐ Palaeontology and archaeology |
| ☒ | ☐ Animals and other organisms |
| ☒ | ☐ Clinical data |
| ☒ | ☐ Dual use research of concern |
| ☐ | ☒ Plants |

## Methods

| n/a | Involved in the study |
|---|---|
| ☒ | ☐ ChIP-seq |
| ☒ | ☐ Flow cytometry |
| ☒ | ☐ MRI-based neuroimaging |

## Antibodies

| | |
|---|---|
| Antibodies used | ANTI-FLAG® M2-Peroxidase (HRP) antibody (Sigma-Aldrich, cat. No.A8592) 1:5000<br>anti-streptavidin-HRP RABHRP3(Sigma-Aldrich) 1:10000<br>anti-GFP (Roche) cat. No. 11814460001 5.5 ng/uL<br>anti-his-HRP HRP-66005 (Proteintech) 1:1000 |
| Validation | ANTI-FLAG antibodies were validated including negative controls that did not express the protein of interest (such negative controls are included in Fig. 2b and Extended Data Fig. 2e.). Anti-GFP antibodies were validated in doi: 10.1016/j.molp.2019.01.007. Anti-streptavidin-HRP and anti-his-HRP antibodies were used to detect purified proteins. Anti-streptavidin-HRP has been validated for detection of BioDiaAlk (doi.org/10.1038/s41589-018-0116-2). Anti-his-HRP has been validated against non-his-tagged AtPCO4 recombinant protein. |

## Eukaryotic cell lines

Policy information about cell lines and Sex and Gender in Research

| | |
|---|---|
| Cell line source(s) | Saccharomyces cerevisiae strain BY4742 (Matα; his3-Δ1; leu2-Δ0; lys2-Δ0; ura3-Δ0) |
| Authentication | Authentication based on auxotrophies |
| Mycoplasma contamination | We did not test the cells for mycoplasma contamination |
| Commonly misidentified lines (See ICLAC register) | N/A |

## Dual use research of concern

Policy information about dual use research of concern

### Hazards

Could the accidental, deliberate or reckless misuse of agents or technologies generated in the work, or the application of information presented in the manuscript, pose a threat to:

| No | Yes | |
|---|---|---|
| ☒ | ☐ | Public health |
| ☒ | ☐ | National security |
| ☒ | ☐ | Crops and/or livestock |
| ☒ | ☐ | Ecosystems |
| ☒ | ☐ | Any other significant area |

## Experiments of concern

Does the work involve any of these experiments of concern:

| No | Yes | |
|----|-----|---|
| ☒ | ☐ | Demonstrate how to render a vaccine ineffective |
| ☒ | ☐ | Confer resistance to therapeutically useful antibiotics or antiviral agents |
| ☒ | ☐ | Enhance the virulence of a pathogen or render a nonpathogen virulent |
| ☒ | ☐ | Increase transmissibility of a pathogen |
| ☒ | ☐ | Alter the host range of a pathogen |
| ☒ | ☐ | Enable evasion of diagnostic/detection modalities |
| ☒ | ☐ | Enable the weaponization of a biological agent or toxin |
| ☒ | ☐ | Any other potentially harmful combination of experiments and agents |

## Plants

| | |
|---|---|
| Seed stocks | A. thaliana Columbia-0 (Col-0) CS70000 from ABRC; erfVII (rap2.2-1 rap2.3-1 rap2.12-1 hre1 hre2) described in https://doi.org/10.1104/pp.114.244723; prt6-5 is SALK_051088 from NASC; ate1/2 described in https://doi.org/10.1073/pnas.0906404106 pco mutants described in https://doi.org/10.1111/pce.14440. roGFP2-Orp1 described in https://doi.org/10.1111/nph.15550. |
| Novel plant genotypes | New plant genotypes were generated using Agriobacterium mediated transformation as detailed in M&M. |
| Authentication | Genotype autentication was performed by PCR |

