## [Peer Review File · Nature]

H₂O₂ repurposes plant O₂ sensing to regulate post-hypoxia responses

Corresponding Author: Professor Emily Flashman

Version 0:

Reviewer comments:

Referee #1

(Remarks to the Author)

This paper by Akter and Perri et al investigates the molecular cross talk between the plant oxygen sensing machinery, oxygen availability, and ROS bursts, to provide new insights into how transcriptional responses to submergence and de-submergence are coordinated. Submergence is a major abiotic stress that triggers hypoxia, and work in the past 15 years (including important studies by the corresponding/lead authors of this paper) has established a mechanism controlling perception and response to low oxygen levels, mediated by the N-degron pathway of protein destabilisation. Essentially, ethylene response factor group 7 (ERF-VII) transcription factors are substrates for plant cysteine oxidases (PCOs), which trigger ERF-VII degradation in the presence of oxygen. Under hypoxia, PCO activity is inhibited and ERF-VIIs accumulate in the nucleus. Upon de-submergence (reoxygenation) the prevailing model (supported by multiple studies, further discussed below) is that ERF-VIIs are rapidly destabilised, allowing a return to aerobic gene expression.

This paper focusses on this reoxygenation phase of the stress, paying particular attention to the well-documented ROS burst that occurs, and how this influences both PCO and ERF-VII activities. The authors posit that, following de-submergence, this ROS burst inhibits PCO activity, and ERF-VIIs are not in fact turned over and instead remain stabilised (likely in situ) at gene promoters. At the same time, they propose that ERF-VII activity is “altered”, and that these TFs switch from activators to repressors at anaerobic response genes. Both of these major conclusions are of potential interest as they would significantly alter our understanding of hypoxic (and ROS) stress response and recovery. However, I do not believe that these conclusions are fully supported by the experimental data that is presented here .

Although this study involves a lot of work, and much of the data is very interesting and of good quality, there are still many unresolved questions (presented in more detail at relevant points below). Furthermore, the general model - ROS-mediated stabilization of ERFVIs following reoxygenation, and a transcriptional activity switch - is too simplified based on the data presented, and doesn't adequately consider previous studies in the field that it often contradicts, or broader ROS-stress implications. I therefore think that the work is too premature to be published without significant further experiments.

Specific comments, in order of appearance in manuscript

Line 135 – although the authors do provide a caveat to what they say, I do not believe that the results presented in figure 1A-E can differentiate between the effects of reoxygenation stress or the hypoxic stress. A “dead” root meristem would not be apparent without testing post hypoxia re-growth. To check this, authors should test root meristem viability directly at the end of the 24h hypoxia stress (e.g., by Evans blue staining of the root tip). Also, 0.1% hypoxia for 24h is quite a severe stress for seedlings, so I was surprised to see such high levels of viability in Col-0 (Fig. 1c) – can the authors comment on this? An important further experiment here would be to pre-treat seedlings +/- TBHP (or another ROS donoer) and then place into hypoxia and assess survival following reoxygenation. Since TBHP is later shown to stabilise ERF-VIIs (but, counterintuitively, negatively affect HRG expression), it is important to investigate how pre-stabisation of ERF-VIIs with ROS might this affect plant survival of low oxygen levels. This experiment is important, as beyond this initial seedling survival assay in figure 1, everything else in the manuscript is shown at the molecular level only without an assessment at the physiological level.

When looking at ROS burst dynamics Figure 1 F-J, it is clear that there is increased ROS production following

re-oxygenation in the *erfVII* mutant. However, to further interrogate this, the authors should test ROS levels in lines that have constitutively stable ERF-VIIs (e.g. MA-RAP2.12, or *prt6* or *ate1/2*). The authors do cite that in SUB1A rice (where they explain that SUB1A is N-degron pathway resistant,) there is reduced ROS upon de-submergence, so it will be important to actually test this in Arabidopsis with lines that have genetically stabilised ERF-VIIs to further support the model they propose in this study.

Section beginning on Line 161 (Figures 2 and 3) - Here the authors investigate RAP2.12 and RAP2.3 stability in the nucleus under normoxia/hypoxia/re-oxygenation and in the presence or absence of exogenous ROS supplementation (TBHP). They show that TBHP stabilises these ERF-VIIs, and suggest that in response to re-oxygenation under mock conditions, the ERF-VIIs remain stable in the nucleus. The authors state that "Upon re-oxygenation, ERFVII may be presumed to destabilise with the resumption of PCO enzyme function in aerobic conditions, however we found that the RAP2-12-GFP signal persisted in the nucleus after 3 h re-oxygenation treatment". I found this to be quite unexpected, as it contradicts the findings of many previous studies, including several by co-authors of this work. i.e., it is not to be "presumed" that these factors are destabilised in aerobic conditions, it has been shown many times that they are. I was surprised to see that the authors make no mention of this in the manuscript, and I cannot see how the new work can be presented without discussing this in detail. Some specific examples from prior works showing clear destabilisation upon re-oxygenation include:

- In Licausi et al 2011 (PMID: 22020282), authors clearly show disappearance of RAP2-12-GFP (is this the same line as used in this current study?) following re-oxygenation. This does not occur in N-degron mutants (*prt6* and *ate1/2*).
- In Kosmacz et al 2015 (PMID: 25438831), the dynamics upon re-oxygenation are examined in more detail, and RAP2.12-GFP is shown to be completely depleted by 3h re-oxygenation (in contrast to what is shown in this current study). In conjunction, anaerobic gene expression was shown to have returned to pre-stress levels within the same time frame as complete RAP2.12 depletion.
- In Zubrycka et al 2023 (PMID: 37537157), RAP2.3-HA is shown to accumulate strongly under hypoxia and be completely destabilised within just 60 min of re-oxygenation.

Related to figure 2 and S2, the data presented are not quantitative, and based on single confocal images only. Authors should conduct and quantify anti-GFP western blots, including over a re-oxygenation time course. This will allow a more detailed understanding of maintained stability following re-oxygenation (which I believe is needed within the context of previous findings of e.g. Kosmacz et al)

From line 207 – here authors use an artificial fusion of the RAP2.12 N-terminus fused to luciferase to test whether the stability dynamics they report are due to the N-terminus of the ERF-VIIs. By introducing an alanine mutation to prevent PCO-initiated proteolysis, the authors show that this reporter no longer responds to hypoxia or TBHP treatments. Here it would be better for the authors to use a mutant stable RAP2.12-GFP line (which is available) as this is more "natural" full length protein, allowing for better direct comparisons to the data presented in figure 2. Also, to further interrogate the link to the N-degron pathway, it is essential that dynamics of MC-RAP2.12 and MC-RAP2.3 stability in response to hypoxia, TBHP, and combinations thereof, are assessed in N-degron pathway mutants (*prt6* or *ate1/2*). In Licausi et al 2011, these mutants were shown to have consistent nuclear localisation and stability of RAP2.12-GFP under normoxia, hypoxia, and re-oxygenation. Understanding if and how ERF-VII behaviour changes in response to de-submergence or TBHP in these mutants is therefore required to support the findings presented in this current study, particularly since the central conclusion (i.e., maintenance of stable ERF-VIIs upon re-oxygenation) is in apparent contradiction to previous findings. Furthermore, although the experiments in the heterologous yeast system do provide some corroborative data, it is more relevant to study the dynamics in planta, especially given that N-degron mutants etc are readily available.

The data in figure 4, showing H₂O₂ inhibition of PCOs (mainly PCO4) show that this isoform is likely inhibited by ROS due to catalytic inactivation and structural changes. Work from co-authors has previously shown PCO4 to be the most active member of the family. PCO4 however is also one of the constitutively expressed members if I understand correctly. In contrast, PCO1 and PCO2 are both strongly hypoxia inducible (Weits et al 2014 PMID: 24599061), and so it has been previously suggested that their upregulation under hypoxia might be a key component of the hypoxia response (potentially required for ensuring more rapid ERFVII turnover upon re-oxygenation). Interestingly, PCO1 and 2 appear to be the least sensitive to ROS inhibition *in vitro* (Figure 4C). The authors should expand their analysis/discussion with respect to these two family members. For example, is their expression affected by TBHP? How do the dynamics of ERF-VIIs change in response to TBHP/hypoxia/re-oxygenation in the previously reported *pco1pco2* double mutant etc?

The authors present several interesting observations relating to ERF-VII-mediated gene expression in response to ROS. TBHP treated plants in the absence of hypoxia show increased ERF-VII stability, but no change in any of the tested ERF-VII-regulated HRGs. In contrast, ERF-VII regulated ROS-responsive genes are upregulated. Although I found it rather difficult to follow this section (and suggest revisiting the narrative when describing the RNA seq analysis, for clarity), the outcome of the transcriptome analysis indicates that TBHP-stabilised ERF-VIIs repress HRGs. The authors summarise all of this by stating they have identified roles for ERF-VIIs as both activators and repressors of transcription in a gene-selective manner in response to hypoxia and oxidative stress, "providing a potential rationale for their implication in responses to both re-oxygenation and a broader range of biotic and abiotic stresses". This is certainly of potential interest, but at the same time quite confounding. I think this section of the study – that is, the condition dependent switching between activation and repression of HRGs – is still rather premature and requires further work to provide a clearer rationale as to how and why it could be working. I also cannot rationalise in my mind how there could be target-gene-specific switching of a bound ERF-VII from positive to negative regulator, without it being actioned via an additional non-ERF mechanism, e.g. elsewhere at the promoter. In the discussion, the known HRA1 regulator, which counteracts ERF-VII activity, is mentioned – but how would

this target only those ERF-VIIs at HRGs and not at other genes bearing HRPEs?

Other comments and considerations:

Throughout, authors use exogenous TBHP to show that ROS inhibits ERF-VII turnover, suggesting this is due to PCO inactivation. In a natural context, is this specific only to ROS-changes upon hypoxia/reoxygenation, or are they proposing that ERF-VIIs are general mediators of ROS signalling? Authors could test this further by doing e.g., low-to-high light treatments to induce ROS stress, as well as through using alternative ROS donors (this whole study is TBHP-reliant only), and then assessing ERF-VII stability and target gene expression patterns. I think this is important, especially given the title of the study.

Did the authors consider using H₂O₂ scavengers to further interrogate their hypothesis? E.g., How are ERF-VII stability dynamics, and HRG expression profiles, affected upon reoxygenation in the presence of a ROS scavenger? If their model is correct, this should limit PCO inhibition during the recovery phase, and have concomitant effects on ERF-VII stability, target gene expression and seedling survival.

As mentioned above, it is important to further disentangle the ROS vs hypoxia responses. Authors should consider investigating ERF-VII stability and HRG expression dynamics in mutants and overexpressor lines that have altered endogenous ROS levels. E.g., how do they behave in catalase mutants or overexpressors that have increased vs reduced H₂O₂ respectively?

Do the authors think that ROS inactivation of PCOs is reversible? At some point the ERF-VIIs must need to be turned over following de-submergence and/or following the alleviation of elevated ROS. How would this happen? There is some discussion of antioxidants being unable to repair H₂O₂-inhibited PCOs in vitro, but this should be further expanded on.

A previous study in mammals (PMID: 34893540) has looked at crosstalk between oxygen and ROS within the context of N-degron pathway substrate turnover; specifically, RGS4. In this paper, it is shown that the N-terminal cysteine of RGS4 is a dual sensor for both oxygen and oxidative stress, and that this is linked to the duration of hypoxia and can result in targeting to different degradation pathways. Have authors considered how and if ROS might directly affect the N-degron of ERF-VIIs? This mammalian study should be more directly discussed in this work, as it could also have direct implications for the situation in plants given the broad conservation of the N-degron pathway in these two different lineages.

Referee #2

(Remarks to the Author)

This manuscript provides interesting new evidence related to the interactions between redox and oxygen sensing in *Arabidopsis thaliana*. This manuscript directly addresses a key question related to role of the burst of reactive oxygen species (ROS) that is observed in plant tissues following a period of hypoxia or anoxia. Convincing evidence is presented showing that ERFVII transcription factors remain stable and localised in plant nuclei as a result of ROS-mediated PCO-inactivation and that ERFVII activate expression of genes associated with oxidative stress responses. The process of recovery from reoxygenation was compared in seven-day old *Arabidopsis* wild type and erfVII mutant seedlings. While the data overall are convincing, the measurements appear to have been performed over different timescales during the hypoxia/reoxygenation treatment. For example, while no immediate differences were observed decreased survival was observed in the erfVII seedlings after 4 days of reoxygenation following hypoxia. However, ROS staining procedures revealed greater hydrogen peroxide accumulation (as shown by increased DAB staining) in the erfVII seedlings compared to wild type after 1 h. However, data using the roGFP2-Orp1 probe showed that sensor oxidation upon reoxygenation which was significantly greater in erfVII mutant than Col-0 plants, but this increase was only significantly greater at the later stages of severe hypoxia. Such differences make it hard to align the different observations. Further analysis of the contribution of ROS to PCO-ERFVII mediated responses to hypoxia and reoxygenation showed that ERFVII remain stable and localised in plant nuclei and that this finding is linked to ROS-mediated PCO-inactivation. The data shown in some of the figures such as Figure 2 is hard to judge because the figures are so small. It might aid clarity and easier to visualise the intracellular localisation of the GFP/stains, if key parts of the figures were enlarged. It is not clear why the authors used tert-butyl hydroperoxide was used to induce cysteine oxidation. Did the authors try other oxidants to confirm the observed effects?

Referee #3

(Remarks to the Author)

This study provides valuable information to help us understand how ERF-VII proteins contribute to increased tolerance to reoxygenation, even in the presence of oxygen. It has been recognized that master regulators of submergence and hypoxia tolerance, ERF-VIIs, play an essential role in the adaptation to reoxygenation, but it was unclear how they work because these proteins are degraded in the presence of oxygen. The data presented in this study show a potential mechanism for

stabilizing ERF-VII proteins by hydrogen peroxide. My comments are as follows.

1) Figure 1 shows that ERF-VIIs suppress ROS accumulation under reoxygenation. Figures 2 and 3 indicate that hydrogen peroxide regulates the localization of RAP2.12 to the nucleus and the stabilization of RAP2.3. I was wondering whether the level of hydrogen peroxide suppressed by ERF-VIIs in wild-type plants was sufficient to localize and stabilize these proteins under reoxygenation. I agree that the exogenous application of hydrogen peroxide has these effects. However, other than hydrogen peroxide, another factor may be responsible for the localization and stabilization of ERF-VIIs under reoxygenation because only a minimal amount of hydrogen peroxide is accumulated in reoxygenated samples. The use of antioxidants and ROS scavengers in reoxygenated plants/samples must help evaluate the role of hydrogen peroxide.

2) Reduced activity of PCO4 by hydrogen peroxide is a key mechanism to explain the stabilization of ERF-VII. However, this data is generated by in vitro assays using recombinant AtPCO4. Is there any way to assay PCO activity using plant extracts? It is unclear if PCO activity is also reduced in reoxygenated plants as observed in the in vitro system.

3) Figure 5C shows ChIP-qPCR results, but the difference between test samples and negative controls is not evaluated by a statistical method.

4) It is exciting that ERF-VIIs physically interact with the promoter regions of both hypoxia- and ROS-responsive genes under reoxygenation, but only the latter is induced. Identifying the regulatory factors involved in this mechanism significantly increases the quality of this paper. This study shows a possible mechanism for stabilizing ERF-VIIs in reoxygenated plants, which is valuable information to advance our understanding of the molecular basis of hypoxia-reoxygenation tolerance in plants. However, more critical information (e.g., the mechanisms underlying selective expression of hypoxia- and ROS-responsive genes) may need to be considered for publication in Nature.

Version 1:

Reviewer comments:

Referee #1

(Remarks to the Author)

The authors have done a considerable amount of work to respond to the specific points I raised following initial submission of the manuscript. Based on the additional experimental data and associated additions to the text, I believe the manuscript to be significantly improved, and that there is now more robust support for the main findings of the study. A minor remaining suggestion is that the title of the report could be revised to more accurately reflect the nature of the work. As it stands, the title implies a broad role for ERF-VIIs in regulating responses to oxidative stress, yet the study really only supports this to be the case for context-specific oxidative stress that occurs following hypoxia or de-submergence.

Referee #2

(Remarks to the Author)

Oxygen is an immensely versatile molecule that fulfills crucial life functions underpinning cell metabolism. The roles of Plant Cysteine Oxidases (PCOs) and group VII Ethylene Response Factors (ERFVIs) in hypoxia sensing is well characterized, but to date the relationships between oxygen signalling through the Cys/Arg branch of the N-degron pathway for protein degradation and the redox signalling pathways driven by reactive oxygen species (ROS), have not been characterized. This manuscript addresses this question directly. Although there may be many points of reciprocal control that remain to be elucidated, the data presented in this manuscript point to a novel regulatory control point. The paper provides evidence that PCO functions are modified in the presence of hydrogen peroxide. Data are presented showing that inhibition of PCO activity occurs in the presence of hydrogen peroxide leading to ERFVII stabilisation. Intriguingly, the data indicate that oxidation causes only minor ERFVII-mediated modifications in the expression of hypoxia responsive genes and is suggested to alter ERFVII function from positive to negative regulation of gene transcription.

The inhibition of PCO activity appears to occur by a combination of oxidation of the active site Fe(II) to Fe(III), and also oxidation of Cys residues in or near the active site. This finding would also suggest that superoxide might also catalyse PCO inhibition through oxidation of the active site Fe(II) to Fe(III). Conversely, only hydrogen peroxide can cause Cys oxidation. Hence, superoxide and hydrogen peroxide might modify PCO functions by mechanism that could lead to slightly different outcomes, for example in terms of protein/protein interactions.

Referee #3

(Remarks to the Author)

I have re-evaluated the revised manuscript, including the rebuttal and newly added data/figures. The revision is improved and addresses the main issues I raised in the first round, particularly by strengthening the causal link between ROS/H₂O₂ and ERF-VII behavior during reoxygenation/oxidative stress, and by improving statistical support for key chromatin-binding conclusions. The remaining concerns are minor and largely relate to framing and clarity.

Comments on revisions

(1) My original concern was whether ROS/H₂O₂ is causal under the relevant reoxygenation context, rather than an artefact of exogenous oxidant treatments. The authors now add antioxidant/scavenger experiments (ascorbate), showing restricted

RAP2.3 reporter accumulation after TBHP and during post-hypoxia reoxygenation, with corresponding effects on a subset of transcriptional outputs. This meaningfully strengthens the causal interpretation and addresses my main request. Minor revision: add 1–2 sentences in the Discussion noting that antioxidant treatments can be pleiotropic, while emphasizing that the consistent directionality across reporter/protein and transcriptional readouts supports a ROS-dependent mechanism.

(2) I previously asked whether reduced PCO activity could be demonstrated directly in vivo (e.g., in plant extracts). The authors indicate that a direct in planta enzymatic assay is not currently available and thus rely on recombinant PCO biochemistry together with in vivo functional readouts (substrate stability/localization/transcription). This limitation is acceptable provided it is framed clearly.

Minor revision: explicitly state (if not already) that direct quantification of PCO catalytic activity in planta remains technically challenging/unavailable, and clarify that the conclusion is supported by recombinant PCO biochemistry plus strong in vivo consequences.

(3) I requested statistical testing to support the claim that ChIP–qPCR signals exceed negative controls. The authors have added statistical analysis (with reporting in supplementary materials/legend), which resolves this point.

(4) I also asked for clearer mechanistic insight into how ERF-VIIs can associate with promoters of hypoxia genes yet show repression while enabling oxidative stress programs during reoxygenation. The revision provides progress by adding new data and strengthening the narrative, including domain-level evidence that specific RAP2.12 C-terminal motifs contribute to ROS-dependent transcriptional behavior. While the precise determinants of promoter selectivity/cofactor usage remain incomplete, the added mechanistic detail improves the study.

Minor revision: briefly clarify in the Discussion that promoter selectivity/cofactor mechanisms during reoxygenation remain to be defined, framing the new domain-level data as important progress while acknowledging this as a future direction.

Ref. Nature manuscript 2024-11-25256

Dear Editor,

Thank you very much for your consideration of our manuscript 'H₂O₂ repurposes the plant oxygen-sensing machinery to control the transcriptional response to oxidative stress'. We are very grateful for all of the Reviewers' positive and constructive comments. In light of the criticisms and suggestions provided, we have added further experimental results and addressed all the points raised, resulting in what we believe to be an improved and robust manuscript.

We have uploaded our revised manuscript as well as the original with major changes tracked for easy reference. A detailed description of the changes we made in light of the Reviewers' suggestions is given below, as well as specific responses to the Reviewers' concerns.

Reviewer 1

This paper by Akter and Perri et al investigates the molecular cross talk between the plant oxygen sensing machinery, oxygen availability, and ROS bursts, to provide new insights into how transcriptional responses to submergence and de-submergence are coordinated. Submergence is a major abiotic stress that triggers hypoxia, and work in the past 15 years (including important studies by the corresponding/lead authors of this paper) has established a mechanism controlling perception and response to low oxygen levels, mediated by the N-degron pathway of protein destabilisation. Essentially, ethylene response factor group 7 (ERF-VII) transcription factors are substrates for plant cysteine oxidases (PCOs), which trigger ERF-VII degradation in the presence of oxygen. Under hypoxia, PCO activity is inhibited and ERF-VIIs accumulate in the nucleus. Upon de-submergence (reoxygenation) the prevailing model (supported by multiple studies, further discussed below) is that ERF-VIIs are rapidly destabilised, allowing a return to aerobic gene expression.

This paper focusses on this reoxygenation phase of the stress, paying particular attention to the well-documented ROS burst that occurs, and how this influences both PCO and ERF-VII activities. The authors posit that, following de-submergence, this ROS burst inhibits PCO activity, and ERF-VIIs are not in fact turned over and instead remain stabilised (likely in situ) at gene promoters. At the same time, they propose that ERF-VII activity is "altered", and that these TFs switch from activators to repressors at anaerobic response genes. Both of these major conclusions are of potential interest as they would significantly alter our understanding of hypoxic (and ROS) stress response and recovery. However, I do not believe that these conclusions are fully supported by the experimental data that is presented here .

Although this study involves a lot of work, and much of the data is very interesting and of good quality, there are still many unresolved questions (presented in more detail at relevant points below). Furthermore, the general model - ROS-mediated stabilization of ERFVIs following reoxygenation, and a transcriptional activity switch - is too simplified based on the data presented, and doesn't adequately consider previous studies in the field that it often contradicts, or broader ROS-stress implications. I therefore think that the work is too premature to be published without significant further experiments.

We thank Reviewer 1 for their careful reading of the manuscript and for its scrutiny in the context of prior literature. We were indeed aware that some of our data appears to contradict previous

published studies in the field and recognise that we should have been presented our data in this context more robustly. We have conducted further experiments (detailed below) and addressed our findings relative to other work more carefully in the text. With the additional work and discussion we include here, we would respectfully defend the importance of our findings and the maturity of the study. Indeed, the paradigm-changing nature of our findings (that PCO-ERFVII respond not only to hypoxia, but also to ROS in order to tailor plant responses to both stresses experienced upon submergence) necessarily opens up new questions in this research field. We address the Reviewer's specific points carefully below:

Specific comments, in order of appearance in manuscript:

1) Line 135 – although the authors do provide a caveat to what they say, I do not believe that the results presented in figure 1A-E of can differentiate between the effects of reoxygenation stress or the hypoxic stress . A “dead” root meristem would not be apparent without testing post hypoxia re-growth. To check this, authors should test root meristem viability directly at the end of the 24h hypoxia stress (e.g., by Evans blue staining of the root tip).

The Reviewer is correct, with the data we presented in our initial draft, it is not possible to distinguish between the results of the hypoxic stress treatment and the subsequent reoxygenation stress. We have therefore repeated the experiments, this time coupling Evans blue staining with post-hypoxia survival assays as suggested, to examine root viability following hypoxic and reoxygenation treatment (**Extended Data Fig. 1f,g**). We categorised roots as alive, damaged or dead based on tissue staining, and found that following hypoxia treatment only, less than 25 % of roots were dead for both the Col-0 and *erfVII* plants, with no significant difference between these two genotypes. Following reoxygenation however, the proportion of dead plants was much higher, at >60 % for Col-0 and >80 % for *erfVII* plants, consistent with root tip death being related to reoxygenation stress. The difference between *erfVII* and Col-0 plants is significant (<0.0001), supportive of a role for ERFVII in post-hypoxia survival and consistent with the impaired root growth observed (**Extended Data Fig. 1d**). We comment on this in lines 94-97 of the revised manuscript.

2) Also, 0.1% hypoxia for 24h is quite a severe stress for seedlings, so I was surprised to see such high levels of viability in Col-0 (Fig. 1c) – can the authors comment on this?

We are aware that Arabidopsis tolerance to hypoxia varies greatly among labs. This is probably dependent on the growth conditions, including light intensity, photoperiod and starting time for the hypoxic treatment. Since submerged Arabidopsis roots in the dark are reported to have very low O₂ concentrations, near anoxic at 0.3 kPa (Vashist D et al., 2010), we wanted to mimic this scenario in our hypoxic chamber. Therefore, we elected to treat seeds with 0.1 % O₂. The viability assays we include in the revised manuscript (**Extended Data Fig. 1f,g**) indeed demonstrate that a low proportion of either Col-0 or *erfVII* seedlings were dead following this treatment (lines 94-97 in the revised manuscript). This is consistent with an extensive investigation of the time to lethality in a range of Arabidopsis successions subjected to dark submergence in soil, reported by Vashist et al., including Col-0 which had a median lethal time of 8 days.

3) An important further experiment here would be to pre-treat seedlings +/- TBHP (or another ROS donor) and then place into hypoxia and assess survival following reoxygenation. Since TBHP is later shown to stabilise ERF-VIIs (but, counterintuitively, negatively affect HRG expression), it is

important to investigate how pre-stabilisation of ERF-VIIs with ROS might this affect plant survival of low oxygen levels. This experiment is important, as beyond this initial seedling survival assay in figure 1, everything else in the manuscript is shown at the molecular level only without an assessment at the physiological level.

We thank the Reviewer for this suggested experiment. We treated and analysed seedlings exactly as we had in the experiments shown in **Fig. 1b-e**, but pre-treated them +/- 1 mM TBHP, for 2 h prior hypoxic or air treatment. We found that TBHP pre-treatment improved plant tolerance following hypoxia and reoxygenation in the wild type but not in the *erfVII* mutant (**Fig. 2c** and **Extended Data Fig. 2f,g**). This improvement in tolerance, small and yet significant, highlights the importance of ERFVII-mediated induction of oxidative stress-mitigating genes. We comment on this in lines 132-136 of the revised manuscript.

4) When looking at ROS burst dynamics Figure 1 F-J, it is clear that there is increased ROS production following reoxygenation in the erfVII mutant. However, to further interrogate this, the authors should test ROS levels in lines that have constitutively stable ERF-VIIs (e.g. MA-RAP2.12, or prt6 or ate1/2). The authors do cite that in SUB1A rice (where they explain that SUB1A is N-degron pathway resistant,) there is reduced ROS upon de-submergence, so it will be important to actually test this in Arabidopsis with lines that have genetically stabilised ERF-VIIs to further support the model they propose in this study.

We thank the Reviewer for this suggested experiment. We have repeated our experiment investigating H₂O₂ dynamics using the roGFP-Orp1 fluorescent biosensor in *prt6* mutant Arabidopsis seedlings, which have stabilised ERFVII due to their impaired degradation via the N-degron pathway (Gibbs et al. 2011, Licausi et al. 2011). Consistent with a role for ERFVII in scavenging H₂O₂, we observed decreased accumulation of H₂O₂ in these lines compared with Col-0 (at 0.1 % O₂, where biosensor oxidation is observed in Col-0), in contrast to biosensor fluorescence in *erfVII* seedlings. These new data have been added to **Fig. 1h-j** and referred to in the text in lines 104-113 of the revised manuscript.

5) Section beginning on Line 161 (Figures 2 and 3) - Here the authors investigate RAP2.12 and RAP2.3 stability in the nucleus under normoxia/hypoxia/re-oxygenation and in the presence or absence of exogenous ROS supplementation (TBHP). They show that TBHP stabilises these ERF-VIIs, and suggest that in response to reoxygenation under mock conditions, the ERF-VIIs remain stable in the nucleus. The authors state that "Upon reoxygenation, ERFVII may be presumed to destabilise with the resumption of PCO enzyme function in aerobic conditions, however we found that the RAP2-12-GFP signal persisted in the nucleus after 3 h reoxygenation treatment". I found this to be quite unexpected, as it contradicts the findings of many previous studies, including several by co-authors of this work. i.e., it is not to be "presumed" that these factors are destabilised in aerobic conditions, it has been shown many times that they are. I was surprised to see that the authors make no mention of this in the manuscript, and I cannot see how the new work can be presented without discussing this in detail. Some specific examples from prior works showing clear destabilisation upon reoxygenation include:

- In Licausi et al 2011 (PMID: 22020282), authors clearly show disappearance of RAP2-12-GFP (is this the same line as used in this current study?) following reoxygenation. This does not occur in N-degron mutants (*prt6* and *ate1/2*).*
- In Kosmacz et al 2015 (PMID: 25438831), the dynamics upon re-oxygenation are examined in more detail, and RAP2.12-GFP is shown to be completely depleted by 3h reoxygenation (in*

contrast to what is shown in this current study). In conjunction, anaerobic gene expression was shown to have returned to pre-stress levels within the same time frame as complete RAP2.12 depletion.

- In Zubrycka et al 2023 (PMID: 37537157), RAP2.3-HA is shown to accumulate strongly under hypoxia and be completely destabilised within just 60 min of reoxygenation.*

We thank the Reviewer again for their careful scrutiny of the manuscript in the context of prior literature and apologise that we did not offer a robust justification for our results in our first submission. We nevertheless assert that our findings are consistent with prior literature, however the conditions for each experiment are important, particularly in the context of the concentration of O₂ and the duration of this treatment in each case:

In Licausi et al. 2011 (PMID 22020282), authors did indeed show disappearance of RAP2.12-GFP following 1 hour reoxygenation after 90 minutes hypoxia at 1 % O₂. In Kosmacz et al. 2015 (PMID 25438831) RAP2.12-GFP was indeed shown to destabilise after a return to normoxia following **3 h hypoxia at 1 % O₂**, however it should be noted that RAP2.12-GFP levels only started to decline at 3 hours post-hypoxia, while hypoxia-responsive gene (HRG) expression declined earlier than this. This trend is consistent with the outcomes of our study. This work was also supported by a more recent report by Brunello et al. 2025 (PMID 39704305). In Zubrycka et al. 2023 (PMID 37537157), RAP2.3^{3xHA} was indeed stabilised in severe (<0.5 % O₂) hypoxia then degraded after 1 hour of reoxygenation, however the duration of hypoxia treatment was only 1 hour.

It is therefore possible that the rate of ERFVII degradation may be dependent on the severity and duration of hypoxia, as well as the production of ROS upon reoxygenation. In combination with our cyt-roGFP-Orp1 oxidation measurements, which indicate that ROS levels increase with duration of hypoxia and are greater in 0.1 % O₂ than in 1 % O₂ (**Fig. 1h,i**) this would suggest that the accumulation of ROS (during hypoxia and/or upon reoxygenation) may impact the duration of ERFVII stability. This would explain the apparent inconsistencies between our data and data published in the literature, as well as between different reports in the literature. We have addressed this important point in the main text, lines 127-131.

6) Related to figure 2 and S2, the data presented are not quantitative, and based on single confocal images only. Authors should conduct and quantify anti-GFP western blots, including over a reoxygenation time course . This will allow a more detailed understanding of maintained stability following re-oxygenation (which I believe is needed within the context of previous findings of e.g. Kosmacz et al)

We believe the confocal images are important to demonstrate the nuclear retention of the RAP2.12 and RAP2.3 post-hypoxia, however agree that a quantitative and time-dependent approach would be supportive. We have found the generation of full-length fusions of RAP2.12 or RAP2.3 with C-terminal tags that can be detected via Western blot to be challenging, however prepared an internally FLAG-tagged form of RAP2.12 which worked well. We generated two independent lines expressing this construct (**Extended Data Fig. 2d**). We used seven-day old *A. thaliana* seedlings exposed to the same treatment as those examined by confocal microscopy (21 % or 1 % O₂ for 6 hours, followed by 1 and 3 h reoxygenation) and quantified the amount of each using anti-FLAG antibodies and Western blot analysis (**Fig. 2b** and **Extended Data Fig. 2e**). Consistent with the nuclear localisation of RAP2.12 and RAP2.3, we observed FLAG-tagged RAP2.12 signal even after reoxygenation, with intensity of signal declining between 1 and 3 hours. We also used a full-length RAP2.3-nanoluciferase construct to quantify protein stability through luciferase activity. In the original submitted manuscript, this was a Supplementary Figure (with similar experiments conducted using a construct comprising the RAP2.12 N-terminus fused to

luciferase presented in the main text). Given the importance of clarifying the duration of the stability of full length ERFVII proteins conferred by TBHP and reoxygenation, we have now rearranged the manuscript so that the description of the full length RAP2.3nLuc experiments (**Fig. 3**) follows the confocal data and Western blot data that reveal prolonged ERFVII stabilisation following reoxygenation (page 4, lines 125-127, and page 5, lines 138-140). The data showing the RAP2.12₁₋₂₈FLuc line are now in the Supplementary Information (**Extended Data Fig. 3h**).

7) From line 207 – here authors use an artificial fusion of the RAP2.12 N-terminus fused to luciferase to test whether the stability dynamics they report are due to the N-terminus of the ERF-VIIs. By introducing an alanine mutation to prevent PCO-initiated proteolysis, the authors show that this reporter no longer responds to hypoxia or TBHP treatments. Here it would be better for the authors to use a mutant stable RAP2.12-GFP line (which is available) as this is more “natural” full length protein, allowing for better direct comparisons to the data presented in figure 2. Also, to further interrogate the link to the N-degron pathway, it is essential that dynamics of MC-RAP2.12 and MC-RAP2.3 stability in response to hypoxia, TBHP, and combinations thereof, are assessed in N-degron pathway mutants (prt6 or ate1/2). In Licausi et al 2011, these mutants were shown to have consistent nuclear localisation and stability of RAP2.12-GFP under normoxia, hypoxia, and reoxygenation. Understanding if and how ERF-VII behaviour changes in response to de-submergence or TBHP in these mutants is therefore required to support the findings presented in this current study, particularly since the central conclusion (i.e., maintenance of stable ERF-VIIs upon reoxygenation) is in apparent contradiction to previous findings. Furthermore, although the experiments in the heterologous yeast system do provide some corroborative data, it is more relevant to study the dynamics in planta, especially given that N-degron mutants etc are readily available.

We agree that these experiments are important. We would again highlight that, as well as conducting stability experiments with the RAP2.12 N-terminus fused to luciferase, we did indeed use a more ‘natural’ full length ERFVII, RAP2.3 fused to nanoluciferase. Indeed, we had already conducted a preliminary assessment of N-degron pathway involvement using the O₂-independent C2A-RAP2.3-nLuc line (**Fig. 3d**). To expand on this as per the Reviewer’s suggestion, we have now generated additional transgenic lines expressing RAP2.3-nLUC in the *ate1/2* and *prt6* mutant backgrounds and assessed protein stability under oxidative stress using luciferase assays. Consistent with the Cys-to-Ala mutant, these lines showed no further stabilization upon TBHP treatment, in line with an N-degron pathway-mediated effect. These new results have been incorporated into **Extended Data Fig. 3j** and are discussed in lines 166-169 of the revised manuscript. Although we agree that the most valuable data are those derived from plant models, the yeast experiment is advantageous in allowing us to examine a system devoid of an endogenous thiol dioxygenase (PCO) enzyme. Thus, it allows us to compare C-DLOR, D-DLOR and R-DLOR systems to verify that the TBHP-mediated effect on Fluc activity involves an Nt-Cys-specific function.

8) The data in figure 4, showing H₂O₂ inhibition of PCOs (mainly PCO4) show that this isoform is likely inhibited by ROS due to catalytic inactivation and structural changes. Work from co-authors has previously shown PCO4 to be the most active member of the family. PCO4 however is also one of the constitutively expressed members if I understand correctly. In contrast, PCO1 and PCO2 are both strongly hypoxia inducible (Weits et al 2014 PMID: 24599061), and so it has been previously suggested that their upregulation under hypoxia might be a key component of the hypoxia response (potentially required for ensuring more rapid ERFVII turnover upon reoxygenation). Interestingly, PCO1 and 2 appear to be the least sensitive to ROS inhibition in vitro (Figure 4C). The authors should expand their analysis/discussion with respect to these two family

members. For example, is their expression affected by TBHP? How do the dynamics of ERF-VIIs change in response to TBHP/hypoxia/reoxygenation in the previously reported pco1pco2 double mutant etc?

We thank the reviewer for this suggestion. The RNA-seq experiment to explore the transcriptome reprogramming caused by TBHP treatment did not reveal significant changes in expression of PCOs (**Supplemental Table 2** and also below). We also generated additional transgenic lines that express the RAP2.3-nLuc reporter in the *pco1pco2* and *pco4pco5* backgrounds. [REDACTED]

[REDACTED] Indeed, although single treatments of 10 μ M AtPCO1/2 with 5 and 10 μ M H₂O₂ suggested PCO4 was more susceptible to inhibition, IC50 determination for PCO1 and PCO2 revealed only minor differences in susceptibility compared with PCO4 (12.51 μ M, 4.18 μ M and 8.36 μ M for PCOs 1, 2 and 4, respectively, where enzyme concentrations were 2 μ M). Therefore, although absolute quantities of PCO1/2 are likely to increase during hypoxia, these enzymes are still susceptible to H₂O₂ inhibition and even any fraction of these enzymes that are not inhibited by H₂O₂ don't appear to be able to promote ERFVII decay on reoxygenation (based on nLuc activity).

[REDACTED]

9) *The authors present several interesting observations relating to ERF-VII-mediated gene expression in response to ROS. TBHP treated plants in the absence of hypoxia show increased ERF-VII stability, but no change in any of the tested ERF-VII-regulated HRGs. In contrast, ERF-VII regulated ROS-responsive genes are upregulated. Although I found it rather difficult to follow this section (and suggest revisiting the narrative when describing the RNA seq analysis, for clarity), the outcome of the transcriptome analysis indicates that TBHP-stabilised ERF-VIIs repress HRGs. The authors summarise all of this by stating they have identified roles for ERF-VIIs as both activators and repressors of transcription in a gene-selective manner in response to hypoxia and oxidative stress, “providing a potential rationale for their implication in responses to both reoxygenation and a broader range of biotic and abiotic stresses”. This is certainly of potential interest, but at the same time quite confounding. I think this section of the study – that is, the condition dependent switching between activation and repression of HRGs – is still rather*

premature and requires further work to provide a clearer rationale as to how and why it could be working.

In response to the reviewer's exhortation, we have tested multiple plausible hypotheses, including those generated by studies published after the pre-print version of this manuscript was made available online on bioRxiv. First, we used an independent *hra1-1* T-DNA line (Giuntoli et al. 2014, PMID 25226037) to exclude the involvement of HRA1 in repressing HRGs in reoxygenation and oxidative stress (**Extended Data Fig. 7a**). This is commented at lines 268-271 of the revised manuscript. Indeed, we also reasoned that it is unlikely that the products of HRGs are responsible for the regulatory mechanism described in our manuscript as they are not induced in case of TBHP treatment.

We also explored the possible role of the ERF transcription factor ORA59 in repressing HRGs, as this was recently proposed to control a subset of these genes (Brunello et al., 2024, PMID: 39704305). In our experimental conditions, a *ora59-1* T-DNA knock out line (Pré et al., 2008, PMID 34890461) and a RNAi-ORA59 silenced line (Yang et al., 2021, PMID 18467450) did not show significant differences in HRG expression with the wild type after 1 h reoxygenation (Licausi F., a separate manuscript is currently in preparation with these data).

A third approach has provided us with more informative results. We generated new transgenic Arabidopsis lines that (over)express truncation versions of RAP2.12 in a *erfVII* knock out background. Here, the RAP2.12 CDS was progressively shortened from the C-terminus to remove common motifs (CMVIIs) previously identified as relevant for transcription regulation activity (Bui et al. 2015, 10.1016/j.plantsci.2015.03.008, Schippers et al. 2024, 10.1111/tpj.17018, Morffy et al., 2024, PMID 39020176). RAP2.12 truncation versions that lack CMVII-8 lost their ability to repress HRGs (**Fig. 5d-f** and **Extended Data Fig. 7b,c**). These results are presented at lines 273-292 and further discussed at lines 391-401.

10) I also cannot rationalise in my mind how there could be target-gene-specific switching of a bound ERF-VII from positive to negative regulator, without it being actioned via an additional non-ERF mechanism, e.g. elsewhere at the promoter. In the discussion, the known HRA1 regulator, which counteracts ERF-VII activity, is mentioned – but how would this target only those ERF-VIIs at HRGs and not at other genes bearing HRPEs?

Although we now demonstrated that HRA1 is not involved in repressing HRGs expression in reoxygenation and oxidative stress (lines 268-271 and response to point (9) above), we agree with the reviewer that such a dichotomic regulation on different genomic targets is at the same time extremely puzzling and intriguing. It is not uncommon that master TF bind different DNA elements and regulate them differently. For example, DNA binding of the WUSCHEL homeodomain was shown to depend on appropriately arranged sequence motifs. In addition to this, homodimerization was demonstrated to be one of the key determinants to achieve high sequence specificity. Recently, the potential for ERFVII dimerisation or ERFVII-ERF interaction was also demonstrated (Kim et al., 2018, 10.3389/fpls.2018.01675 and Yang et al., 2025, 10.1111/nph.70230).

Our RNA-seq analysis of ERFVII-mediated activation of gene expression in response to reoxygenation and oxidative stress revealed a number of genes involved in cell redox homeostasis that are not regulated under hypoxia and do not have an HRPE in their promoter (**Fig. 6a-d** and **Extended Data Fig. 8a-d**). We put forward two non-mutually exclusive hypotheses to explain this observation. According to the first hypothesis, in the absence of additional transcriptional regulators localised on hypoxia-responsive (HRPE-containing) promoters via cognate DNA motifs, ERFVIIs behave as repressors under oxidative stress conditions. On other promoters, additional factors instead partner with the ERFVIIs to promote euchromatin and RNA-polymerase II

recruitment, preventing them from turning into repressors. A second hypothesis might explain recruitment of ERFVIIIs at new targets under reoxygenation and oxidative stress as a redox-dependent change in DNA affinity for unbound ERFVIIIs. We are committed to test these hypotheses, which require substantial additional work (including the generation of additional transgenic lines) and we hope we will be able to submit a second manuscript on this topic in the near future.

On a different note, we realised that in the previous version of our manuscript, we used an HRPE reporter line that included an *ADH1* 5' UTR (Panicucci et al. 2019, [10.3390/bios10120197](https://doi.org/10.3390/bios10120197)). This sequence might bear elements responsible for gene repression in TBHP/reoxygenation. Therefore, we repeated the experiment using another reporter line (also described in Panicucci et al. 2019, [10.3390/bios10120197](https://doi.org/10.3390/bios10120197)) where the *ADH1* 5'UTR is substituted by the omega leader region of the 35S promoter. We did not observe substantial differences with the previous experiment, indicating that the HRPE is sufficient to impart switchable transcriptional regulation to downstream genes (**Fig. 5b**).

Other comments and considerations:

i) Throughout, authors use exogenous TBHP to show that ROS inhibits ERF-VII turnover, suggesting this is due to PCO inactivation. In a natural context, is this specific only to ROS-changes upon hypoxia/reoxygenation, or are they proposing that ERF-VIIs are general mediators of ROS signalling? Authors could test this further by doing e.g., low-to-high light treatments to induce ROS stress, as well as through using alternative ROS donors (this whole study is TBHP-reliant only), and then assessing ERF-VII stability and target gene expression patterns. I think this is important, especially given the title of the study.

We thank the Reviewer for this suggestion. We have tested the effects of other ROS inducers, as well as promoting chloroplast and mitochondrial ROS generation via diuron and antimycin A. In addition to TBHP, RAP2.3-nLuc activity could be increased by both antimycin A and diuron (**Extended Data Fig. 3c,d**), but not by other ROS-inducers including arsenite, cadmium, high light or methyl viologen (**Extended Data Fig. 3e-g**), suggesting that ERFVII stabilisation is induced selectively by specific ROS signals. These data are discussed in the text at lines 152-155. While high light treatment did not increase RAP2.3-nLuc activity, it has been shown that light can stimulate ERFVII protein destabilisation (Abbas et al., 2015, [10.1016/j.cub.2015.03.060](https://doi.org/10.1016/j.cub.2015.03.060)), therefore it is possible that photoreceptor signalling counteracts the ROS-mediated stabilisation under high light intensity.

ii) Did the authors consider using H2O2 scavengers to further interrogate their hypothesis? E.g., How are ERF-VII stability dynamics, and HRG expression profiles, affected upon reoxygenation in the presence of a ROS scavenger? If their model is correct, this should limit PCO inhibition during the recovery phase, and have concomitant effects on ERF-VII stability, target gene expression and seedling survival.

We are grateful to the reviewer for suggesting this experiment, which we carried out. We found that pretreatment with the ROS scavenger ascorbate (10 mM) could partially suppress TBHP-induced RAP2.3 accumulation (**Extended Data Fig. 3a**) and restrict RAP2.3 accumulation on post-hypoxia reoxygenation (**Extended Data Fig. 3b**). Ascorbate supplementation significantly reduced *HRG* inhibition by TBHP in three of the six genes considered (*ADH1*, *HB1* and *PDC1*, **Extended Data Fig. 6b**). Interestingly, ascorbate treatment in the absence of TBHP also elevated expression of *SAD6*, *SUS4* and *LBD41* compared to hypoxia alone. These results and their implications are

discussed in the text at lines 238-241.

iii) As mentioned above, it is important to further disentangle the ROS vs hypoxia responses. Authors should consider investigating ERF-VII stability and HRG expression dynamics in mutants and overexpressor lines that have altered endogenous ROS levels. E.g., how do they behave in calatalse mutants or overexpressors that have increased vs reduced H₂O₂ respectively?

We thank the reviewer for this additional suggested experiment. However, due to time and resource constraints, we opted to prioritise the experiments with exogenous ROS donors and scavengers as we considered that endogenously altered H₂O₂ levels may have potentially confounding effects on HRG expression, e.g. in development (Yang Z et al., 2019, PMID 30291629) making it difficult to disentangle outcomes from hypoxia/reoxygenation-specific effects.

iv) Do the authors think that ROS inactivation of PCOs is reversible? At some point the ERF-VIIs must need to be turned over following de-submergence and/or following the alleviation of elevated ROS. How would this happen? There is some discussion of antioxidants being unable to repair H₂O₂-inhibited PCOs in vitro, but this should be further expanded on.

Our *in vitro* assays show that PCO inactivation cannot be reversed by ascorbate, glutathione or DTT, suggesting that these reducing agents cannot readily penetrate to the PCO active site to reverse Cys or Fe oxidation. However, the reviewer is correct that in the cellular context, PCO activity must ultimately be restored. It could be that slow reversal of PCO damage can occur in a cellular context; cysteine sulfenylation in plant cells is reversed by thioredoxins and glutaredoxins. Interestingly, the cytosolic *Thioredoxin 8* gene is induced by hypoxia, but not by TBHP, in an ERFVII-dependent manner (**Supplemental Tables 1, 2**). Three CC-type glutaredoxins are instead significantly repressed in hypoxia and induced in reoxygenation (*GRX480*, *AT3G62950* and *AT4G15690*), although only *GRX480* among them requires the ERFVIIs for induction in reoxygenation (**Supplemental Tables 1, 2**). A separate study conducted by some of the authors of this manuscript also investigated the role of hydrogen sulfide (H₂S), which accumulates in hypoxia in plant cells (Zhou et al. 2021, 10.1007/s11104-021-05091-9), and can persulfidate Cys172. Here, it has been proposed that H₂S plays a protective role in maintaining AtPCOs functionality under oxidative conditions (Telara et al., 2025, 10.1101/2025.11.05.686772), as persulfidation can be reversed. Ultimately, even in the case that existing PCOs are permanently inactivated, newly synthesised ones will be active and promote ERFVII degradation. The reversibility of PCO inactivation is discussed in lines 376-380 of the Discussion.

v) A previous study in mammals (PMID: 34893540) has looked at crosstalk between oxygen and ROS within the context of N-degron pathway substrate turnover; specifically, RGS4. In this paper, it is shown that the N-terminal cysteine of RGS4 is a dual sensor for both oxygen and oxidative stress, and that this is linked to the duration of hypoxia and can result in targeting to different degradation pathways. Have authors considered how and if ROS might directly affect the N-degron of ERF-VIIs? This mammalian study should be more directly discussed in this work, as it could also have direct implications for the situation in plants given the broad conservation of the N-degron pathway in these two different lineages.

The work by Heo et al. (PMID: 34893540) is of interest as they find that in prolonged hypoxia, ROS oxidise the N-terminal Cys of N-degron substrate RGS4 to Cys-sulfonic acid, redirecting these substrates for lysosomal autophagy. We investigated whether our ERFVII peptides were similarly oxidised by TBHP treatment (non-enzymatically) and found that this did occur but to negligible levels (~1 μM CysO₃ formation following treatment of 200 μM peptide with 1 mM H₂O₂, **Fig. 4 and Extended Data Fig. 4**). The immediate impact of ROS on ERFVIIs is therefore to increase their

stability through inhibited PCO activity. It is feasible that over time, non-enzymatic ERFVII N-terminal Cys oxidation to CysO3 could play a role in ERFVII degradation. We have included this in the Discussion, lines 376-380. Zubrycka et al (2023, PMID 37537157) have recently reported that CysO3 is found in Nt-arginylated ERFVII substrates from wheat germ lysates; it will be of interest to further understand the potential modifications at ERFVII Nt-Cys residues, though out of the scope of this manuscript.

Reviewer 2

1) The process of recovery from reoxygenation was compared in seven-day old Arabidopsis wild type and erfVII mutant seedlings. While the data overall are convincing, the measurements appear to have been performed over different timescales during the hypoxia/reoxygenation treatment. For example, while no immediate differences were observed, decreased survival was observed in the erfVII seedlings after 4 days of reoxygenation following hypoxia. However, ROS staining procedures revealed greater hydrogen peroxide accumulation (as shown by increased DAB staining) in the erfVII seedlings compared to wild type after 1 h. However, data using the roGFP2-Orp1 probe showed that sensor oxidation upon reoxygenation which was significantly greater in erfVII mutant than Col-0 plants, but this increase was only significantly greater at the later stages of severe hypoxia. Such differences make it hard to align the different observations.

We thank the second reviewer for this comment, which was also raised by the first reviewer in multiple instances. Indeed we have described above how the length and intensity of hypoxia treatment is important in determining ERFVII stability on reoxygenation, likely due to accumulation of ROS over time. We hope that clarifications in the revised text are helpful but also clarify some further points here:

Root tip/plant damage is observed several days after H₂O₂ treatments because of the time required by the damaged cell components to cause arrest of essential biological functions (cell death). The stronger effect of strict hypoxia (0.1% O₂) compared with milder treatments (1% O₂) is used in this study when we want to examine the ability of the plant to tolerate ROS accumulation (arising from severe hypoxia), or where we want to see whether the effects can be prevented with antioxidants/scavengers. Severe hypoxia causes substantial H₂O₂ accumulation (**Fig. 1f-j**) and, followed by reoxygenation, causes plant death (**Fig. 1b,c**). Mild hypoxia is used when we wanted to examine molecular responses to the treatment without the risk of affecting cell viability. As explained in the reply to reviewer 1, we selected for our tolerance assays oxygen concentrations similar to those reported for submerged plants (Vashist D et al (2010) New Phytologist 190: 299). Moreover, in our experimental conditions, treatments of up to 3 days with 1% O₂ do not kill Arabidopsis seedlings grown on sucrose-supplemented media.

The dynamics of H₂O₂ accumulation in Arabidopsis seedlings following reoxygenation are not substantially different when measured by DAB-staining or by the roGFP2-Orp1 reporter, showing a decline after 6 h. The short half-life of H₂O₂ in living cells, measured and calculated in the range of milliseconds to tens of seconds (depending on the cell compartment considered), supports such dynamics (Petrov and Van Breusegem, 2012, PMID: 22708052).

2) Further analysis of the contribution of ROS to PCO-ERFVII mediated responses to hypoxia and reoxygenation showed that ERFVIIs remain stable and localised in plant nuclei and that this finding is linked to ROS-mediated PCO-inactivation. The data shown in some of the figures such as Figure 2 is hard to judge because the figures are so small. It might aid clarity and easier to

visualise the intracellular localisation of the GFP/stains, if key parts of the figures were enlarged.

We thank the reviewer for this helpful suggestion. To improve clarity and facilitate visualization of intracellular localization, we have enlarged the relevant regions of the confocal images in Figure 2 by including a zoomed-in panel highlighting key areas of GFP and stain localization.

3) It is not clear why the authors used tert-butyl hydroperoxide was used to induce cysteine oxidation. Did the authors try other oxidants to confirm the observed effects?

We used TBHP as an alternative to H₂O₂ because of its higher stability in solution and during incubations (Munhoz and Netto, 2004, [10.1074/jbc.M313773200](https://doi.org/10.1074/jbc.M313773200)), thus allowing more reproducible dosing and longer experiments without rapid loss of oxidant. Moreover, TBHP is more hydrophobic than H₂O₂ and therefore it crosses biological membranes more readily, leading to uniform intracellular oxidation (Kučera et al., 2014, PMID: [24847414](https://pubmed.ncbi.nlm.nih.gov/24847414/)). However, the Reviewer is correct, it is also valuable to test other oxidants. We therefore measured ERFVII stability (with our nLuc reporters) using other sources of ROS, including antimycin A, diuron, arsenite, cadmium, high light and methyl viologen (**Extended Data Fig. 3c-g**). Only the first two caused RAP2.3 stabilisation, suggesting that ERFVII stabilisation is induced selectively by specific ROS signals. This is now described in the text at lines 152-155.

Reviewer 3

This study provides valuable information to help us understand how ERF-VII proteins contribute to increased tolerance to reoxygenation, even in the presence of oxygen. It has been recognized that master regulators of submergence and hypoxia tolerance, ERF-VIIs, play an essential role in the adaptation to reoxygenation, but it was unclear how they work because these proteins are degraded in the presence of oxygen. The data presented in this study show a potential mechanism for stabilizing ERF-VII proteins by hydrogen peroxide. My comments are as follows.

1) Figure 1 shows that ERF-VIIs suppress ROS accumulation under reoxygenation. Figures 2 and 3 indicate that hydrogen peroxide regulates the localization of RAP2.12 to the nucleus and the stabilization of RAP2.3. I was wondering whether the level of hydrogen peroxide suppressed by ERF-VIIs in wild-type plants was sufficient to localize and stabilize these proteins under reoxygenation. I agree that the exogenous application of hydrogen peroxide has these effects. However, other than hydrogen peroxide, another factor may be responsible for the localization and stabilization of ERF-VIIs under reoxygenation because only a minimal amount of hydrogen peroxide is accumulated in reoxygenated samples. The use of antioxidants and ROS scavengers in reoxygenated plants/samples must help evaluate the role of hydrogen peroxide.

We thank also this reviewer for their comments and suggestion. In the revised version of this manuscript, we show the effect of antioxidants/ROS scavengers on the stability of ERFVIIs (using the RAP2.3nLUC reporter, **Extended Data Fig. 3a,b**) and the consequent regulation of HRGs (**Extended Data Fig. 6b**). These data confirm that (selectively sourced) ROS are responsible for the effects observed. These results are discussed in the text at lines 149-152 and 238-241.

2) Reduced activity of PCO4 by hydrogen peroxide is a key mechanism to explain the stabilization of ERF-VII. However, this data is generated by in vitro assays using recombinant AtPCO4. Is there any way to assay PCO activity using plant extracts? It is unclear if PCO activity is also reduced in

reoxygenated plants as observed in the in vitro system.

Unfortunately, there is no assay to directly measure PCO4 activity *in vivo*. While we are interested in establishing such a method, and are applying for grants to support this, the research community needs to rely on indirect measurements of PCO activity outputs, such as substrate stability and/or (transcriptional) activity.

3) Figure 5C shows ChIP-qPCR results, but the difference between test samples and negative controls is not evaluated by a statistical method.

We thank the reviewer for this suggestion. We have now added the result of this statistical evaluation as Supplementary Table 9.

4) It is exciting that ERF-VIIs physically interact with the promoter regions of both hypoxia- and ROS-responsive genes under reoxygenation, but only the latter is induced. Identifying the regulatory factors involved in this mechanism significantly increases the quality of this paper. This study shows a possible mechanism for stabilizing ERF-VIIs in reoxygenated plants, which is valuable information to advance our understanding of the molecular basis of hypoxia-reoxygenation tolerance in plants. However, more critical information (e.g., the mechanisms underlying selective expression of hypoxia- and ROS-responsive genes) may need to be considered for publication in Nature.

In response to the reviewer's request, which also coincides with two points raised by reviewer 1, we explored several plausible mechanisms underlying the regulation of ERFVII activity under oxidative stress. We excluded the involvement of HRA1 (Giuntoli et al., 2017, 10.3389/fpls.2017.00591) and of hypoxia-responsive gene (HRG) products, as HRGs are not induced by H₂O₂ and independent *hra1* mutants did not alter HRG repression upon reoxygenation. We also tested the proposed role of the ERF transcription factor ORA59 (Brunello et al., 2024, 10.1093/plphys/kiae677), but neither knockout nor silenced ORA59 lines showed altered HRG expression under our experimental conditions.

More informative results were obtained using transgenic lines expressing truncated versions of RAP2.12. Progressive removal of C-terminal motifs revealed that truncations lacking CMVII-8 lost the ability to repress HRGs under oxidative stress, indicating that these domains are required for RAP2.12-mediated repression in response to H₂O₂.

Together with RNA-seq data showing that ERFVIIIs activate a distinct set of redox-related genes lacking HRPEs, these results suggest a context-dependent, redox-regulated function of RAP2.12. We propose that oxidative stress modulates RAP2.12 activity either through interactions with promoter-specific cofactors or through redox-dependent changes in DNA binding or transcriptional complex assembly.

For additional details, we refer to the text posted in reply to comments 9) and 10) from Reviewer 1.

Response to Reviewers

Ref. Manuscript 2024-11-25256A

Dear Dr ,

Thank you very much for your consideration of our revised manuscript, originally titled 'H₂O₂ repurposes the plant oxygen-sensing machinery to control the transcriptional response to oxidative stress'. We are delighted that you are, in principle, able to publish it! We are also very grateful to the Reviewers for their careful initial assessment of the manuscript which guided us to improve its quality, as well as their positive reassessment.

We detail below our responses to the Reviewers' remaining concerns, including highlighting changes we have made to the manuscript. We have uploaded our newly revised manuscript, as well as a version in which the changes to iteration 2024-11-25256A are highlighted for easy reference.

Reviewer 1

The authors have done a considerable amount of work to respond to the specific points i raised following initial submission of the manuscript. Based on the additional experimetnal data and associated additions to the text, i believe the manuscript to be significantly improved, and that there is now more robust support for the main findngs of the study. A minor remaining suggestion is that the title of the report could be revised to more accurately reflect the nature of the work. As it stands, the title implies a broad role for ERF-VIIs in regulating responses to oxidative stress, yet the study really only supports this to be the case for context-specific oxidative stress that occurs following hypoxia or de-submergence.

We agree that the title infers a broader scope for the effect we describe than we can experimentally support. Therefore, we have changed the title from 'H₂O₂ repurposes the plant oxygen-sensing machinery to control the transcriptional response to oxidative stress' to 'H₂O₂ repurposes plant O₂ sensing to regulate post-hypoxia responses' to reflect the context of the oxidative stress that we studied. We also shortened the title to respect the word limit requested by the editorial checklist.

Reviewer 2

Oxygen is an immensely versatile molecule that fulfills crucial life functions underpinning cell metabolism. The roles of Plant Cysteine Oxidases (PCOs) and group VII Ethylene Response Factors (ERFVIIs) in hypoxia sensing is well characterized, but to date the relationships between oxygen signalling through the Cys/Arg branch of the N-degron pathway for protein degradation and the redox signalling pathways driven by reactive oxygen species (ROS), have not been characterized. This manuscript addresses this question directly. Although there may be many points of reciprocal control that remain to be elucidated, the data presented in this manuscript point to a novel regulatory control point. The paper provides evidence that PCO functions are modified in the presence of hydrogen peroxide. Data are presented showing that inhibition of PCO activity occurs in the presence of hydrogen peroxide leading to ERFVII stabilisation. Intriguingly, the data indicate that oxidation causes only minor ERFVII-mediated modifications in the expression of hypoxia responsive genes and is suggested to alter ERFVII function from positive to negative regulation of

gene transcription.

The inhibition of PCO activity appears to occur by a combination of oxidation of the active site Fe(II) to Fe(III), and also oxidation of Cys residues in or near the active site. This finding would also suggest that superoxide might also catalyse PCO inhibition through oxidation of the active site Fe(II) to Fe(III). Conversely, only hydrogen peroxide can cause Cys oxidation. Hence, superoxide and hydrogen peroxide might modify PCO functions by mechanism that could lead to slightly different outcomes, for example in terms of protein/protein interactions.

We thank the Reviewer for these comments. The potential differential role of superoxide and hydrogen peroxide in PCO inhibition may indeed be of interest to pursue further in a new study. Nevertheless, superoxide is short-lived in plant cells and hydrogen peroxide represents the more persistent form of reactive oxygen species (Akter S et al (2021) RSC Chem Biol 2: 1384); we would hypothesise that superoxide-mediated PCO inhibition is therefore only likely under very limited circumstances.

Reviewer 3

I have re-evaluated the revised manuscript, including the rebuttal and newly added data/figures. The revision is improved and addresses the main issues I raised in the first round, particularly by strengthening the causal link between ROS/H₂O₂ and ERF-VII behavior during reoxygenation/oxidative stress, and by improving statistical support for key chromatin-binding conclusions. The remaining concerns are minor and largely relate to framing and clarity.

Comments on revisions

(1) My original concern was whether ROS/H₂O₂ is causal under the relevant reoxygenation context, rather than an artefact of exogenous oxidant treatments. The authors now add antioxidant/scavenger experiments (ascorbate), showing restricted RAP2.3 reporter accumulation after TBHP and during post-hypoxia reoxygenation, with corresponding effects on a subset of transcriptional outputs. This meaningfully strengthens the causal interpretation and addresses my main request.

Minor revision: add 1–2 sentences in the Discussion noting that antioxidant treatments can be pleiotropic, while emphasizing that the consistent directionality across reporter/protein and transcriptional readouts supports a ROS-dependent mechanism.

We have added the following statement to the Discussion (lines 366-368 in the version with changes highlighted): *'The ROS-dependence of these effects was supported with antioxidant treatments, and although these can have pleiotropic effects, collectively the data consistently support a role for ERFVIs in post-hypoxic stress (Extended Data Figure 6b).'*

(2) I previously asked whether reduced PCO activity could be demonstrated directly in vivo (e.g., in plant extracts). The authors indicate that a direct in planta enzymatic assay is not currently available and thus rely on recombinant PCO biochemistry together with in vivo functional readouts (substrate stability/localization/transcription). This limitation is acceptable provided it is framed clearly.

Minor revision: explicitly state (if not already) that direct quantification of PCO catalytic activity in planta remains technically challenging/unavailable, and clarify that the conclusion is supported by recombinant PCO biochemistry plus strong in vivo consequences.

We agree that explicitly stating that we cannot measure PCO activity is a good idea, and helpful for those outside the field to understand why we have not done this experiment. We have therefore added the following statement to the Results section (lines 190-191 of the version with changes highlighted): *'Since direct quantification of PCO catalytic activity in planta remains technically challenging, we...'* We then emphasise the dual approach of biochemistry and in vivo work with the following statement in the Discussion (lines 345-346 of the version with changes highlighted): *'Using a combination of in vitro biochemical assays and in vivo reporter assays,...'*

(3) I requested statistical testing to support the claim that ChIP-qPCR signals exceed negative controls. The authors have added statistical analysis (with reporting in supplementary materials/legend), which resolves this point.

We thank the Reviewer for confirming this point.

(4) I also asked for clearer mechanistic insight into how ERF-VIIs can associate with promoters of hypoxia genes yet show repression while enabling oxidative stress programs during reoxygenation. The revision provides progress by adding new data and strengthening the narrative, including domain-level evidence that specific RAP2.12 C-terminal motifs contribute to ROS-dependent transcriptional behavior. While the precise determinants of promoter selectivity/cofactor usage remain incomplete, the added mechanistic detail improves the study.

Minor revision: briefly clarify in the Discussion that promoter selectivity/cofactor mechanisms during reoxygenation remain to be defined, framing the new domain-level data as important progress while acknowledging this as a future direction.

We thank the Reviewer for appreciating the additional insights we were able to gain into the mechanism. We agree it is useful to acknowledge that additional work is required to fully understand how ERFVII function is reconfigured on reoxygenation, and while we speculated on a potential mechanism, we adjusted lines 409-414 of the Discussion to reflect the need for further work: *'Although the exact mechanism remains to be defined, it is tempting to speculate that CMVII-8 contacts the mediator complex following the acidic exposure model. The neighbouring CMVII-4 and 7 have been shown to support the AP2 domain in binding the MED25 subunit. The nuclear redox milieu could determine the composition of the mediator complex that interact with ERFVII, as shown in animal and plant cells ultimately imparting activation or repressive capacity. Future studies can shed light on this aspect.'*